# Hybrid Local SGD for Federated Learning with Heterogeneous Communications

**Yuanxiong Guo**
The University of Texas at San Antonio
San Antonio, Texas 78249 USA
`yuanxiong.guo@utsa.edu`

**Ying Sun**
Pennsylvania State University
State College, PA 16801 USA
`ysun@psu.edu`

**Rui Hu & Yanmin Gong**
The University of Texas at San Antonio
San Antonio, Texas 78249 USA
`{rui.hu@my.,yanmin.gong@}utsa.edu`

## Abstract

Communication is a key bottleneck in federated learning where a large number of edge devices collaboratively learn a model under the orchestration of a central server without sharing their own training data. While local SGD has been proposed to reduce the number of FL rounds and become the algorithm of choice for FL, its total communication cost is still prohibitive when each device needs to communicate with the remote server repeatedly for many times over bandwidth-limited networks. In light of both device-to-device (D2D) and device-to-server (D2S) cooperation opportunities in modern communication networks, this paper proposes a new federated optimization algorithm dubbed hybrid local SGD (HL-SGD) in FL settings where devices are grouped into a set of disjoint clusters with high D2D communication bandwidth. HL-SGD subsumes previous proposed algorithms such as local SGD and gossip SGD and enables us to strike the best balance between model accuracy and runtime. We analyze the convergence of HL-SGD in the presence of heterogeneous data for general nonconvex settings. We also perform extensive experiments and show that the use of hybrid model aggregation via D2D and D2S communications in HL-SGD can largely speed up the training time of federated learning.

## 1 Introduction

Federated learning (FL) is a distributed machine learning paradigm in which multiple edge devices or clients cooperate to learn a machine learning model under the orchestration of a central server, and enables a wide range of applications such as autonomous driving, extended reality, and smart manufacturing (Kairouz et al., 2021). Communication is a critical bottleneck in FL as the clients are typically connected to the central server over bandwidth-limited networks. Standard optimization methods such as distributed SGD are often not suitable in FL and can cause high communication costs due to the frequent exchange of large-size model parameters or gradients. To tackle this issue, local SGD, in which clients update their models by running multiple SGD iterations on their local datasets before communicating with the server, has emerged as the de facto optimization method in FL and can largely reduce the number of communication rounds required to train a model (McMahan et al., 2017; Stich, 2019).

However, the communication benefit of local SGD is highly sensitive to non-iid data distribution as observed in prior work (Rothchild et al., 2020; Karimireddy et al., 2020). Intuitively, taking many local iterations of SGD on local dataset that is not representative of the overall data distribution will lead to local over-fitting, which will hinder convergence. In particular, it is shown in (Zhao et al., 2018) that the convergence of local SGD on non-iid data could slow down as much as proportionally to the number of local iteration steps taken. Therefore, local SGD with a large aggregation period

can converge very slow on non-iid data distribution, and this may nullify its communication benefit (Rothchild et al., 2020).

Local SGD assumes a star network topology where each device connects to the central server for model aggregation. In modern communication networks, rather than only communicating with the server over slow communication links, devices are increasingly connected to others over fast communication links. For instance, in 5G-and-beyond mobile networks, mobile devices can directly communicate with their nearby devices via device-to-device links of high data rate (Asadi et al., 2014; Yu et al., 2020). Also, edge devices within the same local-area network (LAN) domain can communicate with each other rapidly without traversing through slow wide-area network (WAN) (Yuan et al., 2020). This gives the potential to accelerate the FL convergence under non-iid data distribution by leveraging fast D2D cooperation so that the total training time can be reduced in FL over bandwidth-limited networks.

Motivated by the above observation, this paper proposes hybrid local SGD (HL-SGD), a new distributed learning algorithm for FL with heterogeneous communications, to speed up the learning process and reduce the training time. HL-SGD extends local SGD with fast gossip-style D2D communication after local iterations to mitigate the local over-fitting issue under non-iid data distribution and accelerate convergence. A hybrid model aggregation scheme is designed in HL-SGD to integrate both fast device-to-device (D2D) and slow device-to-server (D2S) cooperations. We analyze the convergence of HL-SGD in the presence of heterogeneous data for general nonconvex settings, and characterize the relationship between the optimality error bound and algorithm parameters. Our algorithm and analysis are general enough and subsume previously proposed SGD variations such as distributed SGD, local SGD and gossip SGD.

Specifically, we consider the FL setting in which all devices are partitioned into disjoint clusters, each of which includes a group of connected devices capable of communicating with each other using fast D2D links. The clustering can be a natural result of devices belonging to different LAN domains so that those devices connected to the same LAN domain are considered as one cluster. In another example, clustering is based on the geographic locations of mobile devices so that devices in a cluster are connected to each other through D2D communication links.

In summary, the paper makes the following main contributions:

• We propose a novel distributed learning algorithm for FL called HL-SGD to address the communication challenge of FL over bandwidth-limited networks by leveraging the availability of fast D2D links to accelerate convergence under non-iid data distribution and reduce training time.

• We provide the convergence analysis of HL-SGD under general assumptions about the loss function, data distribution, and network topology, generalizing previous results on distributed SGD, local SGD, and gossip SGD.

• We conduct extensive empirical experiments on two common benchmarks under realistic network settings to validate the established theoretical results of HL-SGD. Our experimental results show that HL-SGD can largely accelerate the learning process and speed up the runtime.

## 2 BACKGROUND AND RELATED WORK

Large-scale machine learning based on distributed SGD has been well studied in the past decade, but often suffers from large network delays and bandwidth limits (Bottou et al., 2018). Considering that communication is a major bottleneck in federated settings, local SGD has been proposed recently to reduce the communication frequency by running SGD independently in parallel on different devices and averaging the sequences only once in a while (Stich, 2019; Lin et al., 2019; Haddadpour et al., 2019; Yu et al., 2019; Wang et al., 2021). However, they all assume the client-server architecture and do not leverage the fast D2D communication capability in modern communication networks. Some studies (Liu et al., 2020; Abad et al., 2020; Castiglia et al., 2021) develop hierarchical FL algorithms that first aggregate client models at local edge servers before aggregating them at the cloud server or with neighboring edge servers, but they still rely on D2S communication links only and suffer from the scalability and fault-tolerance issues of centralized setting. On the other hand, while existing works on decentralized or gossip SGD consider D2D communications (Tsitsiklis, 1984; Boyd et al., 2006), they assume a connected cluster with homogeneous communication links and will converge

very slow on the large and sparse network topology that is typically found in FL settings. Unlike previous works, HL-SGD leverages both D2S and D2D communications in the system.

Some recent studies aim to encapsulate variants of SGD under a unified framework. Specifically, a cooperative SGD framework is introduced in (Wang & Joshi, 2021) that includes communication reduction through local SGD steps and decentralized mixing between clients under iid data distribution. A general framework for topology-changing gossip SGD under both iid and non-iid data distributions is proposed in (Koloskova et al., 2020). Note that all of the above works assume undirected network topology for communications in every iteration. In comparison, our proposed HL-SGD is different: the D2S communication is asymmetric due to the use of device sampling and model broadcasting in each global aggregation round and cannot be modeled in an undirected graph. Therefore, the convergence analysis of HL-SGD does not fit into the prior frameworks and is much more challenging. Moreover, our major focus is on the runtime of the algorithm rather than its convergence speed in iterations.

## 3    SYSTEM MODEL

In this section, we introduce the FL system model, problem formulation, and assumptions we made.

**Notation.** All vectors in this paper are column vectors by default. For convenience, we use $\mathbf{1}$ to denote the all-ones vector of appropriate dimension, $\mathbf{0}$ to denote the all-zeros vector of appropriate dimension, and $[n]$ to denote the set of integers $\{1, 2, \ldots, n\}$ with any positive integer $n$. Let $\|\cdot\|$ denote the $\ell_2$ vector norm and Frobenius matrix norm and $\|\cdot\|_2$ denote the spectral norm of a matrix.

We consider a FL system consisting of a central server and $K$ disjoint clusters of edge devices. Devices in each cluster $k \in [K]$ can communicate with others across an undirected and connected graph $\mathcal{G}_k = (\mathcal{V}, \mathcal{E}_k)$, where $\mathcal{V}_k$ denotes the set of edge devices in the cluster, and edge $(i, j) \in \mathcal{E}_k$ denotes that the pair of devices $i, j \in \mathcal{V}_k$ can communicate directly using D2D as determined by the communication range of D2D links. Besides, each device can directly communicate with the central server using D2S links. Denote the set of all devices in the system as $\mathcal{V} := \bigcup_{k \in [K]} \mathcal{V}_k$, the number of devices in each cluster $k \in [K]$ as $n := |\mathcal{V}_k|$, and the total number of devices in the system as $N := \sum_{k \in [K]} n$ [1].

The FL goal of the system is to solve an optimization problem of the form:

$$\min_{x \in \mathbb{R}^d} f(x) := \frac{1}{N} \sum_{i \in \mathcal{V}} f_i(x) := \frac{1}{K} \sum_{k \in [K]} \bar{f}_k(x), \tag{1}$$

where $f_i(x) := \mathbb{E}_{z \sim \mathcal{D}_i}[\ell_i(x; z)]$ is the local objective function of device $i$, $\bar{f}_k(x) := (1/n) \sum_{i \in \mathcal{V}_k} f_i(x)$ is the local objective function of cluster $k$, and $\mathcal{D}_i$ is the data distribution of device $i$. Here $\ell_i$ is the (non-convex) loss function defined by the learning model and z represents a data sample from data distribution $\mathcal{D}_i$.

When applying local SGD to (1) in FL with heterogeneous communications, the communications between the server and devices in FL are all through D2S links that are bandwidth-limited, particularly for the uplink transmissions. Therefore, the incurred communication delay is high. Due to the existing of high-bandwidth D2D links that are much more efficient than low-bandwidth D2S links, it would be highly beneficial if we can leverage D2D links to reduce the usage of D2S links such that the total training time can be reduced. This motivates us to design a new learning algorithm for FL with heterogeneous communications.

## 4    HYBRID LOCAL SGD

In this section, we present our HL-SGD algorithm suitable for the FL setting with heterogeneous communications. Algorithm 1 provides pseudo-code for our algorithm.

At the beginning of $r$-th global communication round, the server broadcasts the current global model $x^r$ to all devices in the system via cellular links (Line 4). Note that in typical FL systems, the down-

---

[1]For presentation simplicity, we assume each cluster contains the same number of devices here. The results of this paper can be extended to the case of clusters with different device numbers as well.

---

**Algorithm 1** HL-SGD: Hybrid Local SGD

---

**Input**: initial global model $x^0$, learning rate $\eta$, communication graph $\mathcal{G}_k$ and mixing matrix $W_k$ for all clusters $k \in [K]$, and fraction of sampled devices in each cluster $p$.
**Output**: final global model $x^R$

1: **for** each round $r = 0, \ldots, R-1$ **do**
2:     **for** each cluster $k \in [K]$ **in parallel do**
3:         **for** each device $i \in \mathcal{V}_k$ **in parallel do**
4:             $x_i^{r,0} = x^r$
5:             **for** $s = 0, \ldots, \tau-1$ **do**
6:                 Compute a stochastic gradient $g_i$ over a mini-batch $\xi_i$ sampled from $\mathcal{D}_i$
7:                 $x_i^{r,s+\frac{1}{2}} = x_i^{r,s} - \eta g_i(x_i^{r,s})$                             ▷ local update
8:                 $x_i^{r,s+1} = \sum_{j \in \mathcal{N}_i^k}(W_k)_{i,j} x_j^{r,s+\frac{1}{2}}$            ▷ gossip averaging
9:             **end for**
10:         **end for**
11:     **end for**
12:     **for** each cluster $k \in [K]$ **do**
13:         $m \leftarrow \max(p \cdot n, 1)$
14:         $S_k^r \leftarrow$ (random set of $m$ clients in $\mathcal{V}_k$)         ▷ device sampling
15:     **end for**
16:     $x^{r+1} = \frac{1}{K}\sum_{k \in [K]}\frac{1}{m}\sum_{i \in S_k^r} x_i^{r,\tau}$         ▷ global aggregation
17: **end for**
18: **return** $x^R$

---

link communication is much more efficient than uplink communication due to the larger bandwidth allocation and higher data rate. Therefore, devices only consume a smaller amount of energy when receiving data from the server compared with transmitting data to the server.

After that, devices in each cluster initialize their local models to be the received global model and run $\tau$ iterations of gossip-based SGD via D2D links to update their local models in parallel (lines 5–9). Let $x_i^{r,s}$ denote the local model of device $i$ at the $r$-th local iteration of $s$-th round. Here each gossip-based SGD iteration consists of two steps: (i) SGD update, performed locally on each device (lines 6–7), followed by a (ii) gossip averaging, where devices average their models with their neighbors (line 8). In the gossip averaging protocol, $\mathcal{N}_i^k$ denotes the neighbors of device $i$, including itself, on the D2D communication graph $\mathcal{G}_k$ of cluster $k$, and $W_k \in [0,1]^{n \times n}$ denotes the mixing matrix of cluster $k$ with each element $(W_k)_{i,j}$ being the weight assigned by device $i$ to device $j$. Note that $(W_k)_{i,j} > 0$ only if devices $i$ and $j$ are directly connected via D2D links.

Next, a set $S_k^r$ of $m$ devices are sampled uniformly at random (u.a.r.) with probability $p$ without replacement from each cluster $k \in [K]$ by the server (lines 13–14), and their final updated local models $\{x_i^{r,\tau}, \forall i \in S_k^r\}$ are sent to the server via D2S links. After that, the server updates the global model $x^{r+1}$ by averaging the received local models from all sampled devices (line 16). Note that only $m$ devices per cluster will upload their models to the server in each round to save the usage of expensive D2S uplink transmissions. The intuition is that after multiple iterations of gossip-based SGD, devices have already reached approximate consensus within each cluster, and the sampled average can well represent the true average. By trading D2D local aggregation for D2S global aggregation, the total communication cost can be reduced. We will empirically validate such benefits later in the experiments.

It is worth noting that HL-SGD inherits the privacy benefits of classic FL schemes by keeping the raw data on device and sharing only model parameters. Moreover, HL-SGD is compatible with existing privacy-preserving techniques in FL such as secure aggregation (Bonawitz et al., 2017; Guo & Gong, 2018), differential privacy (McMahan et al., 2018; Hu et al., 2020; 2021), and shuffling (Girgis et al., 2021) since only the sum rather than individual values is needed for the local and global model aggregation steps.

**Runtime analysis of HL-SGD.** We now present a runtime analysis of HL-SGD. Here we ignore the communication time of downloading models from the server by each device since the download bandwidth is often much larger than upload bandwidth for the D2S communication in practice (**?**). In each round of HL-SGD, we denote the *average* time taken by a device to compute a local update,

perform one round of D2D communication and one round of D2S communication as $c_{\text{cp}}$, $c_{\text{d2d}}$ and $c_{\text{d2s}}$, respectively. Assume the uplink bandwidth between the server and devices is fixed and evenly shared among the sampled devices in each round, then $c_{\text{d2s}}$ is linearly proportional to the sampling ratio $p$. Similarly, $c_{\text{cp}}$ depends on the D2D network topology $\mathcal{G}_k$ and typically increases with the maximum node degree $\Delta(\mathcal{G}_k)$. The total runtime of HL-SGD after $R$ communication rounds is

$$R \times [\tau \times (c_{\text{cp}} + c_{\text{d2d}}) + c_{\text{d2s}}]. \tag{2}$$

The specific values of $c_{\text{cp}}$, $c_{\text{d2d}}$ and $c_{\text{d2s}}$ depend on the system configurations and applications. In comparison, the total runtime of local SGD after $R$ communication rounds is $R \times [\tau \times c_{\text{cp}} + c_{\text{d2s}}]$.

**Previous algorithms as special cases.** When devices do not communicate with each other, i.e., $W_k = I, \forall k \in [K]$, and sampling ratio $p = 1$, HL-SGD reduces to distributed SGD (when $\tau = 1$) or local SGD (when $\tau > 1$) where each device only directly communicates with the server with D2S links. Also, when $\tau \to \infty$, HL-SGD reduces to gossip SGD where devices only cooperate with their neighboring devices through a gossip-based communication protocol with D2D links to update their models without relying on the server. Therefore, HL-SGD subsumes existing algorithms and enables us to strike the best balance between runtime and model accuracy by tuning $\tau$, $W_k$, and $p$. However, due to the generality of HL-SGD, there exist significantly new challenges in its convergence analysis, which constitutes one of the main contributions of this paper as elaborated in the following section.

## 5 CONVERGENCE ANALYSIS OF HL-SGD

In this section, we analyze the convergence of HL-SGD with respect to the gradient norm of the objective function $f(\cdot)$, specifically highlighting the effects of $\tau$ and $p$. Before stating our results, we make the following assumptions:

**Assumption 1** (Smoothness). *Each local objective function $f_i : \mathbb{R}^d \to \mathbb{R}$ is $L$-smooth for all $i \in \mathcal{V}$, i.e., for all $x, y \in \mathbb{R}^d$,*

$$\|\nabla f_i(x) - \nabla f_i(y)\| \le L\|x - y\|, \quad \forall i \in \mathcal{V}.$$

**Assumption 2** (Unbiased Gradient and Bounded Variance). *The local mini-batch stochastic gradient in Algorithm 1 is unbiased, i.e., $\mathbb{E}_{\xi_i}[g_i(x)] = \nabla f_i(x)$, and has bounded variance, i.e., $\mathbb{E}_{\xi_i}\|g_i(x) - \nabla f_i(x)\|^2 \le \sigma^2, \forall x \in \mathbb{R}^d, i \in \mathcal{V}$, where the expectation is over all the local mini-batches.*

**Assumption 3** (Mixing Matrix). *For any cluster $k \in [K]$, the D2D network is strongly connected and the mixing matrix $W_k \in [0,1]^{n \times n}$ satisfies $W_k \mathbf{1} = \mathbf{1}$, $\mathbf{1}^\top W_k = \mathbf{1}^\top$, $\text{null}(I - W_k) = \text{span}(\mathbf{1})$. We also assume $\|W_k - (1/n)\mathbf{1}\mathbf{1}^\top\|_2 \le \rho_k$ for some $\rho_k \in [0,1)$.*

**Assumption 4** (Bounded Intra-Cluster Dissimilarity). *There exists a constant $\epsilon_k \ge 0$ such that $(1/n)\sum_{i \in \mathcal{V}_k}\|\nabla f_i(x) - \nabla \bar{f}_k(x)\|^2 \le \epsilon_k^2$ for any $x \in \mathbb{R}^d$ and $k \in [K]$. If local functions are identical to each other within a cluster, then we have $\epsilon_k = 0$.*

**Assumption 5** (Bounded Inter-Cluster Dissimilarity). *There exist constants $\alpha \ge 1$, $\epsilon \ge 0$ such that $(1/K)\sum_{k \in [K]}\|\nabla \bar{f}_k(x)\|^2 \le \alpha^2\|\nabla f(x)\|^2 + \epsilon_g^2$ for any $x \in \mathbb{R}^d$. If local functions are identical to each other across all clusters, then we have $\alpha = 1$, $\epsilon_g = 0$.*

Assumptions 1–3 are standard in the analysis of SGD and decentralized optimization (Bottou et al., 2018; Koloskova et al., 2019). Assumptions 4–5 are commonly used in the federated optimization literature to capture the dissimilarities of local objectives (Koloskova et al., 2020; Wang et al., 2020).

### 5.1 MAIN RESULTS

We now provide the main theoretical results of the paper in Theorem 1 and Theorem 2. The detailed proofs are provided in the appendices. Define the following constants:

$$\rho_{\max} = \max_{k \in [K]} \rho_k, \qquad D_{\tau,\rho} = \min\left\{\frac{1}{1 - \rho_{\max}}, \tau\right\}, \qquad \bar{\epsilon}_L^2 = \frac{1}{K}\sum_{k=1}^{K}\epsilon_k^2 \tag{3}$$

and let

$$r_0 = 8(f(x^0) - f(x^\star)), \; r_1 = 16L\left(\frac{\sigma^2}{N}\right),$$

$$r_2 = 16C_1L^2\tau^2\epsilon_g^2 + 16C_1L^2\left(\tau\rho_{\max}^2 D_{\tau,\rho}\bar{\epsilon}_L^2 + \tau\sigma^2\left(\frac{1}{n} + \rho_{\max}^2\right)\right).$$
(4)

**Theorem 1** (Full device participation). *Let Assumptions 1–5 hold, and let $L$, $\sigma$, $\bar{\epsilon}_L$, $\epsilon_g$, $D_{\tau,\rho}$, $\rho_{\max}$, $r_0$, $r_1$, and $r_2$ be as defined therein. If the learning rate $\eta$ satisfies*

$$\eta = \min\left\{\frac{1}{4C_1\alpha}\cdot\frac{1}{\tau L}, \left(\frac{r_0}{r_1\tau R}\right)^{\frac{1}{2}}, \left(\frac{r_0}{r_2\tau R}\right)^{\frac{1}{3}}\right\},$$
(5)

*then for any $R > 0$, the iterates of Algorithm 1 with full device participation for HL-SGD satisfy*

$$\min_{r,s}\mathbb{E}\|\nabla f(\bar{x}^{r,s})\|^2 = O\left(\frac{\sigma}{\sqrt{N\tau R}} + \frac{\left(\tau^2\epsilon_g^2 + \tau\rho_{\max}^2 D_{\tau,\rho}\bar{\epsilon}_L^2 + \tau\left(\frac{1}{n} + \rho_{\max}^2\right)\sigma^2\right)^{\frac{1}{3}}}{(\tau R)^{\frac{2}{3}}} + \frac{1}{R}\right),$$
(6)

*where $\bar{x}^{r,s} = \frac{1}{N}\sum_{i=1}^{N}x_i^{r,s}$.*

In the following, we analyze the iteration complexity of HL-SGD and compare it with those of some classic and state-of-the-art algorithms relevant to our setting in Table 1. First, we consider two extreme cases of HL-SGD where $\rho_{\max} = 0$ and $\rho_k = 1, \forall k \in [K]$, and show that our analysis recovers the best known rate of local SGD.

**Fully Connected D2D networks.** In this case, $\rho_{\max} = 0$, and each cluster can be viewed as a single device, and thus HL-SGD reduces to local SGD with $K$ devices. Substuting $\rho_{\max} = 0$ into (6), the iteration complexity of HL-SGD reduces to $O(\sigma/\sqrt{N\tau R} + \left(\tau^2\epsilon_g^2 + \tau\cdot(\sigma^2/n)\right)^{1/3}/(\tau R)^{2/3} + 1/R)$. This coincides with the complexity of local SGD provided in Table 1 with device number $K$ and stochastic gradient variance $\sigma^2/n$ thanks to the fully intra-cluster averaging.

**Disconnected D2D networks.** In this case, HL-SGD reduces to local SGD with $N$ devices. Substituting $\rho_{\max} = 1$ into (6), the iteration complexity of HL-SGD becomes $O(\sigma/\sqrt{N\tau R} + \left(\tau^2(\epsilon_g^2 + \bar{\epsilon}_L^2) + \tau\sigma^2\right)^{1/3}/(\tau R)^{2/3} + 1/R)$. This coincides with the complexity of local SGD with $N$ devices, stochastic gradient variance $\sigma^2$, and gradient heterogeneity of order $\epsilon_g^2 + \bar{\epsilon}_L^2$.

Table 1: Comparison of Iteration Complexity. [2]

| | |
|---|---|
| Local SGD | $O\left(\frac{\sigma}{\sqrt{N\tau R}} + \frac{\left(\tau^2\epsilon^2 + \tau\sigma^2\right)^{\frac{1}{3}}}{(\tau R)^{\frac{2}{3}}} + \frac{\tau}{\tau R}\right)$ |
| Gossip SGD | $O\left(\frac{\sigma}{\sqrt{N\tau R}} + \frac{\rho^{\frac{2}{3}}\epsilon^{\frac{2}{3}}}{(\tau R)^{\frac{2}{3}}(1-\rho)^{\frac{2}{3}}} + \frac{\rho^{\frac{2}{3}}\sigma^{\frac{2}{3}}}{(\tau R)^{\frac{2}{3}}(1-\rho)^{\frac{1}{3}}} + \frac{\rho}{(1-\rho)\tau R}\right)$ |
| Gossip PGA (Chen et al., 2021) | $O\left(\frac{\sigma}{\sqrt{N\tau R}} + \frac{C_{\tau,\rho}^{\frac{1}{3}}D_{\tau,\rho'}^{\frac{1}{3}}\rho^{\frac{2}{3}}\epsilon^{\frac{2}{3}}}{(\tau R)^{\frac{2}{3}}} + \frac{C_{\tau,\rho}^{\frac{1}{3}}\rho^{\frac{2}{3}}\sigma^{\frac{2}{3}}}{(\tau R)^{\frac{2}{3}}} + \frac{\rho D_{\tau,\rho'}}{\tau R}\right)$ |
| HL-SGD (this work) | $O\left(\frac{\sigma}{\sqrt{N\tau R}} + \frac{\left(\tau^2\epsilon_g^2 + \tau\rho_{\max}^2 D_{\tau,\rho}\bar{\epsilon}_L^2\right)^{\frac{1}{3}}}{(\tau R)^{\frac{2}{3}}} + \frac{\left(\tau\left(\frac{1}{n} + \rho_{\max}^2\right)\sigma^2\right)^{\frac{1}{3}}}{(\tau R)^{\frac{2}{3}}} + \frac{\tau}{\tau R}\right)$ |

Next, we compare the complexities of HL-SGD, local SGD, gossip SGD and gossip PGA.

**Comparison to Local SGD.** Comparing (6) and the complexity of local SGD, we can see the intra-cluster D2D communication provably improves the iteration complexity by reducing the transient iterations. This is reflected in the smaller coefficient associated with the $O((\tau R)^{-2/3})$ term. In particular, improving D2D communication connectivity will lead to a smaller $\rho_{\max}$ and consequently, mitigate the impact of both local data heterogeneity and stochastic noise on the convergence rate.

---

[1]The convergence rates for gossip SGD and local SGD are from (Koloskova et al. (2020)). The parameters in the table are given by the following: $\sigma^2$: stochastic gradient variance; $\rho$: network connectivity; $\epsilon^2$: data heterogeneity of order $\epsilon_g^2 + \bar{\epsilon}_L^2$; $C_{\tau,\rho} \triangleq \sum_{k=0}^{\tau-1}\rho^k$, $D_{\tau,\rho'} = \min\{1/(1-\rho), \tau\}$. Note that $D_{\tau,\rho} \neq D_{\tau,\rho'}$.

**Comparison to Gossip SGD.** Under the condition that $\rho = \rho_{\max}$, i.e., the connectivity of D2D network in gossip SGD is the same as that of HL-SGD, Table 1 shows HL-SGD outperforms gossip SGD when $\tau/n \leq \rho^2/(1-\rho)$. In other words, HL-SGD is beneficial for weakly connected networks, which is the case in FL settings where a large number of devices are often loosely connected or disconnected into several disjoint clusters via D2D communications only.

**Comparison to Gossip PGA.** Gossip PGA improves local SGD by integrating gossiping among *all* devices in one round using a connected network. Compared to gossip SGD, gossip PGA has one extra full averaging step with period $\tau$. The complexity of gossip PGA improves both by reducing the transient iterations. HL-SGD (full participation) differs from gossip PGA in the sense that gossiping is performed within multiple clusters instead of a single one. The benefit comes from the fact that for many commonly used D2D network topologies, the spectral gap $1 - \rho$ decreases as the network size decreases, see Table 2. Therefore, when employing the same D2D network topology, HL-SGD enjoys a smaller connectivity number $\rho_{\max}$ than $\rho$. Considering the scenario where $\tau$ and $n$ are fixed while the cluster number $K$ grows, the total device number $N = nK$ grows and hence $\rho \to 1$ for gossip PGA. In the case when $\tau = D_{\tau,\rho'} \approx C_{\tau,\rho}$, the fastest decaying $O(1/\tau R)$ terms are comparable for both algorithms. However, the $O((\tau R)^{-2/3})$ term of gossip GPA can be larger than that of HL-SGD since $\rho$ increases with $N$. This observation shows for large-scale networks, it is advantageous to use HL-SGD with multiple connected clusters instead of gossip GPA with a single cluster under the D2D network topology.

Our next result shows the iteration complexity of HL-SGD with partial device participation. We assume the devices participate in synchronizing their models at the end of each FL round following the sampling rule given by Assumption 6.

**Assumption 6** (Sampling strategy). *Each $S_k^\tau$ contains a subset of $m$ indices uniformly sampled from $\{1, \ldots, n\}$ without replacement. Furthermore, $S_k^\tau$ is independent of $S_{k'}^{\tau'}$ for all $(k, r) \neq (k', r')$.*

**Theorem 2** (Partial device participation). *Let Assumptions 1–6 hold, and let $L$, $\sigma$, $\bar{\epsilon}_L$, $\epsilon_g$, $D_{\tau,\rho}$, $\rho_{\max}$, $r_0$, $r_1$, and $r_2$ be as defined therein. If the network connectivity satisfies*

$$\rho_{\max} \leq 1 - 1/\tau, \tag{7}$$

*then for suitably chosen learning rate $\eta$, the iterates of Algorithm 1 with partial device participation for HL-SGD satisfy*

$$\min_{r,s} \mathbb{E}\|\nabla f(\bar{x}^{r,s})\|^2$$

$$= O\left(\frac{\sigma + \mathcal{E}(\epsilon_g, \bar{\epsilon}_L, \sigma, \rho_{\max})}{\sqrt{N\tau R}} + \frac{\left(\tau^2\epsilon_g^2 + \tau\rho_{\max}^2 D_{\tau,\rho}\bar{\epsilon}_L^2 + \tau\left(\frac{1}{n} + \rho_{\max}^2\right)\sigma^2\right)^{\frac{1}{3}}}{(\tau R)^{\frac{2}{3}}} + \frac{\max\{1, G_p D_{\tau,\rho}\rho_{\max}\}}{R}\right),$$

$$\tag{8}$$

*where $\bar{x}^{r,s} = \frac{1}{N}\sum_{i=1}^{N} x_i^{r,s}$,*

$$\mathcal{E}^2(\epsilon_g, \bar{\epsilon}_L, \sigma, \rho_{\max}) = (\epsilon_g^2 D_{\tau,\rho} + \rho_{\max} D_{\tau,\rho}\bar{\epsilon}_L^2 + \sigma^2) \cdot G_p' D_{\tau,\rho}\rho_{\max} N + \frac{n}{m} \cdot \frac{1}{\tau}\rho_{\max}^2\sigma^2. \tag{9}$$

*and*

$$G_p = \frac{n-m}{m(n-1)}, \qquad G_p' = G_p + \frac{1}{\tau^2}. \tag{10}$$

Compared to Theorem 1, Theorem 2 shows partial device participation deteriorates the rate by $O(\mathcal{E}(\epsilon_g, \bar{\epsilon}_L, \sigma, \rho_{\max})/\sqrt{N\tau R})$. From the expression of $\mathcal{E}$, we observe that as $\rho_{\max} \to 0$, $\mathcal{E}(\epsilon_g, \bar{\epsilon}_L, \sigma, \rho_{\max})$ vanishes, which indicates that the loss caused by device sampling can be compensated by increasing network connectivity uniformly for all clusters.

The next corollary finds the critial $\rho_{\max}$ so that $\mathcal{E}^2 = O(1)$, and the order of convergence rate of partial device participation matches that of the full participation case.

**Corollary 1.** *Under the same assumptions as Theorem 2, if the network connectivity satisfies*

$$\rho_{\max} \leq \frac{1}{4N}\min\{m, \tau - 1\}. \tag{11}$$

*then*

$$\min_{r,s} \mathbb{E}\|\nabla f(\bar{x}^{r,s})\|^2 = O\left(\frac{\sigma + \epsilon_g + \bar{\epsilon}_L}{\sqrt{N\tau R}} + \frac{\left(\tau^2\epsilon_g^2 + \tau\rho_{\max}^2 D_{\tau,\rho}\bar{\epsilon}_L^2 + \tau\left(\frac{1}{n} + \rho_{\max}^2\right)\sigma^2\right)^{\frac{1}{3}}}{(\tau R)^{\frac{2}{3}}} + \frac{1}{R}\right). \tag{12}$$

Corollary 1 reveals the tradeoff between sampling intensity and network connectivity. More connected D2D networks result in smaller $\rho_{\max}$, and thus (11) can be satisfied by a smaller $m$. This means we can sample fewer devices at the end of each round and reduce the D2S communication delay when the D2D network is more connected.

## 6 EXPERIMENTAL EVALUATION

### 6.1 EXPERIMENTAL SETTINGS

We use two common datasets in FL literature (McMahan et al., 2017; Reddi et al., 2021; Wang et al., 2020): Federated Extended MNIST (Caldas et al., 2019) (FEMNIST) and CIFAR-10 (Krizhevsky et al., 2009). The 62-class FEMNIST is built by partitioning the data in Extended MNIST (Cohen et al., 2017) based on the writer of the digit/character and has a naturally-arising device partitioning. CIFAR-10 is partitioned across all devices using a Dirichlet distribution $\mathrm{Dir}(0.1)$ as done in (Hsu et al., 2019; Yurochkin et al., 2019; Reddi et al., 2021; Wang et al., 2020). We evaluate our algorithms by training CNNs on both datasets, and the CNN models for FEMNIST and CIFAR-10 were taken from (Caldas et al., 2019) and (McMahan et al., 2017) with around $6.5$ and $1$ million parameters, respectively. For each dataset, the original testing set (without partitioning) is used to evaluate the generalization performances of the trained global model.

We consider a FL system consisting of a central server and 32 devices. The devices are evenly divided into four clusters, and each cluster has a ring topology by default, which provides a conservative estimation for the cluster connectivity and convergence speed. In our experiments, the mixing matrix of each cluster $W_k$ is set according to the Metropolis-Hastings weights (Nedić et al., 2018). According to the real-world measurements in (Yuan et al., 2020; Yang et al., 2021), we set the average time for a device to perform a local update, a round of D2D communication under ring topology, and a round of D2S communication with one device sampled per cluster to be $c_{\mathrm{cp}} = 0.01\mathrm{h}$, $c_{\mathrm{d2d}}(\Delta = 2) = 0.005\mathrm{h}$ and $c_{\mathrm{d2s}}(p = 1/8) = 0.05\mathrm{h}$, respectively, in the runtime model (2). For arbitrary device sampling ratio and D2D network topology, we consider a linear-scaling rule (Wang et al., 2019) and let $c_{\mathrm{d2d}}(\Delta) = (\Delta/2) \times 0.005\mathrm{h}$ and $c_{\mathrm{d2s}}(p) = 8p \times 0.05\mathrm{h}$.

We compare HL-SGD with local SGD in the experiments. For local SGD, devices will only communicate with the central server periodically. In all experiments, we let the local iteration period $\tau$ to be the same for both local SGD and HL-SGD to have a fair comparison. On the FEMNIST dataset, we fix the batch size as 30 and tune the learning rate $\eta$ from $\{0.005, 0.01, 0.02, 0.05, 0.08\}$ for each algorithm separately. On the CIFAR-10 dataset, we fix the batch size as 50 and tune $\eta$ from $\{0.01, 0.02, 0.05, 0.08, 0.1\}$ for each algorithm separately. We run each experiment with 3 random seeds and report the average. All experiments in this paper are conducted on a Linux server with 4 NVIDIA RTX 8000 GPUs. The algorithms are implemented by PyTorch. More details are provided in Appendix F.

### 6.2 EXPERIMENTAL RESULTS

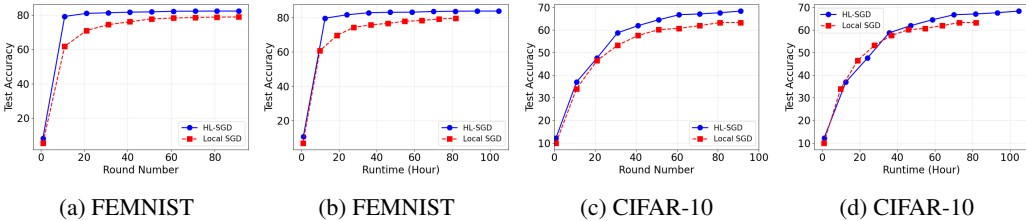

| (a) FEMNIST | (b) FEMNIST | (c) CIFAR-10 | (d) CIFAR-10 |

Figure 1: Convergence rate and runtime comparisons of HL-SGD and local SGD under ring topology when $\tau = 50$ and $p = 1$ for FEMNIST and CIFAR-10 datasets. (a) and (c) show how the accuracy changes over communication round; (b) and (d) show how the accuracy changes over runtime.

We first compare the convergence speed and runtime of HL-SGD and local SGD while fixing $\tau = 50$ and $p = 1$. We measure the best test accuracy of the global model on the server in every FL round. Figure 1 shows the convergence process. From the figure, we can observe that HL-SGD can largely accelerate the model convergence while improving model accuracy in FL. On FEMNIST, the best accuracy of HL-SGD achieved over 100 rounds is $4.78\%$ higher than that of local SGD (i.e., $83.76\%$

vs. $79.94\%$), and its runtime necessary to achieve a target test accuracy of $75\%$ is only $17.64\%$ of that of the baseline (i.e., $5.67\times$ speedup). On CIFAR-10, the best accuracy of HL-SGD achieved over 100 rounds is $9.32\%$ higher than that of local SGD (i.e., $68.71\%$ vs. $63.68\%$), and its runtime necessary to achieve a target test accuracy of $60\%$ is $15.67\%$ less than that of local SGD (i.e., $1.186\times$ speedup).

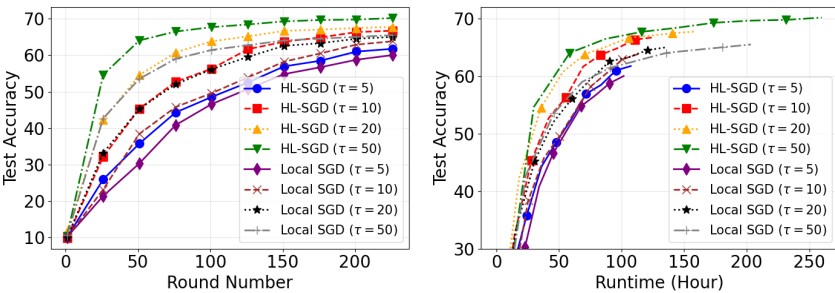

Figure 2: Convergence rate (left) and runtime (right) comparisons of HL-SGD and local SGD on CIFAR-10 under different $\tau$ and ring topology when $p = 1$.

Next, to give a more comprehensive analysis on the runtime benefits of HL-SGD, we vary $\tau$ from $\{5, 10, 20, 50\}$ and compare the performances of HL-SGD and local SGD on CIFAR-10 in Figure 2. From the figure, we can observe that HL-SGD can consistently outperform local SGD across a wide range of $\tau$. In particular, on CIFAR-10, the best accuracy of HL-SGD achieved over 100 rounds is $2.49\%$, $3.99\%$, $4.05\%$, and $7\%$ higher than that of local SGD, respectively, as $\tau$ increases from 5 to 50. At the same time, the runtime of HL-SGD needed to achieve a target test accuracy of $60\%$ is $9.66\%$, $19.76\%$, $33.46\%$, and $45.88\%$ less than that of local SGD, respectively.

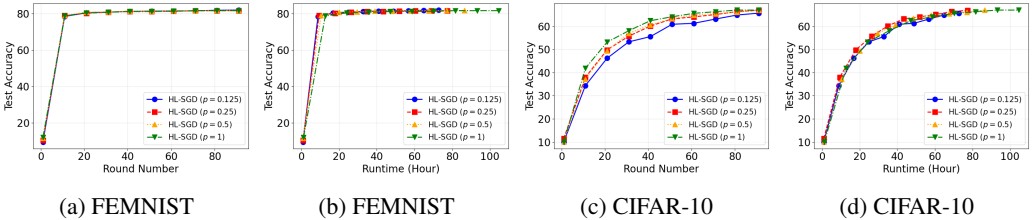

|  (a) FEMNIST | (b) FEMNIST | (c) CIFAR-10 | (d) CIFAR-10 |

Figure 3: Effect of sampling ratio $p$ on the convergence rate and runtime of HL-SGD under ring topology when $\tau = 50$ for FEMNIST and CIFAR-10 datasets. (a) and (c) show how the accuracy changes over communication round in HL-SGD; (b) and (d) show how the accuracy changes over runtime.

Finally, we investigate how the sampling ratio $p$ affects the performance of HL-SGD. We select $p$ from $\{0.125, 0.25, 0.5, 1\}$, corresponding to sampling $\{1, 2, 4, 8\}$ devices from each cluster to upload models to the server. Figure 3 depicts the best value of test accuracy achieved over all prior rounds. As can be observed from the figures, sampling one device per cluster only results in slightly lower model accuracy, e.g., neligible and $1.92\%$ drop compared to full participation on FEMNIST and CIFAR-10, respectively. This matches the theoretical result in Corollary 1 that device sampling does not affect the order of convergence rate under certain conditions. However, decreasing $p$ can lead to faster training speed due to its shorter D2S communication delay as observed in Figures 3b and 3d. In practice, the optimal value of $p$ needs to be tuned to strike a good balance between model accuracy and runtime.

## 7 CONCLUSION

In this paper, we have proposed a new optimization algorithm called HL-SGD for FL with heterogeneous communications. Our algorithm leverages the D2D communication capabilities among edge device to accelerate the model convergence while improving model accuracy in FL. We have provided the theoretical convergence analysis of HL-SGD and conducted experiments to demonstrate the benefits of HL-SGD. In the future, we plan to extend HL-SGD to handle straggler issues under device heterogeneity and provide rigorous privacy protection for HL-SGD.

ACKNOWLEDGMENTS

The work of Y. Guo was supported in part by NSF under the Grant CNS-2106761. The work of Y. Sun was partially supported by the Office of Naval Research under the Grant N00014-21-1-2673. The work of R. Hu and Y. Gong was supported in part by NSF under the Grants CNS-2047761 and CNS-2106761.

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

## A  PRELIMINARIES

**Intra-cluster dynamics.** To facilitate the analysis, we introduce matrices $X_k \in \mathbb{R}^{n \times d}$ and $G_k \in \mathbb{R}^{n \times d}$ constructed by stacking respectively $x_i$ and $g_i$ for $i \in \mathcal{V}_k$ row-wise. Similarly, we define the pseudo-gradient $\nabla F_k(X_k) \in \mathbb{R}^{n \times d}$ associated to cluster $k$ by stacking $\nabla f_i(x_i)$ for $i \in \mathcal{V}_k$ row-wise. In addition, define the following intra-cluster averages for each cluster $k$:

$$\bar{x}_k \triangleq \frac{1}{n} \sum_{i \in \mathcal{V}_k} x_i \quad \text{and} \quad \bar{g}_k \triangleq \frac{1}{n} \sum_{i \in \mathcal{V}_k} g_i. \tag{13}$$

The update within each cluster then can be written compactly in matrix form as

$$X_k^{r,s+1} = W_k(X_k^{r,s} - \eta G_k^{r,s}), \qquad \forall k = 1, \ldots, K. \tag{14}$$

Since each $W_k$ is bi-stochastic, we obtain the following update of the intra-cluster average

$$\bar{x}_k^{r,s+1} = \bar{x}_k^{r,s} - \eta \cdot \bar{g}_k^{r,s}. \tag{15}$$

We proceed to derive the update of the intra-cluster consensus error. Define the averaging matrix

$$J = \frac{1}{n} \mathbf{1} \cdot \mathbf{1}^\top \quad \text{with} \quad \mathbf{1} = [\underbrace{1, \ldots, 1}_{n}]. \tag{16}$$

Multiplying both sides of (14) from the left by $(I - J)$ leads to the following update of the consensus error:

$$\underbrace{(I - J)X_k^{r,s+1}}_{X_{k,\perp}^{r,s+1}} = (I - J)W_k(X_k^{r,s} - \eta G_k^{r,s})$$

$$= (W_k - J)(X_k^{r,s} - \eta G_k^{r,s})$$
$$= (W_k - J)(X_{k,\perp}^{r,s} - \eta G_k^{r,s}). \tag{17}$$

**Global average dynamics.** Define the global average among all $x_i$'s as

$$\bar{x} \triangleq \frac{1}{N} \sum_{i=1}^{N} x_i. \tag{18}$$

Then accordingly to (15) we have the following update of $\bar{x}$ for all $s = 0, \ldots, \tau - 1$:

$$
\begin{aligned}
\bar{x}^{r,s+1} &= \frac{1}{N} \sum_{k=1}^{K} n \bar{x}_k^{r,s+1} = \frac{1}{N} \sum_{k=1}^{K} n \left( \bar{x}_k^{r,s} - \eta \bar{g}_k^{r,s} \right) \\
&= \frac{1}{N} \sum_{i=1}^{N} (x_i^{r,s} - \eta g_i^{r,s}) = \bar{x}^{r,s} - \eta \frac{1}{N} \sum_{i=1}^{N} g_i^{r,s}.
\end{aligned}
\tag{19}
$$

**Filtration.** Let $G = [G_1; \ldots; G_K] \in \mathbb{R}^{N \times d}$ be the matrix constructed by stacking all the stochastic gradients. We introduce the following filtration

$$
\begin{aligned}
\mathcal{F}_{r,s} &= \sigma \left( G^{0,0}, \ldots, G^{0,\tau-1}, S_0, G^{1,0}, \ldots, G^{1,\tau-1}, \ldots, S_{r-1}, G^{r,0}, \ldots, G^{r,s} \right) \\
\mathcal{F}_r &= \sigma \left( G^{0,0}, \ldots, G^{0,\tau-1}, S_0, G^{1,0}, \ldots, G^{1,\tau-1}, \ldots, S_{r-1} \right).
\end{aligned}
\tag{20}
$$

Therefore we have $x_i^{r,0} = x^r \in \mathcal{F}_r$ for $r \geq 1$, and $x_i^{r,s} \in \mathcal{F}_{r,s-1}$ for $1 \leq s \leq \tau$. For simplicity the conditional expectation $\mathbb{E}(\cdot | \mathcal{F}_{r,s})$ is denoted as $\mathbb{E}_{r,s}$, and we define the noise in the stochastic gradient as

$$
\xi_i^{r,s} \triangleq g_i^{r,s} - \nabla f_i(x_i^{r,s}).
\tag{21}
$$

Since at the end of round $r$ all nodes are picked with equal probability, the sampling procedure preserves average in expectation:

$$
\begin{aligned}
\mathbb{E}_{r,\tau-2} x^{r+1} &= \mathbb{E}(\mathbb{E}(x^{r+1} | \mathcal{F}_{r,\tau-1}) | \mathcal{F}_{r,\tau-2}) \\
&= \mathbb{E} \left( \mathbb{E} \left( \frac{1}{K} \sum_{k=1}^{K} \frac{1}{m} \sum_{i \in S_k^r} x_i^{r,\tau} | \mathcal{F}_{r,\tau-1} \right) | \mathcal{F}_{r,\tau-2} \right) \\
&= \mathbb{E} \left( \mathbb{E} \left( \frac{1}{K} \sum_{k=1}^{K} \frac{1}{m} \sum_{i \in \mathcal{V}_k} \mathbb{I}(i \in S_k^r) x_i^{r,\tau} | \mathcal{F}_{r,\tau-1} \right) | \mathcal{F}_{r,\tau-2} \right) \\
&= \mathbb{E}_{r,\tau-2}(\bar{x}^{r,\tau})
\end{aligned}
\tag{22}
$$

where the last equality holds since $\mathbb{P}\left( i \in S_k^r | i \in \mathcal{V}_k \right) = \frac{m}{n}$.

# B CONVERGENCE ANALYSIS

To prove the convergence we first establish in Sec. B.1 that the objective value $\mathbb{E}f(x^r)$ is descending at each round $r$, up to some consensus error terms. Subsequently, bounds on the error terms are provided in Sec. B.2-B.4. Based on these results, the proof of convergence of Algorithm 1 with full and partial device participation are given in Sec. B.5 and B.6, respectively. The proofs of the main propositions are given in Sec. C and that of the supporting lemmas are deferred to Sec. D.

## B.1 OBJECTIVE DESCENT

**Lemma 1.** *Let $\{x_i^{r,s}\}$ be the sequence generated by Algorithm 1 under Assumptions 1-6. If $\eta > 0$, then the following inequality holds for all $r \in \mathbb{N}_+$:*

$$
\begin{aligned}
\mathbb{E}&f(x^{r+1})\\
\leq{}& \mathbb{E}f(x^r) - \frac{\eta}{4}\sum_{s=0}^{\tau-1}\mathbb{E}\|\nabla f(\bar{x}^{r,s})\|^2 - \frac{\eta}{4}\sum_{s=0}^{\tau-1}\mathbb{E}\Big\|\frac{1}{N}\sum_{i=1}^{N}\nabla f_i(x_i^{r,s})\Big\|^2\\
&- \frac{\eta}{4}\sum_{s=0}^{\tau-1}\mathbb{E}\Big\|\frac{1}{K}\sum_{k=1}^{K}\nabla\bar{f}_k(\bar{x}_k^{r,s})\Big\|^2\\
&+ \frac{\eta}{4}\sum_{s=0}^{\tau-1}\mathbb{E}\Big\|\nabla f(\bar{x}^{r,s}) - \frac{1}{N}\sum_{i=1}^{N}\nabla f_i(x_i^{r,s})\Big\|^2 + \frac{\eta}{4}\sum_{s=0}^{\tau-1}\mathbb{E}\Big\|\nabla f(\bar{x}^{r,s}) - \frac{1}{K}\sum_{k=1}^{K}\nabla\bar{f}_k(\bar{x}_k^{r,s})\Big\|^2 \quad (23)\\
&+ \frac{\eta}{4}\sum_{s=0}^{\tau-1}\mathbb{E}\Big\|\frac{1}{N}\sum_{i=1}^{N}\nabla f_i(x_i^{r,s}) - \frac{1}{K}\sum_{k=1}^{K}\nabla\bar{f}_k(\bar{x}_k^{r,s})\Big\|^2\\
&+ \sum_{s=0}^{\tau-2}\mathbb{E}\left(\frac{L}{2}\|\bar{x}^{r,s+1} - \bar{x}^{r,s}\|^2\right) + \mathbb{E}\left(\frac{L}{2}\|x^{r+1} - \bar{x}^{r,\tau-1}\|^2\right).
\end{aligned}
$$

*Proof.* The proof is a standard application of the descent lemma and the sampling rule applied at iteration $\tau$ to obtain $x^{r+1}$. See Appendix D.1. □

Lemma 1 shows the objective value $f(x^r)$ is descending in expectation up to the following error terms:

$$
\begin{aligned}
T_1 &= \mathbb{E}\Big\|\nabla f(\bar{x}^{r,s}) - \frac{1}{N}\sum_{i=1}^{N}\nabla f_i(x_i^{r,s})\Big\|^2, \quad T_2 = \mathbb{E}\Big\|\nabla f(\bar{x}^{r,s}) - \frac{1}{K}\sum_{k=1}^{K}\nabla\bar{f}_k(\bar{x}_k^{r,s})\Big\|^2\\
T_3 &= \mathbb{E}\Big\|\frac{1}{K}\sum_{k=1}^{K}\nabla\bar{f}_k(\bar{x}_k^{r,s}) - \frac{1}{N}\sum_{i=1}^{N}\nabla f_i(x_i^{r,s})\Big\|^2 \quad\quad\quad (24)\\
T_4 &= \mathbb{E}\|\bar{x}^{r,s+1} - \bar{x}^{r,s}\|^2, \quad T_5 = \mathbb{E}\|x^{r+1} - \bar{x}^{r,\tau-1}\|^2.
\end{aligned}
$$

In the sequel, we will show these quantities can be bounded by the optimality gap measured in terms of the gradient norms $\|\nabla f(\bar{x}^{r,s})\|^2$, $\|(1/K)\sum_{k=1}^{K}\nabla\bar{f}_k(\bar{x}_k^{r,s})\|^2$, and $\|(1/N)\sum_{i=1}^{N}\nabla f_i(x_i^{r,s})\|^2$.

## B.2 BOUNDING $T_1$, $T_2$ AND $T_3$.

Define

$$
\rho_{\max} = \max_{k=1,\ldots,K}\rho_k. \quad\quad\quad (25)
$$

Therefore it holds $0 \leq \rho_{\max} \leq 1$ by Assumption 3.

Since each $f_i$ is $L$-smooth by Assumption 1, we have $\bar{f}_k$ and $f$ are also $L$-smooth. Using this fact and the convexity of $\|\cdot\|^2$ we can bound $T_1$, $T_2$ and $T_3$ as

$$
\begin{aligned}
T_1 &= \mathbb{E}\left\|\nabla f(\bar{x}^{r,s}) \pm \frac{1}{K}\sum_{k=1}^{K}\nabla\bar{f}_k(\bar{x}_k^{r,s}) - \frac{1}{N}\sum_{i=1}^{N}\nabla f_i(x_i^{r,s})\right\|^2 \\
&\leq 2\frac{1}{K}\sum_{k=1}^{K}L^2\mathbb{E}\|\bar{x}^{r,s}-\bar{x}_k^{r,s}\|^2 + 2\sum_{k=1}^{K}\frac{1}{N}\sum_{i\in\mathcal{V}_k}L^2\mathbb{E}\|\bar{x}_k^{r,s}-x_i^{r,s}\|^2,
\end{aligned}
$$
$$
T_2 = \mathbb{E}\left\|\nabla f(\bar{x}^{r,s}) - \frac{1}{K}\sum_{k=1}^{K}\nabla\bar{f}_k(\bar{x}_k^{r,s})\right\|^2 \leq \frac{1}{K}\sum_{k=1}^{K}L^2\mathbb{E}\left\|\bar{x}^{r,s}-\bar{x}_k^{r,s}\right\|^2,
$$
$$
T_3 = \mathbb{E}\left\|\frac{1}{K}\sum_{k=1}^{K}\nabla\bar{f}_k(\bar{x}_k^{r,s}) - \frac{1}{N}\sum_{i=1}^{N}\nabla f_i(x_i^{r,s})\right\|^2 \leq \sum_{k=1}^{K}\frac{1}{N}\sum_{i\in\mathcal{V}_k}L^2\mathbb{E}\|\bar{x}_k^{r,s}-x_i^{r,s}\|^2.
$$

(26)

Clearly, in order to bound $T_{1,2,3}$ we first need to bound the inter-cluster consensus error $\|\bar{x}^{r,s}-\bar{x}_k^{r,s}\|$ and the intra-cluster consensus error $\|\bar{x}_k^{r,s}-x_i^{r,s}\|$.

**Lemma 2** (Inter-Cluster Consensus Error Bound). *Let $\{x_i^{r,s}\}$ be the sequence generated by Algorithm 1 under Assumptions 1, 2, 3, and 5. If the learning rate $\eta > 0$ satisfies*

$$
\eta^2 \leq \frac{1}{24\tau(4\tau-1)L^2},
$$

(27)

*then for all $s = 0, \ldots, \tau - 1$ it holds*

$$
\begin{aligned}
\frac{1}{K}\sum_{k=1}^{K}\mathbb{E}\|\bar{x}^{r,s+1}-\bar{x}_k^{r,s+1}\|^2 &\leq C_\tau\frac{1}{K}\sum_{k=1}^{K}\mathbb{E}\|\bar{x}^{r,s}-\bar{x}_k^{r,s}\|^2 \\
&+ 12\tau\eta^2 L^2\left(\frac{1}{N}\sum_{K=1}^{K}\mathbb{E}\left\|X_{k,\perp}^{r,s}\right\|^2\right) + 12\tau\eta^2\left(\alpha^2\mathbb{E}\|\nabla f(\bar{x}^{r,s})\|^2 + \epsilon_g^2\right) + \eta^2\frac{K-1}{N}\sigma^2
\end{aligned}
$$

(28)

*where*

$$
C_\tau \triangleq 1 + \frac{3}{2}\cdot\frac{1}{4\tau-1}.
$$

(29)

*Proof.* See Appendix D.2. $\qquad\square$

**Lemma 3** (Intra-Cluster Consensus Error Bound). *Let $\{x_i^{r,s}\}$ be the sequence generated by Algorithm 1 under Assumptions 1-5. If $\eta > 0$, then for all $s = 0, \ldots, \tau - 1$ it holds*

$$
\begin{aligned}
\frac{1}{N}\sum_{k=1}^{K}\mathbb{E}\|X_{k,\perp}^{r,s+1}\|^2 &\leq \left(\max_{k\in[K]}\rho_k^2(1+\zeta_k^{-1}) + \eta^2\rho_L\cdot 4L^2\right)\frac{1}{N}\sum_{k=1}^{K}\mathbb{E}\|X_{k,\perp}^{r,s}\|^2 \\
&+ 4\eta^2\rho_L L^2\frac{1}{K}\sum_{k=1}^{K}\mathbb{E}\|\bar{x}_k^{r,s}-\bar{x}^{r,s}\|^2 + 4\eta^2\rho_L(\alpha^2\mathbb{E}\|\nabla f(\bar{x}^{r,s})\|^2 + \epsilon_g^2) + 4\eta^2\rho_L\bar{\epsilon}_L^2 + \eta^2\sigma^2\rho_{\max}^2,
\end{aligned}
$$

(30)

*where $\rho_{\max}$ is defined in (25) and*

$$
\rho_L \triangleq \max_{k=1,\ldots,K}\left\{\rho_k^2(1+\zeta_k)\right\}, \quad \bar{\epsilon}_L^2 \triangleq \frac{1}{K}\sum_{k=1}^{K}\epsilon_k^2
$$

(31)

*with $\zeta_k > 0$ being a free parameter to be chosen properly for all $k = 1, \ldots, K$.*

*Proof.* See Appendix D.3. $\qquad\square$

Combining Lemma 2 and 3 we can obtain the following bound on the sum of intra- and inter-consensus errors using gradient $\|\nabla f(\bar{x}^{r,s})\|^2$.

**Proposition 1.** *Let $\{x_i^{r,s}\}$ be the sequence generated by Algorithm 1 under Assumptions 1-5. If the learning rate $\eta > 0$ satisfies*

$$\eta^2 \leq \frac{1}{24\tau(4\tau - 1)L^2}, \tag{32}$$

*then for all $s = 0, \ldots, \tau - 1$ it holds*

$$\frac{1}{K}\sum_{k=1}^{K}\mathbb{E}\|\bar{x}^{r,s+1} - \bar{x}_k^{r,s+1}\|^2 + \sum_{k=1}^{K}\frac{1}{N}\|X_{k,\perp}^{r,s+1}\|^2$$

$$\leq \sum_{\ell=0}^{s} C_1 \eta^2 \left(\tau + \rho_{\max}^2 D_{\tau,\rho}\right)\left(\alpha^2 \mathbb{E}\|\nabla f(\bar{x}^{r,\ell})\|^2 + \epsilon_g^2\right) + C_1 \eta^2 \tau \rho_{\max}^2 D_{\tau,\rho}\bar{\epsilon}_L^2 \tag{33}$$

$$+ C_1 \tau \eta^2 \rho_{\max}^2 \sigma^2 + C_1 (\tau + D_{\tau,\rho}^2 \tau^{-1} \rho_{\max}^2)\eta^2 \frac{1}{n}\sigma^2$$

*where*

$$D_{\tau,\rho} \triangleq \min\left\{\tau, \frac{1}{1 - \rho_{\max}}\right\} \tag{34}$$

*and $C_1 > 0$ is some universal constant.*

*Proof.* See Appendix C.1. $\qquad\square$

Notice that according to (26) the gradient difference terms in Lemma 1 can be bounded as

$$\frac{\eta}{4}\mathbb{E}\left\|\nabla f(\bar{x}^{r,s}) - \frac{1}{N}\sum_{i=1}^{N}\nabla f_i(x_i^{r,s})\right\|^2 + \frac{\eta}{4}\mathbb{E}\left\|\nabla f(\bar{x}^{r,s}) - \frac{1}{K}\sum_{k=1}^{K}\nabla \bar{f}_k(\bar{x}_k^{r,s})\right\|^2$$

$$+ \frac{\eta}{4}\mathbb{E}\left\|\frac{1}{N}\sum_{i=1}^{N}\nabla f_i(x_i^{r,s}) - \frac{1}{K}\sum_{k=1}^{K}\nabla \bar{f}_k(\bar{x}_k^{r,s})\right\|^2$$

$$\leq \frac{\eta}{4}\left(\frac{2}{K}\sum_{k=1}^{K}L^2\mathbb{E}\|\bar{x}^{r,s} - \bar{x}_k^{r,s}\|^2 + \sum_{k=1}^{K}\frac{2}{N}\sum_{i\in\mathcal{V}_k}L^2\mathbb{E}\|\bar{x}_k^{r,s} - x_i^{r,s}\|^2\right) \tag{35}$$

$$+ \frac{\eta}{4}\left(\frac{1}{K}\sum_{k=1}^{K}L^2\mathbb{E}\left\|\bar{x}^{r,s} - \bar{x}_k^{r,s}\right\|^2 + \sum_{k=1}^{K}\frac{1}{N}\sum_{i\in\mathcal{V}_k}L^2\mathbb{E}\|\bar{x}_k^{r,s} - x_i^{r,s}\|^2\right)$$

$$\leq \eta L^2\left(\frac{1}{K}\sum_{k=1}^{K}\mathbb{E}\|\bar{x}^{r,s} - \bar{x}_k^{r,s}\|^2 + \sum_{k=1}^{K}\frac{1}{N}\|X_{k,\perp}^{r,s}\|^2\right)$$

for all $s = 1, \ldots, \tau$. Therefore Proposition 1 immediately leads to the following result.

**Corollary 2.** *Under the same setting as Proposition 1, it holds*

$$\sum_{s=0}^{\tau}\frac{\eta}{4}\mathbb{E}\left\|\nabla f(\bar{x}^{r,s}) - \frac{1}{N}\sum_{i=1}^{N}\nabla f_i(x_i^{r,s})\right\|^2 + \sum_{s=0}^{\tau}\frac{\eta}{4}\mathbb{E}\left\|\nabla f(\bar{x}^{r,s}) - \frac{1}{K}\sum_{k=1}^{K}\nabla \bar{f}_k(\bar{x}_k^{r,s})\right\|^2$$

$$+ \sum_{s=0}^{\tau}\frac{\eta}{4}\sum_{s=0}^{\tau-1}\mathbb{E}\left\|\frac{1}{N}\sum_{i=1}^{N}\nabla f_i(x_i^{r,s}) - \frac{1}{K}\sum_{k=1}^{K}\nabla \bar{f}_k(\bar{x}_k^{r,s})\right\|^2 \tag{36}$$

$$\leq \sum_{s=0}^{\tau-1}C_1 L^2 \eta^3 \left(\tau^2 + \tau \rho_{\max}^2 D_{\tau,\rho}\right)\left(\alpha^2 \mathbb{E}\|\nabla f(\bar{x}^{r,s})\|^2 + \epsilon_g^2\right) + C_1 L^2 \eta^3 \tau^2 \rho_{\max}^2 D_{\tau,\rho}\bar{\epsilon}_L^2$$

$$+ C_1 \tau^2 L^2 \eta^3 \rho_{\max}^2 \sigma^2 + C_1 L^2 (\tau^2 + D_{\tau,\rho}^2 \rho_{\max}^2)\eta^3 \frac{\sigma^2}{n}.$$

We conclude this section by providing a separate bound on the consensus error $\frac{1}{N} \sum_{k=1}^{K} \mathbb{E}_r \|X_{k,\perp}^{r,\tau-1}\|^2$ that will be useful in bounding $T_5$.

**Proposition 2.** *Under the same setting as Proposition 1, if $\rho_{\max} \leq 1 - \frac{1}{\tau}$, then we have*

$$\frac{1}{N} \mathbb{E} \sum_{k=1}^{K} \|X_{k,\perp}^{r,\tau-1}\|^2 \leq 2C_2 \sum_{s=0}^{\tau-2} D_{\tau,\rho}^2 \rho_{\max}^2 \eta^2 (\alpha^2 \|\nabla f(\bar{x}^{r,s})\|^2 + \epsilon_g^2)$$

$$+ C_2 D_{\tau,\rho}^2 \tau \rho_{\max}^2 \eta^2 \bar{\epsilon}_L^2 + C_2 \left( \frac{1}{n} \frac{D_{\tau,\rho}}{\tau} + 1 \right) \rho_{\max}^2 \tau D_{\tau,\rho} \eta^2 \sigma^2. \quad (37)$$

*for some universal constant $C_2 > 0$.*

*Proof.* See Appendix C.2. □

Proposition 2 shows that the average intra-cluster consensus error $\frac{1}{N} \sum_{k=1}^{K} \|X_{k,\perp}^{r,\tau-1}\|^2$ decreases as the network connectivity improves, and vanishes if $\rho_{\max}$ goes to zero.

### B.3 BOUNDING $T_4$

**Proposition 3.** *Under the same setting as Lemma 1, we have*

$$\mathbb{E}\|\bar{x}^{r,s+1} - \bar{x}^{r,s}\|^2 = \eta^2 \mathbb{E} \left\| \frac{1}{N} \sum_{i=1}^{N} \nabla f_i(x_i^{r,s}) \right\|^2 + \eta^2 \frac{\sigma^2}{N} \quad (38)$$

*for $s = 0, \ldots, \tau - 1$ and $r \in \mathbb{N}_+$.*

*Proof.* Recall the algorithmic update at iteration $s$ for all $s = 0, \ldots, \tau - 1$:

$$\begin{aligned} X_k^{r,s+1} &= W_k X_k^{r,s} - \eta W_k G_k^{r,s} \\ \bar{x}_k^{r,s+1} &= \bar{x}_k^{r,s} - \eta \bar{g}_k^{r,s}. \end{aligned} \quad (39)$$

Therefore, it holds under Assumption 2 that

$$\mathbb{E}\|\bar{x}^{r,s+1} - \bar{x}^{r,s}\|^2$$

$$= \mathbb{E} \left\| \frac{\eta}{N} \sum_{i=1}^{N} (g_i^{r,s} \pm \nabla f_i(x_i^{r,s})) \right\|^2 = \mathbb{E} \left\| \frac{\eta}{N} \sum_{i=1}^{N} \nabla f_i(x_i^{r,s}) \right\|^2 + \eta^2 \frac{\sigma^2}{N}. \quad (40)$$

□

### B.4 BOUNDING $T_5$

We provide the bound on $T_5$ separately for the full device participation and partial participation cases.

**Full participation.**

When the sampling probability $p = 1$, we have

$$x^{r+1} = \frac{1}{N} \sum_{i=1}^{N} x_i^{r,\tau} = \bar{x}^{r,\tau}.$$

In this case, it follows from Proposition 3 that

$$\mathbb{E}\|x^{r+1} - \bar{x}^{r,\tau-1}\|^2 = \eta^2 \mathbb{E} \left\| \frac{1}{N} \sum_{i=1}^{N} \nabla f_i(x_i^{r,\tau-1}) \right\|^2 + \eta^2 \frac{\sigma^2}{N}. \quad (41)$$

**Partial participation.**

We proceed to bound $T_5$ for

$$1 \leq m \leq n - 1. \tag{42}$$

Define $p = m/n$. Recall the algorithmic update at iteration $\tau - 1$:

$$X_k^{r,\tau} = W_k X_k^{r,\tau-1} - \eta W_k G_k^{r,\tau-1} \tag{43}$$

and

$$x^{r+1} = \frac{1}{K} \sum_{k=1}^{K} \frac{1}{m} \sum_{i \in S_k^r} x_i^{r,\tau} = \frac{1}{Np} \sum_{k=1}^{K} \sum_{i \in S_k^r} x_i^{r,\tau}. \tag{44}$$

Therefore, with $(W_k)_{i,j}$ being the $ij$-th element of matrix $W_k$ we have under Assumption 2:

$$
\begin{aligned}
&\mathbb{E}\|x^{r+1} - \bar{x}^{r,\tau-1}\|^2 \\
&= \mathbb{E}\left( \left\| \frac{1}{Np} \sum_{k=1}^{K} \sum_{i \in S_k^r} x_i^{r,\tau} - \bar{x}^{r,\tau-1} \right\|^2 \right) \\
&= \mathbb{E}\left\| \frac{1}{Np} \sum_{k=1}^{K} \sum_{i \in S_k^r} \left( \sum_{j \in \mathcal{V}_k} (W_k)_{i,j} (x_j^{r,\tau-1} - \eta g_j^{r,\tau-1}) \right) - \bar{x}^{r,\tau-1} \right\|^2 \\
&= \underbrace{\mathbb{E}\left\| \frac{1}{Np} \sum_{k=1}^{K} \sum_{i \in \mathcal{V}_k} \mathbb{I}(i \in S_k^r) \left( \sum_{j \in \mathcal{V}_k} (W_k)_{i,j} (x_j^{r,\tau-1} - \eta \nabla f_j(x_j^{r,\tau-1})) \right) - \bar{x}^{r,\tau-1} \right\|^2}_{A_{2,1}} \\
&\quad + \eta^2 \underbrace{\mathbb{E}\left\| \frac{1}{Np} \sum_{k=1}^{K} \sum_{i \in \mathcal{V}_k} \mathbb{I}(i \in S_k^r) \left( \sum_{j \in \mathcal{V}_k} (W_k)_{i,j} (\nabla f_j(x_j^{r,\tau-1}) - g_j^{r,\tau-1}) \right) \right\|^2}_{A_{2,2}}.
\end{aligned}
\tag{45}
$$

**Proposition 4.** *Let $\{x_i^{r,s}\}$ be the sequence generated by Algorithm 1 under Assumptions 1-6. If the learning rate $\eta > 0$ satisfies*

$$\eta^2 \leq \frac{1}{24\tau(4\tau - 1)L^2}, \tag{46}$$

*then we have the following bounds on $A_{2,1}$:*

$$A_{2,1} \leq 2\eta^2 \mathbb{E}\left\| \frac{1}{K} \sum_{k=1}^{K} \nabla \bar{f}_k(\bar{x}_k^{r,\tau-1}) \right\|^2 + 8\left( G_p + \frac{1}{\tau^2} \right) \left( \frac{1}{N} \sum_{k=1}^{K} \mathbb{E}\|X_{k,\perp}^{r,\tau-1}\|^2 \right); \tag{47}$$

*where*

$$G_p \triangleq \frac{n - m}{m(n - 1)}. \tag{48}$$

*Proof.* See Appendix C.3.

$\square$

**Proposition 5.** *Under the same setting as Proposition 4, $A_{2,2}$ can be bounded as*

$$A_{2,2} \leq \frac{\sigma^2}{N} \left( 2 + \frac{n}{m} \cdot \rho_{\max}^2 \right). \tag{49}$$

*Proof.* See Appendix C.4

$\square$

### B.5 PROOF OF THEOREM 1 (FULL PARTICIPATION)

We first prove the descent of the objective value under suitable choice of $\eta$.

**Proposition 6.** *If the learning rate satisfies*

$$\eta \leq \frac{1}{4C_1\alpha} \cdot \frac{1}{\tau L}, \tag{50}$$

*then we have*

$$\mathbb{E}f(x^{r+1}) \leq \mathbb{E}f(x^r) - \frac{\eta}{8}\sum_{s=0}^{\tau-1}\mathbb{E}\|\nabla f(\bar{x}^{r,s})\|^2 + \mathcal{R}_{\text{full}}(\eta), \tag{51}$$

*where*

$$\mathcal{R}_{\text{full}}(\eta) = C_1 L^2 \eta^3 \tau^2 \left(\tau + \rho_{\max}^2 D_{\tau,\rho}\right)\epsilon_g^2 + 2C_1 L^2 \eta^3 \left(\tau^2 \rho_{\max}^2 D_{\tau,\rho}\bar{\epsilon}_L^2 + \tau^2 \sigma^2 \left(\frac{1}{n} + \rho_{\max}^2\right)\right)$$
$$+ \eta^2 L \tau \frac{\sigma^2}{N}. \tag{52}$$

$C_1 > 0$ *is some universal constant.*

*Proof.* See Appendix C.5.  $\square$

To attain the expression of the convergence rate, we sum (51) over $r = 0, \ldots, R$:

$$\min_{r\in[R]}\min_{s=0,\ldots,\tau-1}\mathbb{E}\|\nabla f(\bar{x}^{r,s})\|^2$$
$$\leq \frac{8(f(x^0) - f(x^\star))}{\eta\tau(R+1)} + \frac{8\mathcal{R}_{\text{full}}(\eta)}{\eta\tau}$$
$$= \underbrace{\frac{8(f(x^0) - f(x^\star))}{\eta\tau(R+1)} + 16\eta L\frac{\sigma^2}{N}}_{\text{centralized SGD}} \tag{53}$$
$$+ \underbrace{16C_1 L^2 \tau^2 \eta^2 \epsilon_g^2 + 16C_1 L^2 \eta^2 \left(\tau\rho_{\max}^2 D_{\tau,\rho}\bar{\epsilon}_L^2 + \tau\sigma^2\left(\frac{1}{n} + \rho_{\max}^2\right)\right)}_{\text{network effect}}.$$

The first two terms of (53) corresponds to the impact of stochastic noise and is of the same order as the centralized SGD algorithm. The last term is of order $\eta^2$ and corresponds to the deterioration of convergence rate due to the fact that we are not computing the average gradients of all devices at each iteration.

Denote

$$r_0 = 8(f(x^0) - f(x^\star)), \ r_1 = 16L\left(\frac{\sigma^2}{N}\right),$$
$$r_2 = 16C_1 L^2 \tau^2 \epsilon_g^2 + 16C_1 L^2\left(\tau\rho_{\max}^2 D_{\tau,\rho}\bar{\epsilon}_L^2 + \tau\sigma^2\left(\frac{1}{n} + \rho_{\max}^2\right)\right). \tag{54}$$

The rest of the proof follows the same argument as (**?**, Appendix B.5) and thus we omit the details.

### B.6 PROOF OF THEOREM 2 AND COROLLARY 1 (PARTIAL PARTICIPATION)

**Proposition 7.** *Let $\{x_i^{r,s}\}$ be the sequence generated by Algorithm 1 under Assumption 1-5. If the learning rate $\eta$ and the network connectivity satisfies*

$$\eta \leq \frac{1}{C_3\alpha\tau L} \cdot \min\left\{1, \frac{1}{\alpha G_p D_{\tau,\rho}\ \rho_{\max}}\right\} \quad and \quad \rho_{\max} \leq 1 - \frac{1}{\tau}, \tag{55}$$

*then*

$$\mathbb{E}f(x^{r+1}) \le \mathbb{E}f(x^r) - \frac{\eta}{8}\sum_{s=0}^{\tau-1}\mathbb{E}\|\nabla f(\bar{x}^{r,s})\|^2 + \mathcal{R}_{\text{part}}^{(1)}(\eta) + \mathcal{R}_{\text{part}}^{(2)}(\eta) \tag{56}$$

*with*

$$\mathcal{R}_{\text{part}}^{(1)}(\eta) = C_1 L^2 \eta^3 \tau^2 \left(\tau + \rho_{\max}^2 D_{\tau,\rho}\right)\epsilon_g^2 + 2C_1 L^2 \eta^3 \left(\tau^2 \rho_{\max}^2 D_{\tau,\rho}\bar{\epsilon}_L^2 + \tau^2 \sigma^2 \left(\frac{1}{n} + \rho_{\max}^2\right)\right)$$

$$+ L\tau\eta^2 \frac{\sigma^2}{N}, \tag{57}$$

$$\mathcal{R}_{\text{part}}^{(2)}(\eta) = \left(8 C_2 G_p' L D_{\tau,\rho}^2 \tau \rho_{\max}^2\right)\eta^2 \epsilon_g^2$$

$$+ 4 C_2 L G_p' \eta^2 \left(D_{\tau,\rho}^2 \tau \rho_{\max}^2 \bar{\epsilon}_L^2 + \left(\frac{1}{n}\frac{D_{\tau,\rho}}{\tau} + 1\right)\rho_{\max}^2 \tau D_{\tau,\rho}\sigma^2\right) + \frac{L}{2}\left(\frac{n}{m}\rho_{\max}^2\right)\eta^2 \frac{\sigma^2}{N}.$$

$C_1$, $C_3 > 0$ *are some universal constants, and*

$$G_p' = G_p + \frac{1}{\tau^2}, \quad \text{with} \quad G_p = \frac{n-m}{m(n-1)}. \tag{58}$$

*Proof.* See Appendix C.6 □

Comparing (56) to (51) we can see that $\mathcal{R}_{\text{part}}^{(1)}(\eta)$ is of the same order as $\mathcal{R}_{\text{full}}^{(1)}(\eta)$, while $\mathcal{R}_{\text{part}}^{(2)}(\eta)$ is an extra loss term introduced by sampling.

Following the same steps as the proof of Theorem 1 gives

$$\min_{r\in[R]}\min_{s=0,\dots,\tau-1}\mathbb{E}\|\nabla f(\bar{x}^{r,s})\|^2$$

$$= O\left(\frac{\sigma + \mathcal{E}(\epsilon_g,\bar{\epsilon}_L,\sigma,\rho_{\max})}{\sqrt{N\tau R}} + \frac{\left(\tau^2\epsilon_g^2 + \tau\rho_{\max}^2 D_{\tau,\rho}\bar{\epsilon}_L^2 + \tau\left(\frac{1}{n}+\rho_{\max}^2\right)\sigma^2\right)^{\frac{1}{3}}}{(\tau R)^{\frac{2}{3}}}\right. \tag{59}$$

$$\left. + \frac{1}{R}\max\{1, G_p D_{\tau,\rho}\rho_{\max}\}\right),$$

where

$$\mathcal{E}^2(\epsilon_g,\bar{\epsilon}_L,\sigma,\rho_{\max}) = \left(\epsilon_g^2 D_{\tau,\rho} + D_{\tau,\rho}\bar{\epsilon}_L^2 + \sigma^2\right)\cdot G_p' D_{\tau,\rho}\rho_{\max}^2 N + \frac{n}{m}\cdot\frac{1}{\tau}\rho_{\max}^2\sigma^2. \tag{60}$$

Our last step simplifies the overall conditions on $\rho_{\max}$ so that $\mathcal{E}^2(\epsilon_g,\bar{\epsilon}_L,\sigma,\rho_{\max}) = O(1)$:

$$\rho_{\max} \le 1 - \frac{1}{\tau}, \quad G_p' D_{\tau,\rho}^2 \rho_{\max}^2 \le \frac{1}{N}, \quad \rho_{\max} \le \frac{m}{n}\cdot\tau. \tag{61}$$

We claim to fulfill (61) it suffices to require

$$\rho_{\max} \le \frac{1}{4N}\min\{m,\tau-1\}. \tag{62}$$

When $\tau = 1$, the condition trivially requires $\rho_{\max} = 0$. We then consider the case for $\tau \ge 2$. By definition, it can be verified that

$$G_p' \le \frac{1}{m} + \frac{1}{\tau^2}. \tag{63}$$

First notice that

$$\frac{m}{4N} \le \frac{1}{4} \le 1 - \frac{1}{\tau} \quad \text{and} \quad \frac{\tau}{4N} \le \frac{m}{n}\cdot\tau. \tag{64}$$

Therefore, it remains to prove (62) implies

$$G_p' D_{\tau,\rho}^2 \rho_{\max}^2 \le \frac{1}{N}. \tag{65}$$

Using the fact that under (62) $\rho_{\max} \leq 1/4$ we have

$$
\begin{aligned}
G'_p D^2_{\tau,\rho} \rho_{\max} &= G'_p \rho_{\max} \cdot \frac{1}{(1-\rho_{\max})^2} \\
&\leq \frac{16}{9} \rho_{\max} \left( \frac{1}{m} + \frac{1}{\tau^2} \right) \leq \frac{16}{9} \left( \frac{1}{m} + \frac{1}{\tau^2} \right) \frac{1}{4N} \min\{m,\tau\} \leq \frac{1}{N}.
\end{aligned}
\tag{66}
$$

This proves the claim.

## C  PROOF OF MAIN PROPOSITIONS

### C.1  PROOF OF PROPOSITION 1

Denote for short

$$
M^{r,s} \triangleq \begin{pmatrix} \dfrac{1}{N} \displaystyle\sum_{k=1}^{K} \mathbb{E}\|X^{r,s}_{k,\perp}\|^2 \\ \dfrac{1}{K} \displaystyle\sum_{k=1}^{K} \mathbb{E}\|\bar{x}^{r,s}_k - \bar{x}^{r,s}\|^2 \end{pmatrix}.
\tag{67}
$$

Invoking Lemma 2 and Lemma 3 we obtain that under the condition that the learning rate $\eta > 0$ satisfies

$$
\eta^2 \leq \frac{1}{24\tau(4\tau-1)L^2},
\tag{68}
$$

the following inequality is satisfied for all $s = 0, \ldots, \tau - 1$:

$$
M^{r,s+1} \leq G \cdot M^{r,s} + B^{r,s},
\tag{69}
$$

where

$$
G = \begin{pmatrix} \max_{k\in[K]} \rho_k^2(1+\zeta_k^{-1}) + \eta^2\rho_L \cdot 4L^2 & \eta^2\rho_L \cdot 4L^2 \\ 12\tau\eta^2 L^2 & C_\tau \end{pmatrix}
\tag{70}
$$

$$
B^{r,s} = \begin{pmatrix} 4\rho_L\eta^2(\alpha^2\mathbb{E}\|\nabla f(\bar{x}^{r,s})\|^2 + \epsilon_g^2) + 4\eta^2\rho_L\bar{\epsilon}_L^2 + \eta^2\rho_{\max}^2\sigma^2 \\ 12\tau\eta^2(\alpha^2\mathbb{E}\|\nabla f(\bar{x}^{r,s})\|^2 + \epsilon_g^2) + \eta^2\frac{\sigma^2}{n}. \end{pmatrix}
\tag{71}
$$

The inequality in (69) is defined elementwise.

Unrolling (69) yields

$$
M^{r,s+1} \leq \sum_{\ell=0}^{s} G^\ell B^{r,s-\ell},
\tag{72}
$$

where we have used the fact that $M^{r,0} = 0$ due to full synchronization of the $x_i$'s at the beginning of each round $r$.

We first provide a bound on the sum of the two elements of $G^\ell B^{r,s-\ell}$. For simplicity we omit the round index $r$ in the superscript for the rest of this section.

**Lemma 4.** *Let $b_1^{s-\ell}$ and $b_2^{s-\ell}$ be the first and second element of $B^{s-\ell}$, respectively. Suppose the learning rate $\eta > 0$ then*

$$
(1,1)G^\ell B^{s-\ell} \leq \lambda_2^\ell(b_1^{s-\ell} + b_2^{s-\ell}) + \frac{\lambda_2^\ell - \lambda_1^\ell}{\lambda_2 - \lambda_1}\eta^2 \cdot \left(12\tau L^2 b_1^{s-\ell} + 4\rho_L L^2 b_2^{s-\ell}\right)
\tag{73}
$$

*where $\lambda_1 \leq \lambda_2$ are the eigenvalues of $G$; and $\rho_L$ is defined in (31).*

*Proof.* See Appendix D.4. $\qquad\square$

From Lemma 4 we immediately get

$$
\sum_{\ell=0}^{s} (1,1) \cdot G^{\ell} B^{s-\ell}
$$
$$
\leq \sum_{\ell=0}^{s} \left( \lambda_2^{\ell} (b_1^{s-\ell} + b_2^{s-\ell}) + \frac{\lambda_2^{\ell} - \lambda_1^{\ell}}{\lambda_2 - \lambda_1} \eta^2 \cdot \left( 12\tau L^2 b_1^{s-\ell} + 4\rho_L L^2 b_2^{s-\ell} \right) \right).
\tag{74}
$$

Since $\lambda_2 \geq C_\tau > 1$, we have

$$
\frac{\lambda_2^{\ell} - \lambda_1^{\ell}}{\lambda_2 - \lambda_1} = \lambda_2^{\ell-1} \sum_{s=0}^{\ell-1} \left( \frac{\lambda_1}{\lambda_2} \right)^s \leq \lambda_2^{\ell-1} \min \left\{ \frac{\lambda_2}{\lambda_2 - \lambda_1}, \ell \right\} \leq \lambda_2^{\ell} \min \left\{ \frac{1}{\lambda_2 - \lambda_1}, \ell \right\}
\tag{75}
$$

and thus

$$
\sum_{\ell=0}^{s} (1,1) \cdot G^{\ell} B^{s-\ell}
$$
$$
\leq \sum_{\ell=0}^{s} \lambda_2^{\ell} (b_1^{s-\ell} + b_2^{s-\ell}) + \sum_{\ell=0}^{s} \left( \lambda_2^{\ell} \min \left\{ \frac{1}{\lambda_2 - \lambda_1}, \ell \right\} \right) \eta^2 \cdot \left( 12\tau L^2 b_1^{s-\ell} + 4\rho_L L^2 b_2^{s-\ell} \right).
\tag{76}
$$

Recall the definition of $\rho_L$ given by (31):

$$
\rho_L = \max_{k=1,\dots,K} \rho_k^2 (1 + \zeta_k).
\tag{77}
$$

By the Gershgorin's theorem, since $\eta > 0$, we can upperbound $\lambda_2$ as

$$
\lambda_2 \leq \max \left\{ \max_{k \in [K]} \rho_k^2 (1 + \zeta_k^{-1}) + \eta^2 \rho_L \cdot 8L^2, C_\tau + 12\tau \eta^2 L^2 \right\}
$$
$$
\leq \max \left\{ \max_{k \in [K]} \rho_k^2 (1 + \zeta_k^{-1}) + \frac{\rho_L}{(4\tau - 1)3\tau}, 1 + \frac{2}{4\tau - 1} \right\},
\tag{78}
$$

where the last inequality is due to the bound on $\eta$:

$$
\eta^2 \leq \frac{1}{24\tau(4\tau - 1)L^2}.
\tag{79}
$$

Define constant

$$
D_{\tau,\rho} = \min \left\{ \tau, \frac{1}{1 - \rho_{\max}} \right\}.
\tag{80}
$$

We consider two cases.

• Case 1:

$$
\rho_{\max} \leq 1 - \frac{1}{\tau} \Rightarrow \quad \frac{1}{1 - \rho_{\max}} \leq \tau.
\tag{81}
$$

Thus $D_{\tau,\rho} = 1/(1 - \rho_{\max})$. We let $\zeta_k = \rho_k/(1 - \rho_k)$ and it gives

$$
\max_{k \in [K]} \rho_k^2 (1 + \zeta_k^{-1}) = \rho_{\max}, \quad \rho_L = \max_{k=1,\dots,K} \left\{ \frac{\rho_k^2}{1 - \rho_k} \right\} = \frac{\rho_{\max}^2}{1 - \rho_{\max}} = \rho_{\max}^2 D_{\tau,\rho}.
\tag{82}
$$

Substituting into the bound of $\lambda_2$ [cf. (78)] gives

$$
\lambda_2 \leq \max \left\{ \rho_{\max} + \frac{\rho_{\max}^2}{(1 - \rho_{\max})3\tau(4\tau - 1)}, 1 + \frac{2}{4\tau - 1} \right\}
$$
$$
\leq \max \left\{ 1 - \frac{1}{\tau} + \frac{\left( 1 - \frac{1}{\tau} \right)^2}{3(4\tau - 1)}, 1 + \frac{2}{4\tau - 1} \right\} < 1 + \frac{3}{4\tau - 1},
\tag{83}
$$

where in the second inequality we used the condition (81).

Since $s \leq \tau$ and $\lambda_2 \geq 1$, we obtain the following bound

$$\sum_{\ell=0}^{s} \lambda_2^{\ell} b_1^{s-\ell} \leq \left( \left( 1 + \frac{3}{4\tau - 1} \right)^{\tau} \right) \cdot \left( \sum_{\ell=0}^{s} b_1^{\ell} \right) \leq 3 \cdot \left( \sum_{\ell=0}^{s} b_1^{\ell} \right). \tag{84}$$

Moreover, since

$$\rho_{\max} + \eta^2 \rho_L \cdot 4L^2 \leq \rho_{\max} + \frac{\rho_{\max}^2}{(1 - \rho_{\max})(4\tau - 1)6\tau} \overset{(81)}{\leq} 1 - \frac{1}{\tau} + \frac{\left( 1 - \frac{1}{\tau} \right)^2}{6(4\tau - 1)} \leq C_{\tau}, \tag{85}$$

we can bound $\lambda_2 - \lambda_1$ as

$$
\begin{aligned}
\lambda_2 - \lambda_1 &\geq C_{\tau} - \rho_{\max} - \eta^2 \rho_L \cdot 4L^2 \\
&\geq C_{\tau} - \left( \rho_{\max} + \frac{\rho_{\max}^2}{(1 - \rho_{\max})(4\tau - 1)6\tau} \right) \\
&\overset{(81)}{\geq} C_{\tau} - \left( \rho_{\max} + \rho_{\max} \cdot \frac{1 - \frac{1}{\tau}}{6(4\tau - 1)} \right) \\
&\geq 1 + \frac{1}{4\tau - 1} - \left( \rho_{\max} + \rho_{\max} \cdot \frac{1}{4\tau - 1} \right) \\
&= (1 - \rho_{\max}) \left( 1 + \frac{1}{4\tau - 1} \right) \geq 1 - \rho_{\max}.
\end{aligned}
\tag{86}
$$

Collecting (84) and (86) we can bound (76) as

$$
\begin{aligned}
&\sum_{\ell=0}^{s} (1, 1) \cdot G^{\ell} B^{s-\ell} \\
&\leq \sum_{\ell=0}^{s} (b_1^{\ell} + b_2^{\ell}) \cdot 3 + \sum_{\ell=0}^{s} \eta^2 \cdot \left( 12\tau L^2 b_1^{\ell} + 4\rho_L L^2 b_2^{\ell} \right) \cdot 3 \left( \min \left\{ \frac{1}{\lambda_2 - \lambda_1}, \tau \right\} \right) \\
&\leq \sum_{\ell=0}^{s} (b_1^{\ell} + b_2^{\ell}) \cdot 3 + \sum_{\ell=0}^{s} \eta^2 \cdot \left( 12\tau L^2 b_1^{\ell} + D_{\tau,\rho} \rho_{\max}^2 4L^2 b_2^{\ell} \right) \cdot 3 D_{\tau,\rho} \\
&\overset{(79)}{\leq} \sum_{\ell=0}^{s} (b_1^{\ell} + b_2^{\ell}) \cdot 3 + \sum_{\ell=0}^{s} \left( 12\tau b_1^{\ell} + D_{\tau,\rho} \rho_{\max}^2 4 b_2^{\ell} \right) \cdot 3 D_{\tau,\rho} \frac{1}{(4\tau - 1)24\tau} \\
&\leq \sum_{\ell=0}^{s} (b_1^{\ell} + b_2^{\ell}) \cdot 3 + \sum_{\ell=0}^{s} \left( 12\tau b_1^{\ell} + D_{\tau,\rho} \rho_{\max}^2 4 b_2^{\ell} \right) \cdot \frac{1}{8} \frac{D_{\tau,\rho}}{\tau^2} \\
&= \sum_{\ell=0}^{s} (b_1^{\ell} + b_2^{\ell}) \cdot 3 + \sum_{\ell=0}^{s} \left( 12\tau b_1^{\ell} + D_{\tau,\rho} \rho_{\max}^2 4 b_2^{\ell} \right) \cdot \frac{1}{8} \frac{D_{\tau,\rho}}{\tau^2}.
\end{aligned}
\tag{87}
$$

Substituting the expression of $b_1^\ell$ and $b_2^\ell$ gives

$$
\begin{aligned}
\sum_{\ell=0}^{s}(1,1) \cdot G^\ell B^{s-\ell} &\leq \sum_{\ell=0}^{s}(b_1^\ell + b_2^\ell) \cdot 5 + \sum_{\ell=0}^{s}\frac{D_{\tau,\rho}^2}{\tau^2} \cdot \rho_{\max}^2 b_2^\ell \\
&= \sum_{\ell=0}^{s} 5\left(4\eta^2(\rho_L + 3\tau)(\alpha^2\mathbb{E}\|\nabla f(\bar{x}^{r,\ell})\|^2 + \epsilon_g^2) + 4\eta^2\rho_L\bar{\epsilon}_L^2 + \eta^2\rho_{\max}^2\sigma^2 + \eta^2\frac{1}{n}\sigma^2\right) \\
&\quad + \sum_{\ell=0}^{s}\frac{D_{\tau,\rho}^2}{\tau^2} \cdot \rho_{\max}^2 \left(12\tau\eta^2(\alpha^2\mathbb{E}\|\nabla f(\bar{x}^{r,\ell})\|^2 + \epsilon_g^2) + \eta^2\frac{1}{n}\sigma^2\right) \\
&\leq \sum_{\ell=0}^{s}\frac{C_1}{2}\eta^2(\rho_L + \tau + \rho_{\max}^2 D_{\tau,\rho}^2\tau^{-1})(\alpha^2\mathbb{E}\|\nabla f(\bar{x}^{r,\ell})\|^2 + \epsilon_g^2) + \frac{C_1}{2}\eta^2\tau\rho_L\bar{\epsilon}_L^2 \\
&\quad + \frac{C_1}{2}\tau\eta^2\rho_{\max}^2\sigma^2 + \frac{C_1}{2}(\tau + D_{\tau,\rho}^2\tau^{-1}\rho_{\max}^2)\eta^2\frac{1}{n}\sigma^2 \\
&\leq \sum_{\ell=0}^{s} C_1\eta^2\left(\tau + \rho_{\max}^2 D_{\tau,\rho}\right)(\alpha^2\mathbb{E}\|\nabla f(\bar{x}^{r,\ell})\|^2 + \epsilon_g^2) + C_1\eta^2\tau\rho_{\max}^2 D_{\tau,\rho}\bar{\epsilon}_L^2 \\
&\quad + C_1\tau\eta^2\rho_{\max}^2\sigma^2 + C_1(\tau + D_{\tau,\rho}^2\tau^{-1}\rho_{\max}^2)\eta^2\frac{1}{n}\sigma^2
\end{aligned}
\tag{88}
$$

where $C_1$ is some universal constant. The last inequality holds since $\rho_L = \rho_{\max}^2 D_{\tau,\rho}$ and $D_{\tau,\rho} \leq \tau$.

• Case 2:

$$
\rho_{\max} > 1 - \frac{1}{\tau} \Rightarrow D_{\tau,\rho} = \tau.
\tag{89}
$$

In such a case, we let $\zeta_k = (4\tau - 1)$ and thus

$$
\max_{k\in[K]}\rho_k^2(1 + \zeta_k^{-1}) = \rho_{\max}^2(1 + (4\tau - 1)^{-1}), \quad \rho_L = 4\tau\rho_{\max}^2 = 4\rho_{\max}^2 D_{\tau,\rho}.
\tag{90}
$$

Substituting into the bound of $\lambda_2$ given in (78), applying again the learning rate condition (79) and using the fact that $D_{\tau,\rho} = \tau$:

$$
\begin{aligned}
\lambda_2 &\leq \max\left\{\rho_{\max}^2(1 + (4\tau - 1)^{-1}) + \frac{4\rho_{\max}^2}{3(4\tau - 1)}, 1 + \frac{2}{(4\tau - 1)}\right\} \\
&\leq 1 + \frac{3}{4\tau - 1}.
\end{aligned}
\tag{91}
$$

Therefore by (76), (79), (84), and the fact that

$$
\min\left\{\frac{1}{\lambda_2 - \lambda_1}, \ell\right\} \leq \tau = D_{\tau,\rho}
\tag{92}
$$

we obtain

$$
\begin{aligned}
\sum_{\ell=0}^{s}(1,1) \cdot G^\ell B^{s-\ell} \\
\leq \sum_{s=0}^{s}(b_1^\ell + b_2^\ell) \cdot 3 + \sum_{\ell=0}^{s}\eta^2 \cdot \left(12\tau L^2 b_1^\ell + 16\rho_{\max}^2 D_{\tau,\rho}L^2 b_2^\ell\right) \cdot 3D_{\tau,\rho} \\
\leq \sum_{\ell=0}^{s}(b_1^\ell + b_2^\ell) \cdot 3 + \sum_{\ell=0}^{s}\left(12\tau b_1^\ell + 16\rho_{\max}^2 D_{\tau,\rho}b_2^\ell\right)\frac{1}{8}\frac{D_{\tau,\rho}}{\tau^2} \\
\leq \sum_{\ell=0}^{s}(b_1^\ell + b_2^\ell) \cdot 5 + \sum_{\ell=0}^{s} 2\frac{D_{\tau,\rho}^2}{\tau^2}\rho_{\max}^2 b_2^\ell.
\end{aligned}
\tag{93}
$$

Substituting the expression of $b_1$ and $b_2$ and using the fact that

$$
\rho_L = 4\tau\rho_{\max}^2 = 4\rho_{\max}^2 D_{\tau,\rho}
$$

we arrive at the same bound as in Case 1, possibly with a different constant $C_1$.

## C.2 PROOF OF PROPOSITION 2

We are in Case 1 described in the proof of Proposition 1. By letting $\zeta_k = \rho_k/(1 - \rho_k)$ we have

$$G = \begin{pmatrix} \rho_{\max} + \frac{\rho_{\max}^2}{1-\rho_{\max}} \cdot \eta^2 \cdot 4L^2 & \frac{\rho_{\max}^2}{1-\rho_{\max}} \cdot \eta^2 \cdot 4L^2 \\ 12\tau\eta^2 L^2 & C_\tau \end{pmatrix} \tag{94}$$

and the following bound on the difference of the eigenvalues of $G$:

$$\lambda_2 - \lambda_1 \geq 1 - \rho_{\max}. \tag{95}$$

Notice that according to (72) and (148)

$$\frac{1}{N} \sum_{k=1}^K \|X_{k,\perp}^{r,\tau-1}\|^2 = \frac{1}{\det(T)} \sum_{\ell=0}^{\tau-2} t_1^{\tau-2-\ell}.$$

Therefore

$$\frac{1}{N} \sum_{k=1}^K \|X_{k,\perp}^{r,\tau-1}\|^2$$

$$= \frac{1}{12\tau\eta^2 L^2(\lambda_2 - \lambda_1)} \sum_{\ell=0}^{\tau-2} \left( (C_\tau - \lambda_1) \left( \lambda_1^\ell 12\tau\eta^2 L^2 b_1^{\tau-2-\ell} - \lambda_1^\ell(\lambda_2 - C_\tau)b_2^{\tau-2-\ell} \right) \right)$$

$$+ \frac{1}{12\tau\eta^2 L^2(\lambda_2 - \lambda_1)} \sum_{\ell=0}^{\tau-2} \left( (\lambda_2 - C_\tau) \left( \lambda_2^\ell 12\tau\eta^2 L^2 b_1^{\tau-2-\ell} + \lambda_2^\ell(C_\tau - \lambda_1)b_2^{\tau-2-\ell} \right) \right) \tag{96}$$

$$\leq \frac{1}{12\eta^2 L^2 \tau(1 - \rho_{\max})} \sum_{\ell=0}^{\tau-2} \left( (C_\tau - \lambda_1) \lambda_1^\ell \left( 12\tau\eta^2 L^2 b_1^{\tau-2-\ell} - (\lambda_2 - C_\tau)b_2^{\tau-2-\ell} \right) \right)$$

$$+ \frac{1}{12\eta^2 L^2 \tau(1 - \rho_{\max})} \sum_{\ell=0}^{\tau-2} \left( (\lambda_2 - C_\tau) \lambda_2^\ell \left( 12\tau\eta^2 L^2 b_1^{\tau-2-\ell} + (C_\tau - \lambda_1)b_2^{\tau-2-\ell} \right) \right).$$

In the following, we bound $\lambda_1$ and $\lambda_2 - C_\tau$ as a function of $\rho_{\max}$. For notation simplicity we omit the subscript of $\rho_{\max}$ in the rest of the proof. Further, we introduce the following shorthand notation for the elements of $G$:

$$f(\rho) = \rho + \frac{\rho^2}{1-\rho} \cdot \eta^2 \cdot 4L^2, \quad g(\rho) = \frac{\rho^2}{1-\rho} \cdot \eta^2 \cdot 4L^2, \quad \text{and } h(\tau) = 12\tau\eta^2 L^2. \tag{97}$$

Applying the Gershgorin's theorem we obtain

$$\lambda_1 \geq \min\left\{ \rho, C_\tau - 12\tau\eta^2 L^2 \right\} \geq \rho \geq 0, \tag{98}$$

and

$$\lambda_2 \leq \max\{f(\rho) + h(\rho), g(\rho) + C_\tau\}. \tag{99}$$

Under the learning rate condition (79) we can show

$$g(\rho) + C_\tau - (f(\rho) + h(\rho))$$

$$= \frac{\rho^2}{1-\rho} \cdot \eta^2 \cdot 4L^2 + C_\tau - \rho - \frac{\rho^2}{1-\rho} \cdot \eta^2 \cdot 4L^2 - 12\tau\eta^2 L^2 \tag{100}$$

$$= 1 + \frac{3}{2} \cdot \frac{1}{4\tau - 1} - \rho - 12\tau\eta^2 L^2 \geq 0.$$

Therefore,

$$\lambda_2 - C_\tau \leq g(\rho).$$

Substituting the bounds into (96) gives

$$
\frac{1}{N} \sum_{k=1}^{K} \|X_{k,\perp}^{r,\tau-1}\|^2
$$

$$
\leq \frac{1}{12\eta^2 L^2 \tau (1-\rho)} \sum_{\ell=0}^{\tau-2} \left( C_\tau \lambda_1^\ell \left( 12\tau\eta^2 L^2 b_1^{\tau-2-\ell} - (\lambda_2 - C_\tau) b_2^{\tau-2-\ell} \right) \right)
$$

$$
+ \frac{1}{12\eta^2 L^2 \tau (1-\rho)} \sum_{\ell=0}^{\tau-2} \left( (\lambda_2 - C_\tau) \lambda_2^\ell \left( 12\tau\eta^2 L^2 b_1^{\tau-2-\ell} + C_\tau b_2^{\tau-2-\ell} \right) \right)
$$

$$
\leq \frac{1}{12\eta^2 L^2 \tau (1-\rho)} \sum_{\ell=0}^{\tau-2} \left( C_\tau f(\rho)^\ell \left( 12\tau\eta^2 L^2 b_1^{\tau-2-\ell} \right) \right)
$$

$$
\quad (101)
$$

$$
+ \frac{1}{12\eta^2 L^2 \tau (1-\rho)} \sum_{\ell=0}^{\tau-2} \left( g(\rho) \lambda_2^\ell \left( 12\tau\eta^2 L^2 b_1^{\tau-2-\ell} + C_\tau b_2^{\tau-2-\ell} \right) \right)
$$

$$
\leq \frac{1}{1-\rho} C_\tau \left( \sum_{\ell=0}^{\tau-2} b_1^\ell \right) + \frac{1}{1-\rho} g(\rho) \lambda_2^\tau \left( \sum_{\ell=0}^{\tau-2} b_1^\ell \right)
$$

$$
+ \frac{C_\tau}{12\eta^2 L^2 \tau (1-\rho)} g(\rho) \lambda_2^\tau \left( \sum_{\ell=0}^{\tau-2} b_2^\ell \right)
$$

$$
\leq D_{\tau,\rho} C_\tau \left( \sum_{\ell=0}^{\tau-2} b_1^\ell \right) + 12 D_{\tau,\rho}^2 \eta^2 L^2 \rho^2 \left( \sum_{\ell=0}^{\tau-2} b_1^\ell \right) + \frac{C_\tau}{\tau} \rho^2 D_{\tau,\rho}^2 \left( \sum_{\ell=0}^{\tau-2} b_2^\ell \right).
$$

where we have used the bound $\lambda_1 \leq f(\rho) < 1$, $\lambda_2 > 1$ and $\lambda_2^\tau < 3$.

Plug in the expression of $b_1$ and $b_2$ and using the fact that $C_\tau < 2$, $\rho_L = \rho^2 D_{\tau,\rho}$ gives

$$
\frac{1}{N} \sum_{k=1}^{K} \|X_{k,\perp}^{r,\tau-1}\|^2
$$

$$
\leq \left( 2D_{\tau,\rho} + 12 D_{\tau,\rho}^2 \eta^2 L^2 \rho^2 \right) \rho^2 \sum_{s=0}^{\tau-2} \left( 4\eta^2 D_{\tau,\rho} (\alpha^2 \|\nabla f(\bar{x}^{r,s})\|^2 + \epsilon_g^2) + 4\eta^2 D_{\tau,\rho} \bar{\epsilon}_L^2 + \eta^2 \sigma^2 \right)
$$

$$
+ \frac{D_{\tau,\rho}^2 C_\tau}{\tau} \rho^2 \sum_{s=0}^{\tau-2} (12\tau\eta^2 (\alpha^2 \|\nabla f(\bar{x}^{r,s})\|^2 + \epsilon_g^2) + \eta^2 \frac{\sigma^2}{n})
$$

$$
\leq \left( 2D_{\tau,\rho} + \frac{D_{\tau,\rho}^2}{\tau^2} \rho^2 \right) \sum_{s=0}^{\tau-2} \rho^2 \left( 4\eta^2 D_{\tau,\rho} (\alpha^2 \|\nabla f(\bar{x}^{r,s})\|^2 + \epsilon_g^2) + 4\eta^2 D_{\tau,\rho} \bar{\epsilon}_L^2 + \eta^2 \sigma^2 \right)
$$

$$
+ 2 \frac{D_{\tau,\rho}^2}{\tau} \rho^2 \sum_{s=0}^{\tau-2} (12\tau\eta^2 (\alpha^2 \|\nabla f(\bar{x}^{r,s})\|^2 + \epsilon_g^2) + \eta^2 \frac{1}{n} \sigma^2)
$$

$$
\leq 3 D_{\tau,\rho} \rho^2 \sum_{s=0}^{\tau-2} \left( 4\eta^2 D_{\tau,\rho} (\alpha^2 \|\nabla f(\bar{x}^{r,s})\|^2 + \epsilon_g^2) + 4\eta^2 D_{\tau,\rho} \bar{\epsilon}_L^2 + \eta^2 \sigma^2 \right)
$$

$$
+ 2 D_{\tau,\rho}^2 \rho^2 \tau^{-1} \sum_{s=0}^{\tau-2} (12\tau\eta^2 (\alpha^2 \|\nabla f(\bar{x}^{r,s})\|^2 + \epsilon_g^2) + \eta^2 \frac{1}{n} \sigma^2).
$$

$$
\quad (102)
$$

Tidy up the expression gives

$$
\begin{aligned}
\frac{1}{N}\sum_{k=1}^{K}\|X_{k,\perp}^{r,\tau-1}\|^2 &\leq \sum_{s=0}^{\tau-2} 3D_{\tau,\rho}\rho^2\left(4\eta^2 D_{\tau,\rho}(\alpha^2\|\nabla f(\bar{x}^{r,s})\|^2+\epsilon_g^2)+4\eta^2 D_{\tau,\rho}\bar{\epsilon}_L^2+\eta^2\sigma^2\right)\\
&+\sum_{s=0}^{\tau-2} 2D_{\tau,\rho}^2\rho^2\tau^{-1}(12\tau\eta^2(\alpha^2\|\nabla f(\bar{x}^{r,s})\|^2+\epsilon_g^2)+\eta^2\frac{1}{n}\sigma^2)\\
&\leq C_2\sum_{s=0}^{\tau-2}\left(D_{\tau,\rho}^2\rho^2\right)\eta^2(\alpha^2\|\nabla f(\bar{x}^{r,s})\|^2+\epsilon_g^2)\\
&+C_2(D_{\tau,\rho})^2\tau\rho^2\eta^2\bar{\epsilon}_L^2+C_2\tau D_{\tau,\rho}\rho^2\eta^2\sigma^2+C_2(D_{\tau,\rho})^2\rho^2\eta^2\frac{1}{n}\sigma^2\\
&\leq 2C_2\sum_{s=0}^{\tau-2}D_{\tau,\rho}^2\rho^2\eta^2(\alpha^2\|\nabla f(\bar{x}^{r,s})\|^2+\epsilon_g^2)+C_2(D_{\tau,\rho})^2\tau\rho^2\eta^2\bar{\epsilon}_L^2\\
&+C_2\left(\frac{1}{n}\frac{D_{\tau,\rho}}{\tau}+1\right)\rho^2\tau D_{\tau,\rho}\eta^2\sigma^2
\end{aligned}
\tag{103}
$$

for some $C_2>0$.

## C.3 PROOF OF PROPOSITION 4

We prove (47) by splitting the terms $A_{2,1}$ follows:

$$
\begin{aligned}
A_{2,1} &\stackrel{(a)}{=}\mathbb{E}\left\|\frac{1}{Np}\sum_{k=1}^{K}\sum_{i\in\mathcal{V}_k}\mathbb{I}(i\in S_k^r)\left(\sum_{j\in\mathcal{V}_k}(W_k)_{i,j}(x_j^{r,\tau-1}-\eta\nabla f_j(x_j^{r,\tau-1})-\bar{x}_k^{r,\tau-1})\right)\right\|^2\\
&\leq 2\mathbb{E}\left\|\frac{1}{Np}\sum_{k=1}^{K}p\cdot n\eta\nabla\bar{f}_k(\bar{x}_k^{r,\tau-1})\right\|^2+2\mathbb{E}\left\|\frac{1}{Np}\sum_{k=1}^{K}\sum_{i\in\mathcal{V}_k}\mathbb{I}(i\in S_k^r)r_{ik}\right\|^2,
\end{aligned}
\tag{104}
$$

where

$$
r_{ik}\triangleq\sum_{j\in\mathcal{V}_k}(W_k)_{i,j}(x_j^{r,\tau-1}-\bar{x}_k^{r,\tau-1}-\eta(\nabla f_j(x_j^{r,\tau-1})-\nabla\bar{f}_k(\bar{x}_k^{r,\tau-1}))).
\tag{105}
$$

Equality (a) holds since

$$
\begin{aligned}
\sum_{k=1}^{K}\sum_{i\in S_k^r}\sum_{j\in\mathcal{V}_k}(W_k)_{i,j}\bar{x}_k^{r,\tau-1}&=\sum_{k=1}^{K}\sum_{i\in S_k^r}\bar{x}_k^{r,\tau-1}=\sum_{k=1}^{K}m\cdot\bar{x}_k^{r,\tau-1}\\
&=\sum_{k=1}^{K}p\sum_{i\in\mathcal{V}_k}x_i^{r,\tau-1}=Np\bar{x}^{r,\tau-1},
\end{aligned}
\tag{106}
$$

and similarly,

$$
\sum_{i\in S_k^r}\sum_{j\in\mathcal{V}_k}(W_k)_{i,j}\nabla\bar{f}_k(\bar{x}_k^{r,\tau-1})=np\nabla\bar{f}_k(\bar{x}_k^{r,\tau-1}).
\tag{107}
$$

Since samples are taken according to the rule specified by Assumption 6, the following probabilities hold:

$$
\mathbb{P}\big(i\in S_k^r\,|\,i\in\mathcal{V}_k\big)=p,\quad\mathbb{P}\big(i,j\in S_k^r\,|\,i,j\in\mathcal{V}_k\big)=p\cdot\frac{np-1}{n-1},
\tag{108}
$$

$$
\mathbb{P}(i\in S_k^r,j\in S_\ell^r\,|\,i\in\mathcal{V}_k,j\in\mathcal{V}_\ell,k\neq\ell)=p^2.
\tag{109}
$$

Consequently, we can evaluate the second term in (104) and obtain

$$
A_{2,1}=2\mathbb{E}\left\|\frac{1}{K}\sum_{k=1}^{K}\eta\nabla\bar{f}_k(\bar{x}_k^{r,\tau-1})\right\|^2
$$

$$+ \frac{2}{(Np)^2} \mathbb{E} \left( p \sum_{k=1}^{K} \sum_{i \in \mathcal{V}_k} \|r_{ik}\|^2 + p \cdot \frac{np-1}{n-1} \sum_{k=1}^{K} \sum_{i,j \in \mathcal{V}_k} r_{ik}^\top r_{jk} \right)$$

$$+ \frac{2}{(Np)^2} \cdot p^2 \sum_{k \neq \ell} \sum_{i \in \mathcal{V}_k} \sum_{j \in \mathcal{V}_\ell} \mathbb{E}(r_{ik}^\top r_{j\ell})$$

$$= 2\mathbb{E} \left\| \frac{1}{K} \sum_{k=1}^{K} \eta \nabla \bar{f}_k(\bar{x}_k^{r,\tau-1}) \right\|^2$$

$$+ \frac{2}{(Np)^2} \mathbb{E} \left( p \cdot \frac{np-1}{n-1} \left\| \sum_{k=1}^{K} \sum_{i \in \mathcal{V}_k} r_{ik} \right\|^2 + \frac{p(1-p)n}{n-1} \sum_{k=1}^{K} \sum_{i \in \mathcal{V}_k} \|r_{ik}\|^2 \right)$$

$$+ \frac{2}{(Np)^2} \frac{p(1-p)}{n-1} \mathbb{E} \left( \sum_{k \neq \ell} \sum_{i \in \mathcal{V}_k} \sum_{j \in \mathcal{V}_\ell} r_{ik}^\top r_{j\ell} \right)$$

$$\leq 2\mathbb{E} \left\| \frac{1}{K} \sum_{k=1}^{K} \eta \nabla \bar{f}_k(\bar{x}_k^{r,\tau-1}) \right\|^2$$

$$+ \frac{2}{(Np)^2} \mathbb{E} \left( p \cdot \frac{np-1}{n-1} \left\| \sum_{k=1}^{K} \sum_{i \in \mathcal{V}_k} r_{ik} \right\|^2 + \frac{p(1-p)n}{n-1} \sum_{k=1}^{K} \sum_{i \in \mathcal{V}_k} \|r_{ik}\|^2 \right)$$

$$+ \frac{2}{(Np)^2} \frac{p(1-p)}{n-1} \mathbb{E} \left( \sum_{k \neq \ell} \sum_{i \in \mathcal{V}_k} \sum_{j \in \mathcal{V}_\ell} \frac{1}{2}\|r_{ik}\|^2 + \frac{1}{2}\|r_{j\ell}\|^2 \right)$$

$$\leq 2\mathbb{E} \left\| \frac{1}{K} \sum_{k=1}^{K} \eta \nabla \bar{f}_k(\bar{x}_k^{r,\tau-1}) \right\|^2$$

$$+ \frac{2}{(Np)^2} \mathbb{E} \left( p \cdot \frac{np-1}{n-1} \left\| \sum_{k=1}^{K} \sum_{i \in \mathcal{V}_k} r_{ik} \right\|^2 + \frac{p(1-p)n}{n-1} \sum_{k=1}^{K} \sum_{i \in \mathcal{V}_k} \|r_{ik}\|^2 \right)$$

$$+ \frac{2}{(Np)^2} \frac{p(1-p)}{n-1} (K-1)n \sum_{k=1}^{K} \sum_{i \in \mathcal{V}_k} \mathbb{E}\|r_{ik}\|^2. \tag{110}$$

By substituting the expression of $r_{ik}$ we can bound terms $\|\sum_{k=1}^{K} \sum_{i \in \mathcal{V}_k} r_{ik}\|^2$ and $\sum_{k=1}^{K} \sum_{i \in \mathcal{V}_k} \|r_{ik}\|^2$ as

$$\left\| \sum_{k=1}^{K} \sum_{i \in \mathcal{V}_k} r_{ik} \right\|^2 = \left\| \sum_{k=1}^{K} \sum_{i \in \mathcal{V}_k} \sum_{j \in \mathcal{V}_k} (W_k)_{i,j}(x_j^{r,\tau-1} - \bar{x}_k^{r,\tau-1} - \eta(\nabla f_j(x_j^{r,\tau-1}) - \nabla \bar{f}_k(\bar{x}_k^{r,\tau-1}))) \right\|^2$$

$$= \eta^2 \left\| \sum_{k=1}^{K} \sum_{i \in \mathcal{V}_k} \sum_{j \in \mathcal{V}_k} (W_k)_{i,j}(\nabla f_j(x_j^{r,\tau-1}) - \nabla \bar{f}_k(\bar{x}_k^{r,\tau-1})) \right\|^2$$

$$= \eta^2 \left\| \sum_{k=1}^{K} \sum_{i \in \mathcal{V}_k} \sum_{j \in \mathcal{V}_k} (W_k)_{i,j}(\nabla f_j(x_j^{r,\tau-1}) - \nabla f_j(\bar{x}_k^{r,\tau-1})) \right\|^2$$

$$\leq \eta^2 N \sum_{k=1}^{K} \sum_{i \in \mathcal{V}_k} \sum_{j \in \mathcal{V}_k} (W_k)_{i,j} L^2 \|x_j^{r,\tau-1} - \bar{x}_k^{r,\tau-1}\|^2$$

$$\leq \eta^2 N \sum_{k=1}^{K} \sum_{i \in \mathcal{V}_k} L^2 \|x_i^{r,\tau-1} - \bar{x}_k^{r,\tau-1}\|^2 = \eta^2 \cdot NL^2 \sum_{k=1}^{K} \|X_{k,\perp}^{r,\tau-1}\|^2$$

$$\tag{111}$$

and

$$\sum_{k=1}^{K} \sum_{i \in \mathcal{V}_k} \|r_{ik}\|^2$$

$$= \sum_{k=1}^{K} \sum_{i \in \mathcal{V}_k} \left\| \sum_{j \in \mathcal{V}_k} (W_k)_{i,j}(x_j^{r,\tau-1} - \bar{x}_k^{r,\tau-1} - \eta(\nabla f_j(x_j^{r,\tau-1}) - \nabla \bar{f}_k(\bar{x}_k^{r,\tau-1}))) \right\|^2$$

$$\leq \sum_{k=1}^{K} \sum_{i \in \mathcal{V}_k} \sum_{j \in \mathcal{V}_k} (W_k)_{i,j} \left\| x_j^{r,\tau-1} - \bar{x}_k^{r,\tau-1} - \eta(\nabla f_j(x_j^{r,\tau-1}) - \nabla f_j(\bar{x}_k^{r,\tau-1})) \right\|^2$$

$$\leq \sum_{k=1}^{K} \sum_{i \in \mathcal{V}_k} \sum_{j \in \mathcal{V}_k} (W_k)_{i,j} \left( 2 \left\| x_j^{r,\tau-1} - \bar{x}_k^{r,\tau-1} \right\|^2 + 2\eta^2 \left\| \nabla f_j(x_j^{r,\tau-1}) - \nabla f_j(\bar{x}_k^{r,\tau-1}) \right\|^2 \right)$$

$$\leq \sum_{k=1}^{K} \sum_{i \in \mathcal{V}_k} 2 \left\| x_i^{r,\tau-1} - \bar{x}_k^{r,\tau-1} \right\|^2 + 2\eta^2 L^2 \sum_{k=1}^{K} \sum_{i \in \mathcal{V}_k} \| x_i^{r,\tau-1} - \bar{x}_k^{r,\tau-1} \|^2$$

$$= \sum_{k=1}^{K} 2(1 + \eta^2 L^2) \| X_{k,\perp}^{r,\tau-1} \|^2. \tag{112}$$

Tidy up the expression leads to the following bound of $A_{2,1}$:

$$A_{2,1} \leq 2\mathbb{E} \left\| \frac{1}{K} \sum_{k=1}^{K} \eta \nabla \bar{f}_k(\bar{x}_k^{r,\tau-1}) \right\|^2 + \frac{2}{(Np)^2} \left( p \cdot \frac{np-1}{n-1} \mathbb{E} \left\| \sum_{k=1}^{K} \sum_{i \in \mathcal{V}_k} r_{ik} \right\|^2 \right)$$

$$+ \frac{2}{(Np)^2} p(1-p) \frac{N}{n-1} \sum_{k=1}^{K} \sum_{i \in \mathcal{V}_k} \mathbb{E} \|r_{ik}\|^2$$

$$\leq 2\mathbb{E} \left\| \frac{1}{K} \sum_{k=1}^{K} \eta \nabla \bar{f}_k(\bar{x}_k^{r,\tau-1}) \right\|^2 + \frac{2}{(Np)^2} \left( p \cdot \frac{np-1}{n-1} \eta^2 \cdot NL^2 \sum_{k=1}^{K} \mathbb{E} \|X_{k,\perp}^{r,\tau-1}\|^2 \right)$$

$$+ \frac{2}{(Np)^2} p(1-p) \frac{N}{n-1} \left( \sum_{k=1}^{K} 2(1 + \eta^2 L^2) \mathbb{E} \|X_{k,\perp}^{r,\tau-1}\|^2 \right)$$

$$\leq 2\eta^2 \mathbb{E} \left\| \frac{1}{K} \sum_{k=1}^{K} \nabla \bar{f}_k(\bar{x}_k^{r,\tau-1}) \right\|^2 + \frac{2}{N} \eta^2 L^2 \sum_{k=1}^{K} \mathbb{E} \|X_{k,\perp}^{r,\tau-1}\|^2$$

$$+ \frac{4}{N} \frac{1-p}{p(n-1)} \left( \sum_{k=1}^{K} (1 + \eta^2 L^2) \mathbb{E} \|X_{k,\perp}^{r,\tau-1}\|^2 \right)$$

$$\leq 2\eta^2 \mathbb{E} \left\| \frac{1}{K} \sum_{k=1}^{K} \nabla \bar{f}_k(\bar{x}_k^{r,\tau-1}) \right\|^2 + \frac{8}{N} \left( \frac{1-p}{p(n-1)} + \frac{1}{\tau^2} \right) \left( \sum_{k=1}^{K} \mathbb{E} \|X_{k,\perp}^{r,\tau-1}\|^2 \right), \tag{113}$$

where the last inequality holds under the learning rate condition

$$\eta^2 \leq \frac{1}{24\tau(4\tau - 1)L^2}. \tag{114}$$

This completes the proof of (47).

## C.4 Proof of Proposition 5

We bound $A_{2,2}$ in following the same rationale as Proposition 4.

$$
\begin{aligned}
A_{2,2} =& \mathbb{E}\Big\| \frac{1}{Np} \sum_{k=1}^{K} \sum_{i \in \mathcal{V}_k} \mathbb{I}(i \in S_k^r) \underbrace{\Big( \sum_{j \in \mathcal{V}_k} (W_k)_{i,j} (\nabla f_j(x_j^{r,s}) - g_j^{r,s}) \Big)}_{e_{ik}} \Big\|^2 \\
=& \frac{1}{(Np)^2} \left( p \cdot \frac{np-1}{n-1} \mathbb{E}\Big\| \sum_{k=1}^{K} \sum_{i \in \mathcal{V}_k} e_{ik} \Big\|^2 + p(1-p)\frac{n}{n-1} \sum_{k=1}^{K} \sum_{i \in \mathcal{V}_k} \mathbb{E}\|e_{ik}\|^2 \right) \\
&+ \frac{2}{(Np)^2} \frac{p(1-p)}{n-1} \mathbb{E}\left( \sum_{k \neq \ell} \sum_{i \in \mathcal{V}_k} \sum_{j \in \mathcal{V}_\ell} e_{ik}^\top e_{j\ell} \right) \\
=& \frac{1}{(Np)^2} \left( p \cdot \frac{np-1}{n-1} \mathbb{E}\Big\| \sum_{k=1}^{K} \sum_{i \in \mathcal{V}_k} e_{ik} \Big\|^2 + p(1-p)\frac{n}{n-1} \sum_{k=1}^{K} \sum_{i \in \mathcal{V}_k} \mathbb{E}\|e_{ik}\|^2 \right),
\end{aligned}
\tag{115}
$$

where the last equality is due to fact that the inter-cluster stochastic noise is zero mean and independent.

Recall the definition $\xi_i^{r,s} \triangleq g_i^{r,s} - \nabla f_i(x_i^{r,s})$. Using again the independence of the $\xi_i$'s we get

$$
\mathbb{E}\Big\| \sum_{k=1}^{K} \sum_{i \in \mathcal{V}_k} e_{ik} \Big\|^2 = \mathbb{E}\Big\| \sum_{k=1}^{K} \sum_{i \in \mathcal{V}_k} \sum_{j \in \mathcal{V}_k} (W_k)_{i,j} \xi_j^{r,s} \Big\|^2 = \mathbb{E}\Big\| \sum_{i=1}^{N} \xi_i^{r,s} \Big\|^2 = N\sigma^2,
\tag{116}
$$

and

$$
\begin{aligned}
\sum_{k=1}^{K} \sum_{i \in \mathcal{V}_k} \mathbb{E}\|e_{ik}\|^2 &= \sum_{k=1}^{K} \sum_{i \in \mathcal{V}_k} \mathbb{E}\Big\| \sum_{j \in \mathcal{V}_k} (W_k)_{i,j} \xi_j^{r,s} \Big\|^2 = \sum_{k=1}^{K} \|W_k\|^2 \sigma^2 \\
&= \sum_{k=1}^{K} \sigma^2 \sum_{j=1}^{n} d_j^2 \\
&\leq \sum_{k=1}^{K} \sigma^2 \Big(1 + (n-1)\rho_k^2\Big) \leq K\sigma^2 + (n-1)K\rho_{\max}^2 \sigma^2 \\
&= \big(1 + (n-1)\rho_{\max}^2\big) K\sigma^2.
\end{aligned}
\tag{117}
$$

where $d_1 \leq d_2 \leq \cdots \leq d_n = 1$ are the singular values of $W_k$. Therefore,

$$
\begin{aligned}
A_{2,2} &\leq \frac{1}{(Np)^2} \left( p \cdot \frac{np-1}{n-1} \cdot N\sigma^2 + p(1-p)\frac{n}{n-1} \big(1 + (n-1)\rho_{\max}^2\big) K\sigma^2 \right) \\
&= \frac{1}{Np} \left( \frac{np-1}{n-1}\sigma^2 \right) + \frac{1}{Np} \left( \frac{1-p}{n-1} \big(1 + (n-1)\rho_{\max}^2\big) \sigma^2 \right) \\
&\leq \frac{\sigma^2}{N} + \frac{\sigma^2}{N} \frac{p^{-1}-1}{n-1} \big(1 + (n-1)\rho_{\max}^2\big) \leq \frac{\sigma^2}{N} \big(2 + p^{-1}\rho_{\max}^2\big).
\end{aligned}
\tag{118}
$$

The last inequality is due to $p \geq 1/n$.

## C.5 PROOF OF PROPOSITION 6

Invoking the descent inequality Lemma 1 and the error bound for $T_1$-$T_5$ given by Corollary 2, Proposition 3 and Eq. (41):

$$
\begin{aligned}
&\mathbb{E}f(x^{r+1}) \\
&\leq \mathbb{E}f(x^r) - \frac{\eta}{4}\sum_{s=0}^{\tau-1}\mathbb{E}\|\nabla f(\bar{x}^{r,s})\|^2 - \frac{\eta}{4}\sum_{s=0}^{\tau-1}\mathbb{E}\Big\|\frac{1}{N}\sum_{i=1}^{N}\nabla f_i(x_i^{r,s})\Big\|^2 \\
&\quad - \frac{\eta}{4}\sum_{s=0}^{\tau-1}\mathbb{E}\Big\|\frac{1}{K}\sum_{k=1}^{K}\nabla\bar{f}_k(\bar{x}_k^{r,s})\Big\|^2 \\
&\quad + C_1 L^2\eta^3\tau\left(\tau+\rho_{\max}^2 D_{\tau,\rho}\right)\sum_{s=0}^{\tau-1}(\alpha^2\|\nabla f(\bar{x}^{r,s})\|^2+\epsilon_g^2) + C_1 L^2\eta^3\tau^2\rho_{\max}^2 D_{\tau,\rho}\bar{\epsilon}_L^2 \\
&\quad + C_1\tau^2 L^2\eta^3\rho_{\max}^2\sigma^2 + C_1 L^2(\tau^2+D_{\tau,\rho}^2\rho_{\max}^2)\eta^3\frac{\sigma^2}{n} \\
&\quad + \frac{\eta^2 L}{2}\sum_{s=0}^{\tau-1}\mathbb{E}\Big\|\frac{1}{N}\sum_{i=1}^{N}\nabla f_i(x_i^{r,s})\Big\|^2 + \frac{\eta^2 L}{2}\tau\frac{\sigma^2}{N} \\
&\leq \mathbb{E}f(x^r) - \frac{\eta}{4}\sum_{s=0}^{\tau-1}\mathbb{E}\|\nabla f(\bar{x}^{r,s})\|^2 \\
&\quad + C_1 L^2\eta^3\tau\left(\tau+\rho_{\max}^2 D_{\tau,\rho}\right)\sum_{s=0}^{\tau-1}(\alpha^2\mathbb{E}\|\nabla f(\bar{x}^{r,s})\|^2+\epsilon_g^2) + C_1 L^2\eta^3\tau^2\rho_{\max}^2 D_{\tau,\rho}\bar{\epsilon}_L^2 \\
&\quad + C_1\tau^2 L^2\eta^3\rho_{\max}^2\sigma^2 + C_1 L^2(\tau^2+D_{\tau,\rho}^2\rho_{\max}^2)\eta^3\frac{\sigma^2}{n} + \eta^2 L\tau\frac{\sigma^2}{N}.
\end{aligned}
\tag{119}
$$

The last inequality holds under the condition that

$$
\eta \leq \frac{1}{2L}.
\tag{120}
$$

If we further enforce

$$
C_1 L^2\eta^3\tau\left(\tau+\rho_{\max}^2 D_{\tau,\rho}\right)\alpha^2 \leq \frac{\eta}{8} \quad\Leftrightarrow\quad \eta^2 \leq \frac{1}{8C_1 L^2\tau\left(\tau+\rho_{\max}^2 D_{\tau,\rho}\right)\alpha^2},
\tag{121}
$$

then

$$
\begin{aligned}
&\mathbb{E}f(x^{r+1}) \\
&\leq \mathbb{E}f(x^r) - \frac{\eta}{8}\sum_{s=0}^{\tau-1}\mathbb{E}\|\nabla f(\bar{x}^{r,s})\|^2 \\
&\quad + C_1 L^2\eta^3\tau^2\left(\tau+\rho_{\max}^2 D_{\tau,\rho}\right)\epsilon_g^2 + C_1 L^2\eta^3\tau^2\rho_{\max}^2 D_{\tau,\rho}\bar{\epsilon}_L^2 \\
&\quad + C_1\tau^2 L^2\eta^3\rho_{\max}^2\sigma^2 + C_1 L^2(\tau^2+D_{\tau,\rho}^2\rho_{\max}^2)\eta^3\frac{\sigma^2}{n} + \eta^2 L\tau\frac{\sigma^2}{N} \\
&= \mathbb{E}f(x^r) - \frac{\eta}{8}\sum_{s=0}^{\tau-1}\mathbb{E}\|\nabla f(\bar{x}^{r,s})\|^2 + C_1 L^2\eta^3\tau^2\left(\tau+\rho_{\max}^2 D_{\tau,\rho}\right)\epsilon_g^2 \\
&\quad + C_1 L^2\eta^3\left(\tau^2\rho_{\max}^2(D_{\tau,\rho}\bar{\epsilon}_L^2+\sigma^2)+(\tau^2+D_{\tau,\rho}^2\rho_{\max}^2)\frac{\sigma^2}{n}\right) + \eta^2 L\tau\frac{\sigma^2}{N} \\
&\leq \mathbb{E}f(x^r) - \frac{\eta}{8}\sum_{s=0}^{\tau-1}\mathbb{E}\|\nabla f(\bar{x}^{r,s})\|^2 + \eta^2 L\tau\frac{\sigma^2}{N} \\
&\quad + C_1 L^2\eta^3\tau^2\left(\tau+\rho_{\max}^2 D_{\tau,\rho}\right)\epsilon_g^2 + 2C_1 L^2\eta^3\left(\tau^2\rho_{\max}^2 D_{\tau,\rho}\bar{\epsilon}_L^2+\tau^2\sigma^2\left(\frac{1}{n}+\rho_{\max}^2\right)\right),
\end{aligned}
\tag{122}
$$

the last inequality is due to $D_{\tau,\rho}\leq\tau$ and $\rho_{\max}\leq 1$.

## C.6 PROOF OF PROPOSITION 7

Invoking the descent inequality Lemma 1 and the error bound for $T_1$-$T_5$ given by Corollary 2, Proposition 3, and 4:

$$\mathbb{E}f(x^{r+1})$$

$$\leq \mathbb{E}f(x^r) - \frac{\eta}{4}\sum_{s=0}^{\tau-1}\mathbb{E}\|\nabla f(\bar{x}^{r,s})\|^2 - \frac{\eta}{4}\sum_{s=0}^{\tau-1}\mathbb{E}\Big\|\frac{1}{N}\sum_{i=1}^{N}\nabla f_i(x_i^{r,s})\Big\|^2$$

$$- \frac{\eta}{4}\sum_{s=0}^{\tau-1}\mathbb{E}\Big\|\frac{1}{K}\sum_{k=1}^{K}\nabla \bar{f}_k(\bar{x}_k^{r,s})\Big\|^2$$

$$+ \frac{\eta}{4}\sum_{s=0}^{\tau-1}\mathbb{E}\Big\|\nabla f(\bar{x}^{r,s}) - \frac{1}{N}\sum_{i=1}^{N}\nabla f_i(x_i^{r,s})\Big\|^2 + \frac{\eta}{4}\sum_{s=0}^{\tau-1}\mathbb{E}\Big\|\nabla f(\bar{x}^{r,s}) - \frac{1}{K}\sum_{k=1}^{K}\nabla \bar{f}_k(\bar{x}_k^{r,s})\Big\|^2$$

$$+ \frac{\eta}{4}\sum_{s=0}^{\tau-1}\mathbb{E}\Big\|\frac{1}{N}\sum_{i=1}^{N}\nabla f_i(x_i^{r,s}) - \frac{1}{K}\sum_{k=1}^{K}\nabla \bar{f}_k(\bar{x}_k^{r,s})\Big\|^2$$

$$+ \sum_{s=0}^{\tau-2}\mathbb{E}\left(\frac{L}{2}\|\bar{x}^{r,s+1} - \bar{x}^{r,s}\|^2\right) + \mathbb{E}\left(\frac{L}{2}\|x^{r+1} - \bar{x}^{r,\tau-1}\|^2\right)$$

$$\leq \mathbb{E}f(x^r) - \frac{\eta}{4}\sum_{s=0}^{\tau-1}\mathbb{E}\|\nabla f(\bar{x}^{r,s})\|^2 - \frac{\eta}{4}\sum_{s=0}^{\tau-1}\mathbb{E}\Big\|\frac{1}{N}\sum_{i=1}^{N}\nabla f_i(x_i^{r,s})\Big\|^2$$

$$- \frac{\eta}{4}\sum_{s=0}^{\tau-1}\mathbb{E}\Big\|\frac{1}{K}\sum_{k=1}^{K}\nabla \bar{f}_k(\bar{x}_k^{r,s})\Big\|^2$$

$$+ C_1 L^2 \eta^3 \tau \left(\tau + \rho_{\max}^2 D_{\tau,\rho}\right)\sum_{s=0}^{\tau-1}(\alpha^2\|\nabla f(\bar{x}^{r,s})\|^2 + \epsilon_g^2) + C_1 L^2 \eta^3 \tau^2 \rho_{\max}^2 D_{\tau,\rho}\bar{\epsilon}_L^2$$

$$+ C_1 \tau^2 L^2 \eta^3 \rho_{\max}^2 \sigma^2 + C_1 L^2(\tau^2 + D_{\tau,\rho}^2\rho_{\max}^2)\eta^3 \frac{\sigma^2}{n}$$

$$+ \frac{L}{2}\eta^2 \sum_{s=0}^{\tau-2}\mathbb{E}\Big\|\frac{1}{N}\sum_{i=1}^{N}\nabla f_i(x_i^{r,s})\Big\|^2 + \frac{L}{2}\eta^2(\tau-1)\frac{\sigma^2}{N}$$

$$+ L\eta^2 \mathbb{E}\Big\|\frac{1}{K}\sum_{k=1}^{K}\nabla \bar{f}_k(\bar{x}_k^{r,\tau-1})\Big\|^2$$

$$+ \frac{L}{2}\left(G_p + \frac{1}{\tau^2}\right)\left(\frac{8}{N}\sum_{k=1}^{K}\mathbb{E}\|X_{k,\perp}^{r,\tau-1}\|^2\right) + \frac{L}{2}\eta^2\frac{\sigma^2}{N}\left(2 + \frac{n}{m}\rho_{\max}^2\right). \tag{123}$$

Denote for short

$$G_p' \triangleq G_p + \frac{1}{\tau^2}. \tag{124}$$

Further applying the bounds on the consensus error derived in Proposition 2:

$$\mathbb{E}f(x^{r+1})$$

$$\leq \mathbb{E}f(x^r) - \frac{\eta}{4}\sum_{s=0}^{\tau-1}\mathbb{E}\|\nabla f(\bar{x}^{r,s})\|^2 - \frac{\eta}{4}\sum_{s=0}^{\tau-1}\mathbb{E}\Big\|\frac{1}{N}\sum_{i=1}^{N}\nabla f_i(x_i^{r,s})\Big\|^2$$

$$- \frac{\eta}{4}\sum_{s=0}^{\tau-1}\mathbb{E}\Big\|\frac{1}{K}\sum_{k=1}^{K}\nabla \bar{f}_k(\bar{x}_k^{r,s})\Big\|^2$$

$$+ C_1 L^2 \eta^3 \tau \left(\tau + \rho_{\max}^2 D_{\tau,\rho}\right)\sum_{s=0}^{\tau-1}(\alpha^2\|\nabla f(\bar{x}^{r,s})\|^2 + \epsilon_g^2) + C_1 L^2 \eta^3 \tau^2 \rho_{\max}^2 D_{\tau,\rho}\bar{\epsilon}_L^2$$

$$+ C_1 \tau^2 L^2 \eta^3 \rho_{\max}^2 \sigma^2 + C_1 L^2 (\tau^2 + D_{\tau,\rho}^2 \rho_{\max}^2) \eta^3 \frac{\sigma^2}{n}$$

$$+ \frac{L}{2} \eta^2 \sum_{s=0}^{\tau-2} \mathbb{E} \left\| \frac{1}{N} \sum_{i=1}^{N} \nabla f_i(x_i^{r,s}) \right\|^2 + \frac{L}{2} \eta^2 (\tau - 1) \frac{\sigma^2}{N}$$

$$+ L\eta^2 \mathbb{E} \left\| \frac{1}{K} \sum_{k=1}^{K} \nabla \bar{f}_k(\bar{x}_k^{r,\tau-1}) \right\|^2$$

$$+ 4LG_p' \left( 2C_2 \sum_{s=0}^{\tau-2} D_{\tau,\rho}^2 \rho_{\max}^2 \eta^2 (\alpha^2 \|\nabla f(\bar{x}^{r,s})\|^2 + \epsilon_g^2) \right)$$

$$+ 4LG_p' \left( C_2 D_{\tau,\rho}^2 \tau \rho_{\max}^2 \eta^2 \bar{\epsilon}_L^2 + C_2 \left( \frac{1}{n} \frac{D_{\tau,\rho}}{\tau} + 1 \right) \rho_{\max}^2 \tau D_{\tau,\rho} \eta^2 \sigma^2 \right)$$

$$+ \frac{L}{2} \eta^2 \frac{\sigma^2}{N} \left( 2 + \frac{n}{m} \rho_{\max}^2 \right) \tag{125}$$

Rearranging terms and tidy up the expression we have

$$\mathbb{E} f(x^{r+1})$$

$$\leq \mathbb{E} f(x^r) - \frac{\eta}{4} \sum_{s=0}^{\tau-1} \mathbb{E} \|\nabla f(\bar{x}^{r,s})\|^2 - \frac{\eta}{4} \sum_{s=0}^{\tau-1} \mathbb{E} \left\| \frac{1}{N} \sum_{i=1}^{N} \nabla f_i(x_i^{r,s}) \right\|^2$$

$$- \frac{\eta}{4} \sum_{s=0}^{\tau-1} \mathbb{E} \left\| \frac{1}{K} \sum_{k=1}^{K} \nabla \bar{f}_k(\bar{x}_k^{r,s}) \right\|^2 + \frac{L}{2} \eta^2 \sum_{s=0}^{\tau-2} \mathbb{E} \left\| \frac{1}{N} \sum_{i=1}^{N} \nabla f_i(x_i^{r,s}) \right\|^2$$

$$+ L\eta^2 \mathbb{E} \left\| \frac{1}{K} \sum_{k=1}^{K} \nabla \bar{f}_k(\bar{x}_k^{r,\tau-1}) \right\|^2$$

$$+ C_1 L^2 \eta^3 \tau \left( \tau + \rho_{\max}^2 D_{\tau,\rho} \right) \sum_{s=0}^{\tau-1} (\alpha^2 \|\nabla f(\bar{x}^{r,s})\|^2 + \epsilon_g^2) \tag{126}$$

$$+ 8C_2 G_p' L D_{\tau,\rho}^2 \rho_{\max}^2 \eta^2 \left( \sum_{s=0}^{\tau-2} (\alpha^2 \|\nabla f(\bar{x}^{r,s})\|^2 + \epsilon_g^2) \right)$$

$$+ C_1 L^2 \eta^3 \left( \tau^2 D_{\tau,\rho} \rho_{\max}^2 \bar{\epsilon}_L^2 + \tau^2 \rho_{\max}^2 \sigma^2 + (\tau^2 + D_{\tau,\rho}^2 \rho_{\max}^2) \frac{\sigma^2}{n} \right)$$

$$+ \frac{L}{2} (\tau - 1) \eta^2 \frac{\sigma^2}{N} + \frac{L}{2} \left( 2 + \frac{n}{m} \rho_{\max}^2 \right) \eta^2 \frac{\sigma^2}{N}$$

$$+ 4C_2 L G_p' \eta^2 \left( D_{\tau,\rho}^2 \tau \rho_{\max}^2 \bar{\epsilon}_L^2 + \left( \frac{1}{n} \frac{D_{\tau,\rho}}{\tau} + 1 \right) \rho_{\max}^2 \tau D_{\tau,\rho} \sigma^2 \right).$$

Notice that if the following conditions on the learning rate are satisfied

$$\frac{\eta}{4} \geq \eta^2 L,$$
$$\frac{\eta}{8} \geq C_1 L^2 \eta^3 \tau \left( \tau + \rho_{\max}^2 D_{\tau,\rho} \right) \alpha^2 + 8C_2 G_p' L D_{\tau,\rho}^2 \rho_{\max}^2 \eta^2 \alpha^2, \tag{127}$$

then the terms associated to the gradients will be negative and

$$\mathbb{E}f(x^{r+1})$$

$$\leq \mathbb{E}f(x^r) - \frac{\eta}{8}\sum_{s=0}^{\tau-1}\mathbb{E}\|\nabla f(\bar{x}^{r,s})\|^2$$

$$+ C_1 L^2\eta^3\tau^2\left(\tau + \rho_{\max}^2 D_{\tau,\rho}\right)\epsilon_g^2 + 2C_1 L^2\eta^3\left(\tau^2\rho_{\max}^2 D_{\tau,\rho}\bar{\epsilon}_L^2 + \tau^2\sigma^2\left(\frac{1}{n} + \rho_{\max}^2\right)\right)$$

$$+ L\tau\eta^2\frac{\sigma^2}{N} + \left(8C_2 G_p' LD_{\tau,\rho}^2\tau\rho_{\max}^2\right)\eta^2\epsilon_g^2$$

$$+ 4C_2 LG_p'\eta^2\left(D_{\tau,\rho}^2\tau\rho_{\max}^2\bar{\epsilon}_L^2 + \left(\frac{1}{n}\frac{D_{\tau,\rho}}{\tau} + 1\right)\rho_{\max}^2\tau D_{\tau,\rho}\sigma^2\right) + \frac{L}{2}\left(\frac{n}{m}\rho_{\max}^2\right)\eta^2\frac{\sigma^2}{N}.$$

In the last step we clean the condition on the learning rate $\eta$. Collecting all the conditions on $\eta$:

$$\eta^2 \leq \frac{1}{24L^2\tau(4\tau-1)}, \tag{128}$$

$$\frac{\eta}{4} \geq \eta^2 L, \tag{129}$$

$$\frac{\eta}{8} \geq C_1 L^2\eta^3\tau\left(\tau + \rho_{\max}^2 D_{\tau,\rho}\right)\alpha^2 + 8C_2 G_p' LD_{\tau,\rho}^2\rho_{\max}^2\eta^2\alpha^2. \tag{130}$$

Clearly, (128) implies (129). To ensure (130) it suffices to require

$$\frac{\eta}{16} \geq C_1 L^2\eta^3\tau\left(\tau + \rho_{\max}^2 D_{\tau,\rho}\right)\alpha^2$$

$$\frac{\eta}{16} \geq 8C_2 G_p' LD_{\tau,\rho}^2\rho_{\max}^2\eta^2\alpha^2. \tag{131}$$

Recall the definition of $G_p'$:

$$G_p' = G_p + \frac{1}{\tau^2}, \quad \text{and} \quad G_p = \frac{n-m}{m(n-1)}. \tag{132}$$

It can be verified that if

$$\eta \leq \frac{1}{C_3'\alpha^2\tau L}. \tag{133}$$

for some $C_3' > 0$ large enough, then

$$\frac{\eta}{16} \geq C_1 L^2\eta^3\tau\left(\tau + \rho_{\max}^2 D_{\tau,\rho}\right)\alpha^2$$

$$\frac{\eta}{32} \geq 8C_2\left(\frac{1}{\tau^2}\right)LD_{\tau,\rho}^2\rho_{\max}^2\eta^2\alpha^2.$$

It remains to guarantee

$$\frac{\eta}{32} \geq 8C_2 G_p LD_{\tau,\rho}^2\rho_{\max}^2\eta^2\alpha^2. \tag{134}$$

Rearranging terms gives the condition

$$\eta \leq \frac{1}{256C_2 G_p D_{\tau,\rho}^2\,\rho_{\max}^2\alpha^2 L}. \tag{135}$$

Combining with (133) and using the fact that $D_{\tau,\rho}\rho_{\max} \leq \tau$ provides the final condition on $\eta$ as

$$\eta \leq \frac{1}{C_3\alpha\tau L} \cdot \min\left\{1, \frac{1}{\alpha G_p D_{\tau,\rho}\,\rho_{\max}}.\right\} \tag{136}$$

# D  SUPPORTING LEMMAS

## D.1  PROOF OF LEMMA 1

Since the global average of the local copies follows the update [cf. (19)]:

$$\bar{x}^{r,s+1} = \bar{x}^{r,s} - \eta \frac{1}{N} \sum_{i=1}^{N} g_i^{r,s}, \ \forall s = 0, \ldots, \tau - 1. \tag{137}$$

Under Assumption 1, we can apply the descent lemma at points $\bar{x}^{r,s+1}$ and $\bar{x}^{r,s}$ for $s = 0, \ldots, \tau - 2$, conditioned on $\mathcal{F}_{r,s-1}$:

$$\mathbb{E}_{r,s-1} f(\bar{x}^{r,s+1}) \leq f(\bar{x}^{r,s}) + \nabla f(\bar{x}^{r,s})^\top \mathbb{E}_{r,s-1} \left( \bar{x}^{r,s+1} - \bar{x}^{r,s} \right) + \mathbb{E}_{r,s-1} \left( \frac{L}{2} \| \bar{x}^{r,s+1} - \bar{x}^{r,s} \|^2 \right)$$

$$\overset{(a)}{=} f(\bar{x}^{r,s}) - \eta \nabla f(\bar{x}^{r,s})^\top \left( \frac{1}{N} \sum_{i=1}^{N} \nabla f_i(x_i^{r,s}) \right) + \mathbb{E}_{r,s-1} \left( \frac{L}{2} \| \bar{x}^{r,s+1} - \bar{x}^{r,s} \|^2 \right)$$

$$= f(\bar{x}^{r,s}) - \frac{1}{2} \eta \nabla f(\bar{x}^{r,s})^\top \left( \frac{1}{N} \sum_{i=1}^{N} \nabla f_i(x_i^{r,s}) \right)$$

$$- \frac{1}{2} \eta \nabla f(\bar{x}^{r,s})^\top \left( \frac{1}{N} \sum_{i=1}^{N} \nabla f_i(x_i^{r,s}) \pm \frac{1}{K} \sum_{k=1}^{K} \nabla \bar{f}_k(\bar{x}_k^{r,s}) \right) + \mathbb{E}_{r,s-1} \left( \frac{L}{2} \| \bar{x}^{r,s+1} - \bar{x}^{r,s} \|^2 \right)$$

$$\overset{(b)}{\leq} f(\bar{x}^{r,s}) - \frac{\eta}{2} \left( \frac{1}{2} \| \nabla f(\bar{x}^{r,s}) \|^2 + \frac{1}{2} \left\| \frac{1}{N} \sum_{i=1}^{N} \nabla f_i(x_i^{r,s}) \right\|^2 - \frac{1}{2} \left\| \nabla f(\bar{x}^{r,s}) - \frac{1}{N} \sum_{i=1}^{N} \nabla f_i(x_i^{r,s}) \right\|^2 \right)$$

$$- \frac{\eta}{2} \left( \frac{1}{2} \| \nabla f(\bar{x}^{r,s}) \|^2 + \frac{1}{2} \left\| \frac{1}{K} \sum_{k=1}^{K} \nabla \bar{f}_k(\bar{x}_k^{r,s}) \right\|^2 - \frac{1}{2} \left\| \nabla f(\bar{x}^{r,s}) - \frac{1}{K} \sum_{k=1}^{K} \nabla \bar{f}_k(\bar{x}_k^{r,s}) \right\|^2 \right)$$

$$+ \frac{\eta}{2} \| \nabla f(\bar{x}^{r,s}) \| \left\| \frac{1}{N} \sum_{i=1}^{N} \nabla f_i(x_i^{r,s}) - \frac{1}{K} \sum_{k=1}^{K} \nabla \bar{f}_k(\bar{x}_k^{r,s}) \right\|$$

$$+ \mathbb{E}_{r,s-1} \left( \frac{L}{2} \| \bar{x}^{r,s+1} - \bar{x}^{r,s} \|^2 \right)$$

$$\overset{(c)}{\leq} f(\bar{x}^{r,s}) - \frac{\eta}{4} \| \nabla f(\bar{x}^{r,s}) \|^2 - \frac{\eta}{4} \left\| \frac{1}{N} \sum_{i=1}^{N} \nabla f_i(x_i^{r,s}) \right\|^2 - \frac{\eta}{4} \left\| \frac{1}{K} \sum_{k=1}^{K} \nabla \bar{f}_k(\bar{x}_k^{r,s}) \right\|^2$$

$$+ \frac{\eta}{4} \left\| \nabla f(\bar{x}^{r,s}) - \frac{1}{N} \sum_{i=1}^{N} \nabla f_i(x_i^{r,s}) \right\|^2 + \frac{\eta}{4} \left\| \nabla f(\bar{x}^{r,s}) - \frac{1}{K} \sum_{k=1}^{K} \nabla \bar{f}_k(\bar{x}_k^{r,s}) \right\|^2$$

$$+ \frac{\eta}{4} \left\| \frac{1}{N} \sum_{i=1}^{N} \nabla f_i(x_i^{r,s}) - \frac{1}{K} \sum_{k=1}^{K} \nabla \bar{f}_k(\bar{x}_k^{r,s}) \right\|^2 + \mathbb{E}_{r,s-1} \left( \frac{L}{2} \| \bar{x}^{r,s+1} - \bar{x}^{r,s} \|^2 \right),$$

where (a) is due to Assumption 2, (b) is due to $2ab = \|a\|^2 + \|b\|^2 - \|a-b\|^2$ and $ab \leq \|a\| \|b\|$, and (c) is due to $\|a\| \|b\| \leq \frac{1}{2} \|a\|^2 + \frac{1}{2} \|b\|^2$. Notation $a \pm b$ stands for adding and subtracting, i.e., $a \pm b = a + b - b$.

For the pair $(\bar{x}^{r,\tau-1}, x^{r+1})$ we have according to (19) and (22):

$$\mathbb{E}_{r,\tau-2} x^{r+1} = \mathbb{E}_{r,\tau-2}(\bar{x}^{r,\tau}) = \mathbb{E}_{r,\tau-2} \left( \bar{x}^{r,\tau-1} - \eta \frac{1}{N} \sum_{i=1}^{N} g_i^{r,\tau-1} \right).$$

Applying the descent lemma in the same way as before yields

$$\mathbb{E}_{r,\tau-2} f(x^{r+1})$$

$$\leq f(\bar{x}^{r,\tau-1}) + \nabla f(\bar{x}^{r,\tau-1})^\top \mathbb{E}_{r,\tau-2}(x^{r+1} - \bar{x}^{r,\tau-1}) + \frac{L}{2} \mathbb{E}_{r,\tau-2} \| x^{r+1} - \bar{x}^{r,\tau-1} \|^2$$

$$\leq f(\bar{x}^{r,\tau-1}) - \frac{\eta}{4}\|\nabla f(\bar{x}^{r,\tau-1})\|^2 - \frac{\eta}{4}\left\|\frac{1}{N}\sum_{i=1}^{N}\nabla f_i(x_i^{r,\tau-1})\right\|^2 - \frac{\eta}{4}\left\|\frac{1}{K}\sum_{k=1}^{K}\nabla \bar{f}_k(\bar{x}_k^{r,\tau-1})\right\|^2$$

$$+ \frac{\eta}{4}\left\|\nabla f(\bar{x}^{r,\tau-1}) - \frac{1}{N}\sum_{i=1}^{N}\nabla f_i(x_i^{r,\tau-1})\right\|^2 + \frac{\eta}{4}\left\|\nabla f(\bar{x}^{r,\tau-1}) - \frac{1}{K}\sum_{k=1}^{K}\nabla \bar{f}_k(\bar{x}_k^{r,\tau-1})\right\|^2$$

$$+ \frac{\eta}{4}\left\|\frac{1}{N}\sum_{i=1}^{N}\nabla f_i(x_i^{r,\tau-1}) - \frac{1}{K}\sum_{k=1}^{K}\nabla \bar{f}_k(\bar{x}_k^{r,\tau-1})\right\|^2 + \mathbb{E}_{r,\tau-2}\left(\frac{L}{2}\|x^{r+1} - \bar{x}^{r,\tau-1}\|^2\right).$$

Taking expectation, summing over the iterations in round $r$ over $s = 0, \ldots, \tau - 1$ and using the fact that $x^r = \bar{x}^{r,0}$ completes the proof.

### D.2 PROOF OF LEMMA 2

Recall the average update of the $k$-th cluster and that of the global average given by (15) and (19), respectively, for $s = 0, \ldots, \tau - 1$:

$$\bar{x}_k^{r,s+1} = \bar{x}_k^{r,s} - \eta \cdot \bar{g}_k^{r,s} \tag{138}$$

$$\bar{x}^{r,s+1} = \bar{x}^{r,s} - \eta \cdot \frac{1}{N}\sum_{i=1}^{N}g_i^{r,s}. \tag{139}$$

Taking the difference gives

$$\mathbb{E}\|\bar{x}^{r,s+1} - \bar{x}_k^{r,s+1}\|^2$$

$$= \mathbb{E}\left\|(\bar{x}^{r,s} - \bar{x}_k^{r,s}) - \eta\left(\frac{1}{n}\sum_{i\in\mathcal{V}_k}\nabla f_i(x_i^{r,s}) - \frac{1}{N}\sum_{i=1}^{N}\nabla f_i(x_i^{r,s})\right)\right\|^2$$

$$+ \eta^2\mathbb{E}\left\|\frac{1}{n}\sum_{i\in\mathcal{V}_k}\xi_i^{r,s} - \frac{1}{N}\sum_{i=1}^{N}\xi_i^{r,s}\right\|^2 \tag{140}$$

$$\leq (1+\epsilon)\mathbb{E}\|\bar{x}^{r,s} - \bar{x}_k^{r,s}\|^2 + (1+\epsilon^{-1})\eta^2\mathbb{E}\left\|\frac{1}{n}\sum_{i\in\mathcal{V}_k}\nabla f_i(x_i^{r,s}) - \frac{1}{N}\sum_{i=1}^{N}\nabla f_i(x_i^{r,s})\right\|^2$$

$$+ \eta^2\mathbb{E}\left\|\frac{1}{n}\sum_{i\in\mathcal{V}_k}\xi_i^{r,s} - \frac{1}{N}\sum_{i=1}^{N}\xi_i^{r,s}\right\|^2,$$

where $\epsilon > 0$ is some constant to be chosen.

Averaging over $k = 1, \ldots, K$:

$$\frac{1}{K}\sum_{k=1}^{K}\mathbb{E}\|\bar{x}^{r,s+1} - \bar{x}_k^{r,s+1}\|^2$$

$$\leq (1+\epsilon)\frac{1}{K}\sum_{k=1}^{K}\mathbb{E}\|\bar{x}^{r,s} - \bar{x}_k^{r,s}\|^2 + (1+\epsilon^{-1})\eta^2\frac{1}{K}\sum_{k=1}^{K}\mathbb{E}\left\|\frac{1}{n}\sum_{i\in\mathcal{V}_k}\nabla f_i(x_i^{r,s}) - \frac{1}{N}\sum_{i=1}^{N}\nabla f_i(x_i^{r,s})\right\|^2$$

$$+ \eta^2\frac{1}{K}\sum_{k=1}^{K}\mathbb{E}\left\|\frac{1}{n}\sum_{i\in\mathcal{V}_k}\xi_i^{r,s} - \frac{1}{N}\sum_{i=1}^{N}\xi_i^{r,s}\right\|^2$$

$$\overset{(a)}{=} (1+\epsilon)\frac{1}{K}\sum_{k=1}^{K}\mathbb{E}\|\bar{x}^{r,s} - \bar{x}_k^{r,s}\|^2$$

$$+ (1+\epsilon^{-1})\eta^2\left(\frac{1}{K}\sum_{k=1}^{K}\mathbb{E}\left\|\frac{1}{n}\sum_{i\in\mathcal{V}_k}\nabla f_i(x_i^{r,s})\right\|^2 - \mathbb{E}\left\|\frac{1}{N}\sum_{i=1}^{N}\nabla f_i(x_i^{r,s})\right\|^2\right)$$

$$+ \eta^2 \left( \frac{1}{K} \sum_{k=1}^{K} \mathbb{E} \left\| \frac{1}{n} \sum_{i \in \mathcal{V}_k} \xi_i^{r,s} \right\|^2 - \mathbb{E} \left\| \frac{1}{N} \sum_{i=1}^{N} \xi_i^{r,s} \right\|^2 \right)$$

$$\leq (1+\epsilon) \frac{1}{K} \sum_{k=1}^{K} \mathbb{E} \|\bar{x}^{r,s} - \bar{x}_k^{r,s}\|^2 + (1+\epsilon^{-1}) \eta^2 \left( \frac{1}{K} \sum_{k=1}^{K} \mathbb{E} \left\| \frac{1}{n} \sum_{i \in \mathcal{V}_k} \nabla f_i(x_i^{r,s}) \right\|^2 \right) + \eta^2 \frac{K-1}{N} \sigma^2$$

$$= (1+\epsilon) \frac{1}{K} \sum_{k=1}^{K} \mathbb{E} \|\bar{x}^{r,s} - \bar{x}_k^{r,s}\|^2$$

$$+ (1+\epsilon^{-1}) \eta^2 \left( \frac{1}{K} \sum_{k=1}^{K} \mathbb{E} \left\| \frac{1}{n} \sum_{i \in \mathcal{V}_k} \left( \nabla f_i(x_i^{r,s}) \pm \nabla \bar{f}_k(\bar{x}_k^{r,s}) \pm \nabla \bar{f}_k(\bar{x}^{r,s}) \right) \right\|^2 \right) + \eta^2 \frac{K-1}{N} \sigma^2$$

$$\leq (1+\epsilon) \frac{1}{K} \sum_{k=1}^{K} \mathbb{E} \|\bar{x}^{r,s} - \bar{x}_k^{r,s}\|^2 + (1+\epsilon^{-1}) \eta^2 \left( \frac{3}{K} \sum_{k=1}^{K} \mathbb{E} \left\| \frac{1}{n} \sum_{i \in \mathcal{V}_k} \left( \nabla f_i(x_i^{r,s}) - \nabla \bar{f}_k(\bar{x}_k^{r,s}) \right) \right\|^2 \right)$$

$$+ (1+\epsilon^{-1}) \eta^2 \frac{3}{K} \sum_{k=1}^{K} \mathbb{E} \left\| \nabla \bar{f}_k(\bar{x}_k^{r,s}) - \nabla \bar{f}_k(\bar{x}^{r,s}) \right\|^2 + (1+\epsilon^{-1}) \eta^2 \frac{3}{K} \sum_{k=1}^{K} \mathbb{E} \| \nabla \bar{f}_k(\bar{x}^{r,s}) \|^2$$

$$+ \eta^2 \frac{K-1}{N} \sigma^2$$

$$\overset{(b)}{\leq} (1+\epsilon) \frac{1}{K} \sum_{k=1}^{K} \mathbb{E} \|\bar{x}^{r,s} - \bar{x}_k^{r,s}\|^2 + (1+\epsilon^{-1}) \eta^2 \left( \frac{3}{N} \sum_{k=1}^{K} L^2 \mathbb{E} \left\| X_{k,\perp}^{r,s} \right\|^2 \right)$$

$$+ (1+\epsilon^{-1}) \eta^2 \frac{3}{K} \sum_{k=1}^{K} L^2 \mathbb{E} \left\| \bar{x}_k^{r,s} - \bar{x}^{r,s} \right\|^2 + (1+\epsilon^{-1}) \eta^2 \frac{3}{K} \sum_{k=1}^{K} \mathbb{E} \left\| \nabla \bar{f}_k(\bar{x}^{r,s}) \right\|^2 + \eta^2 \frac{K-1}{N} \sigma^2$$

$$= \left( 1 + \epsilon + 3L^2 \eta^2 (1+\epsilon^{-1}) \right) \frac{1}{K} \sum_{k=1}^{K} \mathbb{E} \|\bar{x}^{r,s} - \bar{x}_k^{r,s}\|^2 + (1+\epsilon^{-1}) \eta^2 \left( \frac{3}{N} \sum_{k=1}^{K} L^2 \mathbb{E} \left\| X_{k,\perp}^{r,s} \right\|^2 \right)$$

$$+ (1+\epsilon^{-1}) \eta^2 \frac{3}{K} \sum_{k=1}^{K} \mathbb{E} \left\| \nabla \bar{f}_k(\bar{x}^{r,s}) \right\|^2 + \eta^2 \frac{K-1}{N} \sigma^2. \tag{141}$$

In (a) we used the fact that

$$\frac{1}{K} \sum_{k=1}^{K} \frac{1}{n} \sum_{i \in \mathcal{V}_k} \nabla f_i(x_i^{r,s}) = \frac{1}{N} \sum_{i=1}^{N} \nabla f_i(x_i^{r,s}), \quad \frac{1}{K} \sum_{k=1}^{K} \frac{1}{n} \sum_{i \in \mathcal{V}_k} \xi_i^{r,s} = \frac{1}{N} \sum_{i=1}^{N} \xi_i^{r,s}. \tag{142}$$

and

$$\sum_{i=1}^{K} \|x_i - \bar{x}\|^2 = \sum_{i=1}^{K} \|x_i\|^2 - K \|\bar{x}\|^2 \quad \text{with} \quad \bar{x} = \frac{1}{K} \sum_{k=1}^{K} x_i. \tag{143}$$

In (b) we applied the $L$-smoothness of $f_i$ and $\bar{f}_k$.

Choosing $\epsilon = \frac{1}{4\tau - 1}$ and using the condition that

$$\eta^2 \leq \frac{1}{24\tau(4\tau - 1)L^2}$$

we have

$$\frac{1}{K} \sum_{k=1}^{K} \mathbb{E} \|\bar{x}^{r,s+1} - \bar{x}_k^{r,s+1}\|^2$$

$$\leq \left( 1 + \frac{1}{4\tau - 1} + \frac{1}{2(4\tau - 1)} \right) \frac{1}{K} \sum_{k=1}^{K} \mathbb{E} \|\bar{x}^{r,s} - \bar{x}_k^{r,s}\|^2 + 12\tau \eta^2 L^2 \left( \frac{1}{N} \sum_{k=1}^{K} \mathbb{E} \left\| X_{k,\perp}^{r,s} \right\|^2 \right)$$

$$+ 12\tau\eta^2 \frac{1}{K}\sum_{k=1}^{K}\mathbb{E}\left\|\nabla\bar{f}_k(\bar{x}^{r,s})\right\|^2 + \eta^2\frac{K-1}{N}\sigma^2$$

$$\leq C_\tau \frac{1}{K}\sum_{k=1}^{K}\mathbb{E}\|\bar{x}^{r,s} - \bar{x}_k^{r,s}\|^2 + 12\tau\eta^2 L^2\left(\frac{1}{N}\sum_{k=1}^{K}\mathbb{E}\left\|X_{k,\perp}^{r,s}\right\|^2\right)$$

$$+ 12\tau\eta^2\left(\alpha^2\mathbb{E}\|\nabla f(\bar{x}^{r,s})\|^2 + \epsilon_g^2\right) + \eta^2\frac{K-1}{N}\sigma^2.$$

In the last inequality we applied Assumption 5 on the inter-cluster heterogeneity.

### D.3 PROOF OF LEMMA 3

We follow the perturbed average consensus analysis. Recall the update equation of the consensus error given in (17):

$$X_{k,\perp}^{r,s+1} = (W_k - J)(X_{k,\perp}^{r,s} - \eta G_k^{r,s}). \tag{144}$$

Squaring both sides and conditioning:

$$\mathbb{E}\|X_{k,\perp}^{r,s+1}\|^2 = \mathbb{E}\left(\mathbb{E}\left(\|(W_k - J)(X_{k,\perp}^{r,s} \pm \eta\nabla F_k(X_k^{r,s}) - \eta G_k^{r,s})\|^2|\mathcal{F}_{r,s-1}\right)\right)$$

$$\leq \mathbb{E}\|(W_k - J)(X_{k,\perp}^{r,s} - \eta\nabla F_k(X_k^{r,s}))\|^2 + \eta^2\rho_k^2 n\sigma^2$$

$$\leq \rho_k^2(1 + \zeta_k^{-1})\cdot\mathbb{E}\|X_{k,\perp}^{r,s}\|^2 + \rho_k^2(1 + \zeta_k)\eta^2\mathbb{E}\|\nabla F_k(X_k^{r,s})\|^2 + \eta^2\rho_k^2 n\sigma^2,$$

where $\zeta_k > 0$ is some free parameter to be properly chosen. Next, we bound the norm of the pseudo-gradient $\nabla F_k(X_k^{r,s})$.

$$\|\nabla F_k(X_k^{r,s})\|^2 = \sum_{i\in\mathcal{V}_k}\|\nabla f_i(x_i^{r,s})\|^2$$

$$= \sum_{i\in\mathcal{V}_k}\|\nabla f_i(x_i^{r,s}) \pm \nabla f_i(\bar{x}_k^{r,s}) \pm \nabla\bar{f}_k(\bar{x}_k^{r,s}) \pm \nabla\bar{f}_k(\bar{x}^{r,s})\|^2$$

$$\leq \sum_{i\in\mathcal{V}_k}\left(4\|\nabla f_i(x_i^{r,s}) - \nabla f_i(\bar{x}_k^{r,s})\|^2 + 4\|\nabla f_i(\bar{x}_k^{r,s}) - \nabla\bar{f}_k(\bar{x}_k^{r,s})\|^2 + 4\|\nabla\bar{f}_k(\bar{x}_k^{r,s}) - \nabla\bar{f}_k(\bar{x}^{r,s})\|^2\right)$$

$$+ \sum_{i\in\mathcal{V}_k}4\|\nabla\bar{f}_k(\bar{x}^{r,s})\|^2 \tag{145}$$

$$\leq \sum_{i\in\mathcal{V}_k}\left(4\|\nabla f_i(\bar{x}_k^{r,s}) - \nabla\bar{f}_k(\bar{x}_k^{r,s})\|^2 + 4L^2\|x_i^{r,s} - \bar{x}_k^{r,s}\|^2 + 4L^2\|\bar{x}_k^{r,s} - \bar{x}^{r,s}\|^2 + 4\|\nabla\bar{f}_k(\bar{x}^{r,s})\|^2\right)$$

$$\leq 4L^2\|X_{k,\perp}^{r,s}\|^2 + 4L^2 n\|\bar{x}_k^{r,s} - \bar{x}^{r,s}\|^2 + 4n\|\nabla\bar{f}_k(\bar{x}^{r,s})\|^2 + 4n\epsilon_k^2.$$

The last inequality is due to Assumption 4 on the intra-cluster heterogeneity.

Averaging over $k = 1, \dots, K$ clusters:

$$\frac{1}{N}\sum_{k=1}^{K}\mathbb{E}\|X_{k,\perp}^{r,s+1}\|^2$$

$$\leq \frac{1}{N}\sum_{k=1}^{K}\rho_k^2(1 + \zeta_k^{-1})\mathbb{E}\|X_{k,\perp}^{r,s}\|^2 + \frac{1}{N}\sum_{k=1}^{K}\rho_k^2(1 + \zeta_k)\eta^2\mathbb{E}\|\nabla F_k(X_k^{r,s})\|^2 + \eta^2\left(\frac{1}{K}\sum_{k=1}^{K}\rho_k^2\right)\sigma^2$$

$$\leq \frac{1}{N}\sum_{k=1}^{K}\rho_k^2(1 + \zeta_k^{-1})\mathbb{E}\|X_{k,\perp}^{r,s}\|^2 + \eta^2\frac{1}{N}\sum_{k=1}^{K}\rho_k^2(1 + \zeta_k)\cdot 4L^2\mathbb{E}\|X_{k,\perp}^{r,s}\|^2$$

$$+ \eta^2\frac{1}{K}\sum_{k=1}^{K}\rho_k^2(1 + \zeta_k)4L^2\mathbb{E}\|\bar{x}_k^{r,s} - \bar{x}^{r,s}\|^2 + \eta^2\frac{1}{K}\sum_{k=1}^{K}\rho_k^2(1 + \zeta_k)4\mathbb{E}\|\nabla\bar{f}_k(\bar{x}^{r,s})\|^2$$

$$+ \eta^2\frac{1}{K}\sum_{k=1}^{K}\rho_k^2(1 + \zeta_k)4\epsilon_k^2 + \eta^2\rho_{\max}^2\sigma^2 \tag{146}$$

$$\leq \left( \max_{k \in [K]} \rho_k^2 (1 + \zeta_k^{-1}) + \eta^2 \cdot 4L^2 \underbrace{\max_{k \in [K]} \left\{ \rho_k^2 (1 + \zeta_k) \right\}}_{\rho_L} \right) \frac{1}{N} \sum_{k=1}^K \mathbb{E} \| X_{k,\perp}^{r,s} \|^2$$

$$+ \eta^2 \max_{k \in [K]} \left\{ \rho_k^2 (1 + \zeta_k) \right\} \cdot 4L^2 \frac{1}{K} \sum_{k=1}^K \mathbb{E} \| \bar{x}_k^{r,s} - \bar{x}^{r,s} \|^2$$

$$+ \eta^2 \max_{k \in [K]} \left\{ \rho_k^2 (1 + \zeta_k) \right\} \cdot 4 (\alpha^2 \mathbb{E} \| \nabla f(\bar{x}^{r,s}) \|^2 + \epsilon_g^2) + \eta^2 \frac{1}{K} \sum_{k=1}^K \rho_k^2 (1 + \zeta_k) 4 \epsilon_k^2 + \eta^2 \rho_{\max}^2 \sigma^2.$$

## D.4 PROOF OF LEMMA 4

To simplify the notation we omit the superscript in $B^{r, s-\ell}$ in this section.

Let $\Lambda = \text{diag}(\lambda_1, \lambda_2)$ and the eigendecomposition of $G = T \Lambda T^{-1}$, we can obtain the closed form expression of $T$ as

$$T = \begin{pmatrix} \lambda_1 - C_\tau & \lambda_2 - C_\tau \\ 12 \tau \eta^2 L^2 & 12 \tau \eta^2 L^2 \end{pmatrix}$$

and

$$T^{-1} = \frac{1}{\det(T)} \begin{pmatrix} 12 \tau \eta^2 L^2 & -\lambda_2 + C_\tau \\ -12 \tau \eta^2 L^2 & \lambda_1 - C_\tau \end{pmatrix},$$

where

$$\det(T) = 12 \tau \eta^2 L^2 (\lambda_1 - \lambda_2). \tag{147}$$

Consequently

$$\begin{aligned}
\det(T) \cdot G^\ell B &= \det(T) \cdot T \Lambda^\ell T^{-1} B \\
&= \begin{pmatrix} \lambda_1 - C_\tau & \lambda_2 - C_\tau \\ 12 \tau \eta^2 L^2 & 12 \tau \eta^2 L^2 \end{pmatrix} \begin{pmatrix} \lambda_1^\ell & 0 \\ 0 & \lambda_2^\ell \end{pmatrix} \begin{pmatrix} 12 \tau \eta^2 L^2 & -\lambda_2 + C_\tau \\ -12 \tau \eta^2 L^2 & \lambda_1 - C_\tau \end{pmatrix} \begin{pmatrix} b_1 \\ b_2 \end{pmatrix} \\
&= \begin{pmatrix} \lambda_1 - C_\tau & \lambda_2 - C_\tau \\ 12 \tau \eta^2 L^2 & 12 \tau \eta^2 L^2 \end{pmatrix} \begin{pmatrix} \lambda_1^\ell & 0 \\ 0 & \lambda_2^\ell \end{pmatrix} \begin{pmatrix} 12 \tau \eta^2 L^2 b_1 + (-\lambda_2 + C_\tau) b_2 \\ -12 \tau \eta^2 L^2 b_1 + (\lambda_1 - C_\tau) b_2 \end{pmatrix} \\
&= \begin{pmatrix} \lambda_1 - C_\tau & \lambda_2 - C_\tau \\ 12 \tau \eta^2 L^2 & 12 \tau \eta^2 L^2 \end{pmatrix} \begin{pmatrix} \lambda_1^\ell 12 \tau \eta^2 L^2 b_1 + \lambda_1^\ell (-\lambda_2 + C_\tau) b_2 \\ -\lambda_2^\ell 12 \tau \eta^2 L^2 b_1 + \lambda_2^\ell (\lambda_1 - C_\tau) b_2 \end{pmatrix} \\
&= \begin{pmatrix} t_1 \\ t_2 \end{pmatrix}
\end{aligned} \tag{148}$$

with

$$\begin{aligned}
t_1 &= (\lambda_1 - C_\tau) \left( \lambda_1^\ell 12 \tau \eta^2 L^2 b_1 + \lambda_1^\ell (-\lambda_2 + C_\tau) b_2 \right) \\
&\quad + (\lambda_2 - C_\tau) \left( -\lambda_2^\ell 12 \tau \eta^2 L^2 b_1 + \lambda_2^\ell (\lambda_1 - C_\tau) b_2 \right),
\end{aligned} \tag{149}$$

$$\begin{aligned}
t_2 &= 12 \tau \eta^2 L^2 \left( \lambda_1^\ell 12 \tau \eta^2 L^2 b_1 + \lambda_1^\ell (-\lambda_2 + C_\tau) b_2 \right) \\
&\quad + 12 \tau \eta^2 L^2 \left( -\lambda_2^\ell 12 \tau \eta^2 L^2 b_1 + \lambda_2^\ell (\lambda_1 - C_\tau) b_2 \right).
\end{aligned} \tag{150}$$

Therefore

$$\begin{aligned}
\det(T)(1,1) T \Lambda^\ell T^{-1} B &= t_1 + t_2 \\
&= (\lambda_1 - C_\tau) \left( \lambda_1^\ell L^2 12 \tau \eta^2 b_1 + \lambda_1^\ell (-\lambda_2 + C_\tau) b_2 \right) \\
&\quad + (\lambda_2 - C_\tau) \left( -\lambda_2^\ell L^2 12 \tau \eta^2 b_1 + \lambda_2^\ell (\lambda_1 - C_\tau) b_2 \right) \\
&\quad + L^2 12 \tau \eta^2 \left( \lambda_1^\ell L^2 12 \tau \eta^2 b_1 + \lambda_1^\ell (-\lambda_2 + C_\tau) b_2 \right) \\
&\quad + L^2 12 \tau \eta^2 \left( -\lambda_2^\ell L^2 12 \tau \eta^2 b_1 + \lambda_2^\ell (\lambda_1 - C_\tau) b_2 \right).
\end{aligned} \tag{151}$$

Substituting the expression of $\det(T)$ and dividing both sides of the equality by $12 \tau \eta^2 (\lambda_1 - \lambda_2)$ we have

$L^2 (1,1) T \Lambda^\ell T^{-1} B$

$$
= \frac{1}{12\tau\eta^2(\lambda_1 - \lambda_2)} (\lambda_1 - C_\tau) \left( \lambda_1^\ell L^2 12\tau\eta^2 b_1 + \lambda_1^\ell(-\lambda_2 + C_\tau)b_2 \right)
$$

$$
+ \frac{1}{12\tau\eta^2(\lambda_1 - \lambda_2)} (\lambda_2 - C_\tau) \left( -\lambda_2^\ell L^2 12\tau\eta^2 b_1 + \lambda_2^\ell(\lambda_1 - C_\tau)b_2 \right)
$$

$$
+ \frac{1}{(\lambda_1 - \lambda_2)} L^2 \left( \lambda_1^\ell L^2 12\tau\eta^2 b_1 + \lambda_1^\ell(-\lambda_2 + C_\tau)b_2 \right)
$$

$$
+ \frac{1}{(\lambda_1 - \lambda_2)} L^2 \left( -\lambda_2^\ell L^2 12\tau\eta^2 b_1 + \lambda_2^\ell(\lambda_1 - C_\tau)b_2 \right)
$$

$$
= \frac{1}{(\lambda_1 - \lambda_2)} (\lambda_1 - C_\tau) \lambda_1^\ell L^2 b_1 + \frac{-1}{(\lambda_1 - \lambda_2)} (\lambda_2 - C_\tau) \lambda_2^\ell L^2 b_1
$$

$$
+ \frac{1}{(\lambda_1 - \lambda_2)} L^4 \lambda_1^\ell 12\tau\eta^2 b_1 - \frac{1}{(\lambda_1 - \lambda_2)} L^4 \lambda_2^\ell 12\tau\eta^2 b_1
$$

$$
+ \frac{1}{12\tau\eta^2(\lambda_1 - \lambda_2)} (\lambda_1 - C_\tau) \lambda_1^\ell(-\lambda_2 + C_\tau)b_2 + \frac{1}{12\tau\eta^2(\lambda_1 - \lambda_2)} (\lambda_2 - C_\tau) \lambda_2^\ell(\lambda_1 - C_\tau)b_2
$$

$$
+ \frac{1}{(\lambda_1 - \lambda_2)} L^2 \lambda_1^\ell(-\lambda_2 + C_\tau)b_2 + \frac{1}{(\lambda_1 - \lambda_2)} L^2 \lambda_2^\ell(\lambda_1 - C_\tau)b_2
$$

$$
= \frac{1}{(\lambda_1 - \lambda_2)} \left( -\lambda_2^\ell(\lambda_2 - C_\tau) - \lambda_1^\ell(C_\tau - \lambda_1) \right) L^2 b_1 + \frac{1}{(\lambda_1 - \lambda_2)} 12\tau\eta^2 (\lambda_1^\ell - \lambda_2^\ell) L^4 b_1
$$

$$
+ \frac{1}{12\tau\eta^2(\lambda_1 - \lambda_2)} (\lambda_1 - C_\tau) \lambda_1^\ell(-\lambda_2 + C_\tau)b_2 + \frac{1}{12\tau\eta^2(\lambda_1 - \lambda_2)} (\lambda_2 - C_\tau) \lambda_2^\ell(\lambda_1 - C_\tau)b_2
$$

$$
+ \frac{1}{(\lambda_1 - \lambda_2)} L^2 \lambda_1^\ell(-\lambda_2 + C_\tau)b_2 + \frac{1}{(\lambda_1 - \lambda_2)} L^2 \lambda_2^\ell(\lambda_1 - C_\tau)b_2
$$

$$
\leq \lambda_2^\ell L^2 b_1 + \frac{\lambda_2^\ell - \lambda_1^\ell}{\lambda_2 - \lambda_1} \cdot 12\tau\eta^2 L^4 b_1 + \frac{\lambda_2^\ell - \lambda_1^\ell}{\lambda_2 - \lambda_1} (\lambda_2 - C_\tau)(C_\tau - \lambda_1) \frac{1}{12\tau\eta^2} b_2 + \lambda_2^\ell L^2 b_2, \tag{152}
$$

where in the last inequality we used the fact that $\lambda_1 \leq C_\tau \leq \lambda_2$.

Note that

$$
(\lambda_2 - C_\tau)(C_\tau - \lambda_1)
$$

$$
= - C_\tau^2 - \lambda_1 \lambda_2 + (\lambda_1 + \lambda_2)C_\tau
$$

$$
= - C_\tau^2 - \det(G) + \mathrm{Tr}(G)C_\tau
$$

$$
= - C_\tau^2 - \left( C_\tau(\max_{k \in [K]} \rho_k^2(1 + \zeta_k^{-1}) + \eta^2 \rho_L \cdot 4L^2) - 48\rho_L \tau\eta^4 L^4 \right) \tag{153}
$$

$$
+ C_\tau(\max_{k \in [K]} \rho_k^2(1 + \zeta_k^{-1}) + \eta^2 \rho_L \cdot 4L^2 + C_\tau)
$$

$$
= 48\rho_L \tau\eta^4 L^4.
$$

Therefore, we further obtain

$$
L^2(1,1)T\Lambda^\ell T^{-1}B
$$

$$
\leq \lambda_2^\ell L^2(b_1 + b_2) + \frac{\lambda_2^\ell - \lambda_1^\ell}{\lambda_2 - \lambda_1} \eta^2 \cdot \left( 12\tau L^4 b_1 + 4\rho_L L^4 b_2 \right) \tag{154}
$$

Dividing both sides by $L^2$ completes the proof.

## E  NETWORK CONNECTIVITY CONDITIONS IN THEOREMS AND COROLLARY

Both Theorem 2 and Corollary 1 impose some sufficient conditions on the network connectivity $\rho_{\max}$ for convergence. This can be satisfied in practice as follows. For Theorem 2, as long as $\rho_{\max} < 1$, we can choose $\tau$ large enough so that (7) is fulfilled. Corollary 1 strengthens the result of Theorem 2 by requiring no loss in the order of convergence rate compared to full device participation. This naturally leads to a more stringent condition on $\rho_{\max}$ given by (11). For any given D2D network topology, this can be satisfied by running multiple D2D gossip averaging steps per SGD

update in Algorithm 1. Since the right hand side of (11) depends only on the algorithmic parameters, we can choose the suitable gossip averaging steps to fulfill this condition before launching the algorithm.

## F MORE EXPERIMENTAL DETAILS

In this section, we provide additional experimental results on CIFAR-10 dataset. We follow the same CNN model and non-iid data partition strategy as before and run each experiments for 3 times with different random seeds to report the mean values of best test accuracy. Instead of using a constant learning rate, we decay the local learning rate $\eta$ by half after finishing 50% and 75% of the communication rounds and tune the initial learning rate from $\{0.01, 0.02, 0.05, 0.08, 0.1\}$ for each algorithm.

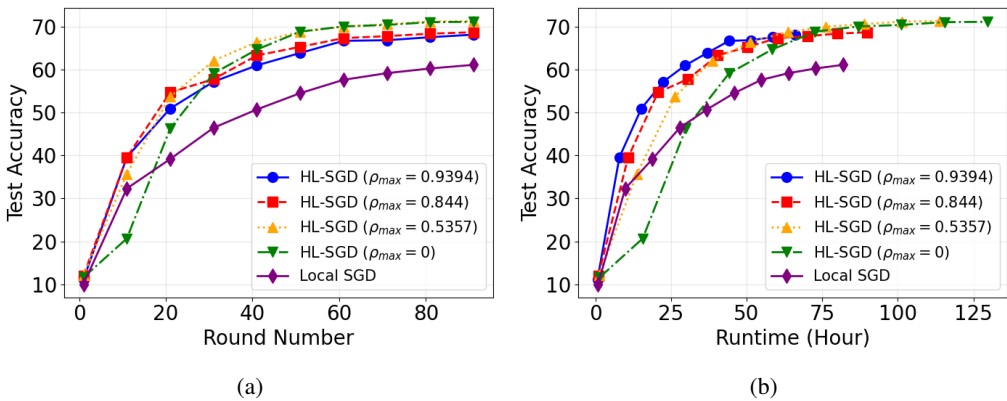

(a)                    (b)

Figure 4: Convergence rate and runtime comparisons of HL-SGD and local SGD on CIFAR-10 under ER random D2D network topology. The device sampling ratio $p = 1/8$ and local iteration period $\tau = 50$.

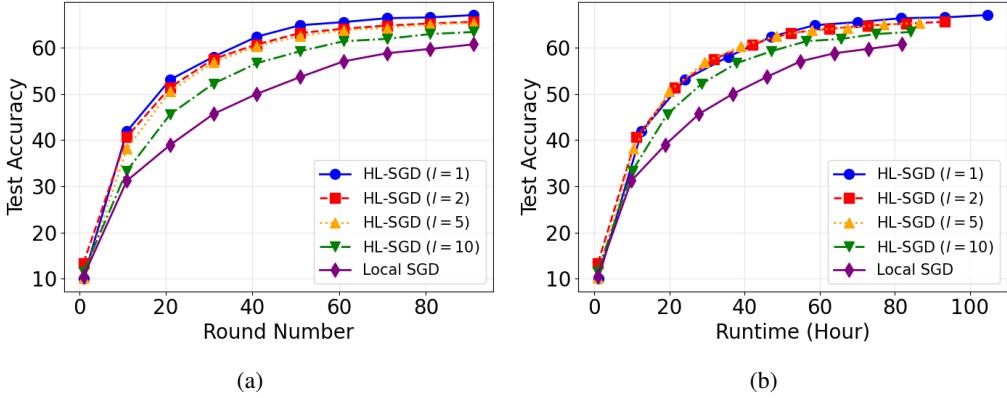

(a)                    (b)

Figure 5: Convergence rate and runtime comparisons of HL-SGD and local SGD on CIFAR-10 under ring topology and multiple SGD updates before gossip averaging. The device sampling ratio $p = 1$, and the local iteration period $\tau = 50$.

First, we evaluate the convergence processes of HL-SGD and local SGD under varying D2D network topologies in Figure 4. We generate random network topologies by Erdős-Rényi model with edge probability from $\{0.2, 0, 5, 0.8, 1\}$ and use Metropolis-Hastings weights to set $W_k$, corresponding to spectral norm $\rho_{\max} = \{0.9394, 0.844, 0.5357, 0\}$. As observed in Figure 4a, a more connected D2D network topology (i.e., a smaller value of $\rho_{\max}$) generally accelerates the convergence and leads to a higher model accuracy achieved over 100 communication rounds in HL-SGD. However, in terms of runtime, a more connected D2D network topology corresponds to a larger D2D communication delay $c_{\text{d2d}}$ per round, and hence the total runtime is larger as well, which can be clearly observed in Figure 4b. Therefore, to achieve a target level of model accuracy within the shortest time in HL-SGD, a sparse D2D network topology could work better than the fully connected one in practice.

Second, we consider an extension of HL-SGD by allowing each device to perform multiple SGD updates before the gossip averaging step in Algorithm 1 and empirically evaluate its performance. Specifically, each device performs $l = \{1, 5, 10\}$ steps of SGD update before aggregating models with their neighbors in the same cluster. Note that $l = 1$ corresponds to the original version of HL-SGD in Algorithm 1. As observed in Figure 5a, when communicating and aggregating models with neighbors more frequently, HL-SGD with $l = 1$ has the best convergence speed and will converge to the highest level of test accuracy. In terms of runtime, choosing a value of $l > 1$ might be favorable in some cases due to the reduced D2D communication delay per round. For instance, to achieve a target level of $60\%$ test accuracy, HL-SGD with $l = 5$ needs $5.22\%$ less amount of time than $l = 1$. It is an interesting direction to rigorously analyze the convergence properties of HL-SGD with arbitrary $l$ and find the best hyperparameter tuning method for minimizing the runtime to achieve a target level of model accuracy in the future.

## G  RELATIONSHIP BETWEEN SPECTRAL GAP AND NETWORK TOPOLOGY

|  | ring/path | 2D-grid | 2-D torus | Erdős-Rényi | exponential |
|---|---|---|---|---|---|
| $1 - \rho$ | $O(1/n^2)$ | $O(1/(n\log n))$ | $O(1/n)$ | $O(1)$ | $O(1/\log(n))$ |

Table 2: Spectral gap of some commonly used graphs, where $n$ denotes the number of nodes. Results are taken from (Nedić et al. (2018); Ying et al. (2021)).

