# OpenReview forum: "Hybrid Local SGD for Federated Learning with Heterogeneous Communications"
_ICLR.cc/2022/Conference — ICLR 2022 Spotlight_

### Official Review · Reviewer_y58n · 2021-11-01

**Correctness:** 4
**Technical Novelty And Significance:** 3
**Empirical Novelty And Significance:** 3
**Recommendation:** 8
**Confidence:** 4

**Main Review:**

The presentation of the paper is very clear and easy to follow. The strength of the paper lies in the theoretical analysis, where combining decentralization and federated average is not straightforward at all. The theoretical result is a bit hard to digest, further clarification and comparison with existing method would make it more readable. The experimental section is also less convincing. My detailed comments are as follows.

1 Theoretical discussion:  it is not clear from the current presentation how to choose the parameter $\tau$ (number of decentralized updates per rounds) Ideally, it would need to balance (i) the D2S communication cost and the D2D communication cost (ii) intra-cluster convergence and outer-loop (round) convergence. In other words, is there any tradeoff between improving the heterogeneous communication versus having better convergence rate. For the discussion on $\rho_{max}$, how it scales with $n$ in a non-extreme situation. The current discussion mostly focus on it being 0 (fully connected) or 1 (no connection), it would be good to discuss its scaling with $n$ in a less trivial situation.

2 Empirical study:  the experiments are performed with a fixed $\tau$. However, the local SGD may have a different regime on the dependency of $\tau$. I encourage the authors to provide more comparison on how $\tau$ influence the performance in figure, for instance with $\tau =5, 10, 20$. The ablation study in Figure 4& 5 is also not convincing as it seems no difference. In particular, changing $\rho$ does not lead to any performance or run time change. This might suggest that the local problem is too simple that be efficiently solved even the connectivity is bad.

3 Privacy concern: the lack of discussion on the privacy issue is a major omission in the paper. The decentralized communication within cluster requires client to exchange information with others, raising potential concerns on leaking informations. I believe this need to be carefully addressed as privacy is one of the major difference in Federated Learning compared to traditional distributed training.

Overall, the theoretical study has merit while the empirical study is less convincing,

**Summary Of The Paper:**

The paper proposes a new method to address the heterogeneous communication setting in a hierarchical distributed system.  The system consists of a central server and several disjoint clusters, and each cluster is a local network connection between multiple edge devices. The communications are heterogeneous in the sense that in cluster device to device (D2D) communication are much faster than the device to server (D2S) communication. The proposes algorithm is a hybrid method where decentralized method is applied within clusters (on edge devices) and federated average is applied between cluster and server. Convergence analysis is provided showing the benefit of variance reduction using decentralization within cluster. Empirical studies are conducted to demonstrate the improvement under synthetic federated learning environment.

**Summary Of The Review:**

The theoretical study has merit while the empirical study is less convincing,

---

> ### Author Response · Authors · 2021-11-17
> **Response to Reviewer y58n (Part 1)**
>
> - **Q1: Theoretical discussion: it is not clear from the current presentation how to choose the parameter  $\tau$ (number of decentralized updates per rounds) Ideally, it would need to balance (i) the D2S communication cost and the D2D communication cost (ii) intra-cluster convergence and outer-loop (round) convergence. In other words, is there any tradeoff between improving the heterogeneous communication versus having better convergence rate. For the discussion on $\rho_{\max}$, how it scales with $n$ in a non-extreme situation. The current discussion mostly focus on it being 0 (fully connected) or 1 (no connection), it would be good to discuss its scaling with $n$ in a less trivial situation.**
>
> - **A1:** Thanks for the interesting question. The convergence rate with full device participation stated in Theorem 1 is obtained for any choice of $\tau$ and $\rho_{\max} \in [0,1]$. As for partial device participation, $\tau$ should be chosen so that (7) in Theorem 2 or (11) in Corollary 1 is satisfied. The impact of $\tau$ and $\rho_{\max}$ on the convergence rate, however, needs to be read from the expression of the complexity. In particular, (6) shows $\rho_{\max}$ is associated to the intra-cluster data heterogeneity and the stochastic gradient variance $\sigma^2$; while the number of local steps $\tau$ is associated to the inter-cluster data heterogeneity $\epsilon_g^2$ and the average stochastic gradient variance $\sigma^2/n$ within each cluster. Improving the D2D network connectivity (corresponding to a smaller $\rho_{\max}$) thus mitigates the impact of both local data heterogeneity and stochastic noise on the convergence rate. This provides the flexibility of choosing a larger $\tau$ so that the overall complexity is invariant.
>
>
>   The discussion on $\rho_{\max}$ in the extreme cases is to show the tightness of the analysis in the sense that it recovers the state-of-the-art convergence rate of Local SGD. The non-extreme case stays in between. The detailed discussion is provided in the **Comparison to Local SGD** part of the main text and cited below for completeness:
>   "Comparing (6) and the complexity of local SGD, we can see the intra-cluster D2D communication provably improves the iteration complexity by reducing the transient iterations. This is reflected in the smaller coefficient associated with the $O(({\tau R})^{-\frac{2}{3}})$ term. In particular, improving D2D communication connectivity will lead to a smaller $\rho_{\max}$ and consequently, mitigate the impact of both local data heterogeneity and stochastic noise on the convergence rate."
>
>
>   The dependency of $\rho$ on $n$, the number of devices in each cluster, has been discussed in literature and is provided in Table 2 of Appendix F in the revised manuscript.

---

> > ### Author Response · Authors · 2021-11-17
> > **Response to Reviewer y58n (Part 2)**
> >
> > - **Q2: Empirical study: the experiments are performed with a fixed $\tau$. However, the local SGD may have a different regime on the dependency of $\tau$. I encourage the authors to provide more comparison on how $\tau$ influence the performance in figure, for instance with $\tau = 5, 10, 20$. The ablation study in Figure 4& 5 is also not convincing as it seems no difference. In particular, changing $\rho$ does not lead to any performance or run time change. This might suggest that the local problem is too simple that be efficiently solved even the connectivity is bad.**
> >
> > - **A2:** Thank you for the suggestion. Here we select $\tau = 50$ because local SGD has the shortest runtime towards the target accuracy at that hyperparameter setting. By fixing $\tau = 50$, we show that local SGD at the optimal setting can be further improved in non-IID data distribution by incorporating D2D communications as in HL-SGD. To give a more comprehensive analysis on the advantages of HL-SGD, we have performed additional experiments by varying $\tau = (5, 10, 20, 50)$ and compared the performances of HL-SGD and local SGD in Figure 5 of Appendix E. The results show that HL-SGD can consistently outperform local SGD across a wide range of $\tau$.
> >
> >
> >   As for Figure 4 & 5 (i.e., Figure 3 in the revised manuscript), sorry for the confusion as there might be some misunderstandings. These figures actually show the impact of $p$ (i.e., sampling ratio), not $\rho_\max$ (i.e., the D2D network connectivity), on the performance of HL-SGD. According to Corollary 1 in the paper, the sampling ratio does not affect the order of convergence rate under condition (11). Therefore, the difference between different sampling ratios is marginal in the experiment results. In practice, we could select a small sampling ratio to save D2S communication and reduce total runtime with little impact on model accuracy.
> >
> >
> >   To see the impact of D2D network topology $\rho$, we have performed additional experiments on CIFAR-10 and added Figure 4 in Appendix E. Specifically, we generate random network topologies by Erdős–Rényi model with edge probability from $(0.2, 0,5, 0.8, 1)$, corresponding to spectral norm $\rho_{\max} = (0.9394, 0.844, 0.5357, 0)$. As shown in the figure, increasing the D2D network connectivity generally speeds up the convergence of HL-SGD in terms of FL rounds but may lead to inferior performance on runtime since the D2D communication delay $c_{d2d}$ per round might be larger under a more connected D2D network topology. When D2D network topology can be configured in some application scenarios, choosing a sparse topology could be better in terms of accuracy-runtime trade-off in HL-SGD.
> >
> > - **Q3: Privacy concern: the lack of discussion on the privacy issue is a major omission in the paper. The decentralized communication within cluster requires client to exchange information with others, raising potential concerns on leaking informations. I believe this need to be carefully addressed as privacy is one of the major difference in Federated Learning compared to traditional distributed training.**
> >
> > - **A3:** Thank you for this insightful comment. Indeed, protection of user privacy has been a major motivation for FL and also our work. It is worth noting that HL-SGD inherits the privacy advantages of FL in the following perspectives: (1) As in traditional FL algorithms (e.g., local SGD), raw data never leaves the device in HL-SGD and only model parameters are exchanged. This greatly reduces the risk of sensitive information leakage; (2) HL-SGD can be integrated with advanced techniques to provide stronger and rigorous privacy protection, including secure aggregation, differential privacy, and shuffling.
> >
> >   As the reviewer has pointed out, HL-SGD and traditional FL algorithms do have different patterns of information exchange. In traditional FL algorithms, the central server receives model parameters from all selected devices in each communication round, which is a major cause for privacy leakage. In HL-SGD, information exchange between device and the server still exists, but the exchanged information could be less sensitive since only in-cluster aggregated models are shared with the server. From this perspective, HL-SGD may provide better privacy protection against the server. There is also device-to-device information exchange in HL-SGD, which shares individual model updates with some other devices in the same cluster. Privacy enhancement mechanisms such as secure aggregation and differential privacy could be applied over these exchanged individual models. The exact privacy guarantee of HL-SGD depends on the adopted privacy definition, threat model, and privacy-enhancing mechanisms, but intuitively we would be able to achieve comparable privacy guarantees since raw data is always kept on device both in traditional FL algorithms and HL-SGD. We have added some discussions and references on privacy in the revised manuscript.

---

> > > ### Comment · Reviewer_y58n · 2021-11-23
> > > **RE: Response to Reviewer y58n (Part 2)**
> > >
> > > Thank you for your detailed response. I appreciate the adding of comparison table, extensive experiments and further discussion on privacy. Just a quick remark, in Figure 3 (c) and (d), the legend is referring to the parameter $\rho$ instead of $p$, that's something mislead my earlier thought. I raise my score accordingly as my concerns are addressed carefully.

---

> > > > ### Author Response · Authors · 2021-11-24
> > > > **RE: RE: Response to Reviewer y58n**
> > > >
> > > > Thank you for pointing out the typo and giving us good suggestions on further improving our manuscript.

---

### Official Review · Reviewer_87Gc · 2021-11-02

**Correctness:** 4
**Technical Novelty And Significance:** 4
**Empirical Novelty And Significance:** 4
**Recommendation:** 8
**Confidence:** 3

**Main Review:**

Strength:
The proposed algorithm takes good advantage of the fast local connectivity among the nodes to improve the convergence. It is a novel contribution.

Detailed convergence analysis of the proposed algorithm is provided.

Weakness:
While the Theoretical results emphasize the role of the local topology in each cluster, the corresponding numerical analysis to understand the impact of topology is missing. It would be interesting to see how the connectivity parameter $\rho$ impacts the results in Figure 1 to 5, instead of just taking a ring topology.

**Summary Of The Paper:**

This paper proposes a new Federated learning algorithm which can take advantage of the fast D2D (Device-to-Device) connections among the devices. In particular, it is shown that the proposed algorithm has better performance in terms of convergence rate and accuracy compared to the existing algorithms such as 'Local SGD'-based FL which rely only on the slow D2S connections for model updating. The convergence of the proposed algorithm is theoretically characterized in terms of the D2D connectivity (topology). Numerical results are provided to show the improved performance compared to the local sgd-based FL.


**Summary Of The Review:**

The proposed algorithm is novel and the the performance is rigorously characterized. The improvement compared to existing algorithms is significant and hence I believe it should be accepted for publication.

---

> ### Author Response · Authors · 2021-11-17
> **Response to Reviewer 87Gc**
>
> - **Q1: While the Theoretical results emphasize the role of the local topology in each cluster, the corresponding numerical analysis to understand the impact of topology is missing. It would be interesting to see how the connectivity parameter  impacts the results in Figure 1 to 5, instead of just taking a ring topology.**
>
> - **A1:** Thank you for the great comments. We have performed additional experiments on CIFAR-10 and added Figure 4 in Appendix E to show the impact of D2D network topology on the performance of HL-SGD. Specifically, we generate random network topologies by Erdős–Rényi model with edge probability from $(0.2, 0,5, 0.8, 1)$, corresponding to spectral norm $\rho_{\max} = (0.9394, 0.844, 0.5357, 0)$. As shown in the figure, increasing the D2D network connectivity generally speeds up the convergence of HL-SGD in terms of FL rounds but may lead to inferior performance on runtime since the D2D communication delay $c_{d2d}$ per round is typically larger under a more connected D2D network topology. When D2D network topology can be configured in some application scenarios, choosing a sparse topology could be better in terms of accuracy-runtime trade-off in HL-SGD. This result is also consistent with the literature on decentralized algorithms [1].
>
>   **Reference**
>
>   [1] Lian et al., “Can Decentralized Algorithms Outperform Centralized Algorithms? A Case Study for Decentralized Parallel Stochastic Gradient Descent”, NIPS 2017.

---

### Official Review · Reviewer_4Rc7 · 2021-11-02

**Correctness:** 3
**Technical Novelty And Significance:** 2
**Empirical Novelty And Significance:** 2
**Recommendation:** 6
**Confidence:** 3

**Main Review:**

The paper proposes to use a hybrid model aggregation to conduct federated learning leveraging both high-speed D2D network and low-speed D2S network.

Pros:
1. The idea proposed in the paper is easy to follow, and the research direction of enhancing the communication efficiency of federated learning is potentially impactful.
2. Both theoretical analysis and empirical results are provided to justify the effectiveness of HL-SGD.
Cons:
1. The main concern I have for the paper is that the proposed idea lacks novelty. Hierarchical federated learning has been studied extensively in the literature [1,6] as well as hierarchical local SGD [3]. Why is HL-SGD fundamentally novel?
2. The theoretical analysis also seems to make sense, but it would be valid to show the comparison among the convergence rates among this work and the prior methods.
3. The scale of the experiments is too small. What are the sizes of the CNNs used for training EMNIST and CIFAR-10 in the experiments? I won’t assume the authors are using large CNNs, but for small-scale models, why will train them to incur communication bottlenecks?
4. Apart from communication efficiency, another big motivation for federated learning is data privacy. Does HL-SGD come with any privacy guarantee? If not, can it be compatible with existing privacy-preserving methods, e.g., Secure Aggregation [4]?
Minor comments:
Typo: “no-iid” —> “non-iid”

Missing references: [4-5]

[1] https://arxiv.org/pdf/1905.06641.pdf

[2] https://arxiv.org/pdf/1909.02362.pdf

[3] https://arxiv.org/abs/2007.13819

[4] https://eprint.iacr.org/2017/281.pdf

[5] https://arxiv.org/abs/2107.06917

[6] https://arxiv.org/pdf/2103.10481.pdf

**Summary Of The Paper:**

This paper introduces HL-SGD, a method to design a hybrid federated learning method to leverage a hybrid high-speed D2D and low-speed D2S network and speed up the communication of federated learning applications. Both theoretical analysis and empirical results are provided to show the effectiveness of HL-SGD.

**Summary Of The Review:**

Overall I think the proposed method is a natural extension from the previous work. And the novelty is quite marginal. The experimental evaluation can be largely improved.

---

### Official Review · Reviewer_6z2f · 2021-11-05

**Correctness:** 3
**Technical Novelty And Significance:** 3
**Empirical Novelty And Significance:** 3
**Recommendation:** 8
**Confidence:** 3

**Main Review:**

Strengths:

1. The paper adds a new dimension to the federated learning setup by combining local and decentralized SGD approaches and showing that doing so can give more accurate models than those obtained with just local SGD.

2. The theoretical results are intuitive and the convergence analysis explores and highlights the important tradeoffs between intra-cluster connectivity, device sampling rate and convergence rate.

Weaknesses:

1. My main concern is that the evaluation is a bit unfair to local SGD. It seems fairly apparent that averaging models within a cluster after every iteration will outperform local SGD where models are only averaged after $\tau$ iterations. Moreover while D2D communication over LAN or across nearby devices may be cheap it is potentially more expensive than averaging computations in a single node or a datacenter network. I also find considering a single choice of $W_k$ in experiments to be a bit limiting (since it seems like the mixing matrix really depends on the setting). Therefore I would suggest the following performing the following additional experiments:

a) D2D communication after multiple SGD updates (the analysis for this may be complicated but it would be instructive to see the empirical performance).

b) Experiments with varying $W_k$ (also do mention the value of $\rho_{\max}$ corresponding to the choice of $W_k$)

2. A couple of claims rely on being able to tune the intra-cluster communication parameters -  $W_k$ (as in Corollary 1) and $c_{\text{d2d}}$ (as in the results in Fig. 3) - to obtain improved performance. Is it really possible to tune these parameters in real world settings? I would suggest adding a citation to support this.

**Summary Of The Paper:**

The paper introduces a new hybrid scheme combining local and decentralized SGD by considering a network with one central node (server) and multiple worker nodes (devices) grouped into clusters with fast intra-cluster communication but slow device to server communication. The authors show that in such settings replacing local SGD with intra-cluster averaging of iterates after each SGD update can lead to faster convergence. Theoretical analysis explores the tradeoff between intra-cluster connectivity, device sampling rate, and convergence rate, while experiments on CIFAR10 and FEMNIST with non-iid splits shows that the proposed approach yields higher test accuracy than local SGD for a given runtime/number of rounds.

**Summary Of The Review:**

The paper definitely adds to the discussion in Federated Learning and by introducing an interesting new setting of hybrid local SGD. Therefore I am leaning towards accepting it. However while the analysis captures the important tradeoffs, the experiments are a bit limited and seem slightly unfair to the local SGD baseline. If these can be rectified (following my suggestions above or otherwise) I would be completely convinced about accepting it

Comments after Rebuttal: Thank you for adequately responding to my queries. I am satisfied with the revisions made to the paper and have updated my score to reflect the same.

---

> ### Author Response · Authors · 2021-11-17
> **Response to Reviewer 6z2f (Part 1)**
>
> - **Q1: My main concern is that the evaluation is a bit unfair to local SGD. It seems fairly apparent that averaging models within a cluster after every iteration will outperform local SGD where models are only averaged after $\tau$ iterations. Moreover, while D2D communication over LAN or across nearby devices may be cheap it is potentially more expensive than averaging computations in a single node or a datacenter network. I also find considering a single choice of $W_k$ in experiments to be a bit limiting (since it seems like the mixing matrix really depends on the setting). Therefore I would suggest performing the following additional experiments:
> a) D2D communication after multiple SGD updates (the analysis for this may be complicated but it would be instructive to see the empirical performance).
> b) Experiments with varying $W_k$ (also do mention the value of $\rho_{max}$ corresponding to the choice of $W_k$)**
>
>
> - **A1**: Thank you for the great suggestions. Yes, by fixing $\tau$ to be the same for both HL-SGD and local SGD in the experiments, it is not hard to imagine that HL-SGD can outperform local SGD in terms of accuracy-round trade-off as shown in Figure 1(a) and Figure 1(c) due to more frequent communications. What is more interesting here is that although D2D communication in HL-SGD introduces additional communication overhead in each round compared with local SGD, HL-SGD can largely outperform local SGD in terms of accuracy-runtime tradeoff as shown in Figure 1(b) and Figure 1(d). That is, HL-SGD can achieve a target model accuracy within a shorter runtime than local SGD, which is a critical requirement in many FL applications.
>
>   Following the suggestions, we have performed additional experiments on CIFAR-10 and added the following experimental results in Appendix E:
>
>   1. In Figure 6 of Appendix E, we empirically evaluate the performance of HL-SGD with multiple steps of SGD before gossip averaging. We allow each device to perform $l = (1, 2, 5, 10)$ steps of SGD update before aggregating models with their neighbors in the same cluster. Note that $l = 1$ corresponds to the original version of HL-SGD in Algorithm 1. The figure shows that HL-SGD with $l = 1$ has the best convergence speed and can converge to the highest level of test accuracy under non-IID data distribution after 100 rounds. In terms of runtime, choosing a value of $l > 1$ might be favorable in some cases due to the reduced D2D communication delay per round. For instance, to achieve a target level of 60% test accuracy, HL-SGD with $l = 5$ needs a slightly less amount of time (5.22\%) than $l = 1$. It is an interesting research direction to rigorously analyze the convergence properties of HL-SGD with arbitrary $l$ and find the best hyperparameter tuning method to minimize the runtime under a target level of accuracy in the future.
>
>   2. In Figure 4 of Appendix E, we show the performance of HL-SGD on different D2D network topologies generated by Erdős–Rényi model
>   with random edge probability in $(0.2, 0,5, 0.8, 1)$, corresponding to spectral norm $\rho_{max} = (0.9394, 0.844, 0.5357, 0)$. As shown in the figure, increasing the D2D network connectivity speeds up the convergence of HL-SGD in terms of FL rounds, matching our convergence results, but may lead to inferior performance on runtime because the D2D communication delay $c_{d2d}$ per round is typically larger under a more connected D2D network topology. When D2D network topology can be configured in some application scenarios, choosing a sparse topology could be better in terms of accuracy-runtime trade-off in HL-SGD. This result is also consistent with the literature on decentralized algorithms [1].
>
>   **Reference**
>
>    [1] Lian et al., “Can Decentralized Algorithms Outperform Centralized Algorithms? A Case Study for Decentralized Parallel Stochastic
>   Gradient Descent”, NIPS 2017.

---

> > ### Author Response · Authors · 2021-11-17
> > **Response to Reviewer 6z2f (Part 2)**
> >
> > - **Q2: A couple of claims rely on being able to tune the intra-cluster communication parameters -
> >  (as in Corollary 1) and  (as in the results in Fig. 3) - to obtain improved performance. Is it really possible to tune these parameters in real world settings? I would suggest adding a citation to support this.**
> >
> >
> > - **A2:** Thanks for your comments and suggestions. The conditions on $W_k$ in Theorem 2 and Corollary 1 are sufficient conditions for convergence when allowing partial device participation at the end of each round. Condition (7) as required in Theorem 2 can be satisfied for any given $\rho_{\max} < 1$ by choosing $\tau$ large enough. This is a tuning parameter of the algorithm that can be set a priori. Corollary 1 strengthens the result of Theorem 2 by requiring no loss in the order of convergence rate compared to full device participation. This naturally leads to a more stringent condition on $\rho_{\max}$ given in (11). Condition (11) can be always satisfied by running multiple D2D gossip averaging steps per SGD update in Algorithm 1. For instance, $h$ steps of D2D gossip averaging improves $\rho_{\max}$ to $(\rho_{\max})^{h}$. According to (11) we can choose the suitable $h$ before launching the algorithm, and thus it can always be satisfied in practice. We have included this clarification in the revised manuscript.
> >
> >
> >
> >      The specific value of $c_{d2d}$ depends on the real-world FL applications. $c_{d2d}$ is determined by network topology and link speed, and fixed for a given D2D network. Figure 3 aims to show that the advantage of HL-SGD becomes more significant when $c_{d2d}$ is smaller, and when $c_{d2d}$ is relatively large compared with $c_{d2s}$, the additional D2D communication overhead may offset the benefit of convergence acceleration in HL-SGD and could lead to inferior runtime performance. Under the real-world measurements of $c_{d2d}$ and $c_{d2s}$ in typical FL applications, HL-SGD will largely outperform local SGD in terms of runtime to achieve a target level of accuracy as shown in Figure 1.

---

> > > ### Comment · Reviewer_6z2f · 2021-11-19
> > > **Comment after rebuttal**
> > >
> > > Thank you for adequately responding to my queries. I am satisfied with the revisions made to the paper and have updated my score to reflect the same.

---

> > > > ### Author Response · Authors · 2021-11-19
> > > > **Response to Reviewer 6z2f**
> > > >
> > > > Thank you for your constructive comments and great suggestions, which are really helpful to us in further improving our manuscript.

---

### Public Comment · ~Eric_James1 · 2021-11-17
**novelty**


It looks this method is identical to "Accelerating Gossip SGD with Periodic Global Averaging".

If combining adjacency matrices of all clusters into a large block-diagonal matrix, these two methods are the same.

Can you please comment on their difference? Thanks.

---

> ### Author Response · Authors · 2021-11-17
> **Response to difference**
>
> Thank you for the comment. The Gossip PGA algorithm in that work assumes a single cluster while we consider multiple clusters, which is motivated by the real-world FL applications where only nearby devices or devices in the same LAN domain can communicate via fast D2D links. Indeed, it is possible to construct an augmented matrix by stacking the $W_k$'s in blocks. This block diagonal augmented matrix $W$ has spectral norm $\rho = \|W - (1/N) \mathbf{1} \mathbf{1}^\top\|_2 = 1$. The convergence analysis provided by Gossip PGA is applicable but does not exploit the block diagonal structure since it is done for a general consensus matrix $W$. More specifically, the complexity of Gossip GPA is
>
> $O\left( \frac{\sigma}{\sqrt{N \tau R}} + \frac{ C_{\tau,\rho}^{\frac{1}{3} }  D_{\tau,\rho}^{\frac{1}{3} } \rho^{\frac{2}{3}} \epsilon^{\frac{2}{3}}}{ (\tau R)^{\frac{2}{3}} } + \frac{C_{\tau,\rho}^{\frac{1}{3}} \rho^{\frac{2}{3}} \sigma^{\frac{2}{3}}}{(\tau R)^{\frac{2}{3}}} + \frac{\rho D_{\tau, \rho}}{\tau R}\right),$
>
> where $C_{\tau, \rho} \triangleq \sum_{k = 0}^{\tau - 1} \rho^k$, $D_{\tau, \rho} = \min\{1/(1 - \rho), \tau\}$, and $\rho$ is the connectivity of $W$. Letting $\rho \to 1$ (since the overall network is disconnected), the complexity becomes
> $$
> O \left( \frac{{\sigma}}{\sqrt{N \tau R}}  + \frac{\left(  \tau^2  \epsilon^2  +  \tau   \sigma^2 \right)^{\frac{1}{3}}}{  {(\tau R})^{\frac{2}{3}}} + \frac{\tau}{ \tau R} \right).
> $$
> This coincides with the complexity of Local SGD. The effect of D2D networks is not reflected in the expression. Our analysis, on the other hand, exploits the block diagonal structure of $W$ and provides the explicit dependency of the rate on $\rho_{\max}$ -- the worst-case connectivity of the D2D subgraphs. When all the  D2D networks are connected, our rate is more informative and better than the rate of Gossip GPA as described above. See Table 1 and the related comments for the comparison with Local SGD. In fact, understanding the impact of D2D networks on the convergence rate is the key question we would like to address in this work.
>
> Moreover, considering the unique communication challenge in FL, our algorithm uses device sampling at the end of each round to reduce the uplink D2S communication cost, while Gossip PGA requires all devices to upload models to the server. Experimental results show that this technique can reduce the required runtime to achieve a target level of model accuracy in HL-SGD.

---

> > ### Public Comment · ~Eric_James1 · 2021-11-17
> > **Thanks for your reply**
> >
> > Thanks for your reply. I have a simple follow-up question. Why is the spectral norm of the block diagonal augmented matrix equal to 1?

---

> > > ### Author Response · Authors · 2021-11-17
> > > **spectral gap of $W$**
> > >
> > > Thanks. We apologize that in our previous reply the wording is not precise. The claim is: the spectral norm of matrix $W - (1/N) \mathbf{1}_N \mathbf{1}_N^\top$, defined as $\rho$, is equal to 1.
> > >
> > > For example, consider a $2n \times 2n$ matrix
> > > $$
> > >     W = (
> > >     W_1,  0;
> > >     0, W_2),
> > > $$
> > > where $W_1, W_2 \in \mathbb{R}^{n \times n}$ are symmetric bi-stochastic matrices assigned to two connected D2D subgraphs satisfying the standard Assumption 3 in the paper.
> > > In addition, we construct vector $x = ( \mathbf{1}_n; -   \mathbf{1}_n)$.
> > >
> > > Then
> > > $$
> > >     y  = ( W -  (1/N) \mathbf{1}_{N} \mathbf{1}_N^\top) x =       ( \mathbf{1}_n; -   \mathbf{1}_n),
> > > $$
> > > where $N = 2n$.
> > >
> > > Therefore, by definition
> > > $$
> > >     \|W - (1/N) \mathbf{1}_N \mathbf{1}_N^\top \|_2 = \sup \Big( \frac{ \|( W - (1/N) \mathbf{1}_N \mathbf{1}_N^\top ) x\|_2  }{ \|x\|_2} : x \in \mathbb{R}^{N}, x \neq 0 \Big) \geq 1.
> > > $$
> > > The equality follows from the properties of bi-stochastic matrix.
> > >
> > > Generally, $\rho = 1$ is due to the fact that the graph of all devices is disconnected even though the subgraphs are connected.
> > > Does this address your concern?

---

### Author Response · Authors · 2021-11-21
**Response to All Reviewers**

We want to thank all the reviewers for giving many constructive comments and great suggestions to our manuscript. For the most recent uploaded version, the major changes compared with the submitted original version are summarized as follows:

1. We have performed additional experiments and added the following new experimental results in Appendix E: (i) Figure 4 shows the performance of HL-SGD on different D2D network topologies generated by Erdős–Rényi model; (ii) Figure 5 shows the performance of HL-SGD on different $\tau$; and (iii) Figure 6 shows the performance of an extension of HL-SGD by allowing each device to perform multiple updates before the gossip averaging step.

2. We have clarified the differences of HL-SGD with prior hierarchical FL algorithms in the “backgrounds and related work” section.

3. We have expanded the discussions after Theorem 1 and added Table 1 to compare HL-SGD with prior methods (local SGD, gossip SGD, and gossip GPA) relevant to our problem setting in their algorithmic complexities. The tightness of our derived bounds and insights about the hyperparameter choices in HL-SGD are further discussed.

4. We have added discussions on the privacy protection properties of HL-SGD and more details about the experimental settings.

We believe that this new version of the paper has addressed the reviewers' comments.

---

### Decision · Program_Chairs · 2022-01-20

**Decision:**

Accept (Spotlight)

**Comment:**

The paper contributes to the literature on federated learning by introducing a hybrid local SGD (HL-SGD) method. HL-SGD is motivated by the setups where edge devices are grouped into clusters with fast connections within the cluster, but slower connection between the devices and the server. HL-SGD uses hybrid updates: decentralized updates within the clusters and federated averaging steps between the clusters and the server.

Initially, the reviews expressed concerns regarding comparison to prior work, empirical results, and privacy of the proposed scheme. However, the authors adequately addressed all of the concerns, added relevant discussions and results to the paper, and a consensus was reached that the paper should be accepted.